# Mechanisms and interventions promoting healthy frontostriatal dynamics in obsessive-compulsive disorder

Sebastien Naze [1] ✉, Luke J. Hearne [1], Paula Sanz-Leon[1,2], Conor Robinson [1], Caitlin V. Hall[1], Saurabh Sonkusare [1], Bjorn Burgher[1], Andrew Zalesky[3,4], James A. Roberts [1] & Luca Cocchi [1,5] ✉

Changes in the frontostriatal system activity support individuals' perseverance in distressful thoughts and rigid, repetitive behaviours that define obsessive-compulsive disorder (OCD). Converging evidence from preclinical and clinical work suggests that OCD maps onto a functional imbalance in the ventral and dorsal frontostriatal circuits. However, the neural mechanisms supporting these dysregulations remain elusive, their association with symptom severity is unclear, and therapeutic interventions are limited. To address these gaps, we combined neuroimaging and behavioural data from individuals with OCD and controls with computational modelling. We found that bidirectionally decreasing spontaneous neural coupling in the ventromedial circuit while concurrently increasing dorsolateral cortico-striatal coupling delivered the highest functional improvements in OCD. The analysis of longitudinal changes in obsessions and compulsions with respect to modelled neural interventions supported our predictions. By highlighting behaviourally meaningful neural mechanisms hidden from traditional neuroimaging analysis, this study advances knowledge on the neural basis of OCD and provides new therapeutic targets for obsessions and compulsions.

Obsessive-compulsive disorder (OCD) is a condition characterised by recurring intrusive thoughts (obsessions) and set repetitive behaviours (compulsions) that consume hours every day[1]. Deregulation of frontostriatal brain circuit activity has been reliably associated with the emergence of OCD[2], as well as behaviours that transcend nosological boundaries and support adaptive emotional and cognitive functions[3]. In line with preclinical[4,5] and neurosurgical studies[6–9], neuroimaging work has shown increased resting-state functional connectivity between the ventral striatum and the medial prefrontal cortex and reduced connectivity between the dorsal striatum and the lateral prefrontal cortex in OCD[2,10–12]. Further, anxiety and impulsivity associated with OCD

diagnosis have also been hypothesised to emerge from this circuit imbalance[13,14].

While progress toward understanding OCD pathology has accelerated in recent years[2], the neural mechanisms driving imbalances in frontostriatal circuit activity in OCD remain unclear. This knowledge is essential to develop targeted therapeutic interventions to reduce OCD symptom severity[15,16]. Preclinical work suggests a key role for inhibitory balance in striatal microcircuit dynamics and its coupling to prefrontal activity[17,18]. However, directly translating these preclinical results to individuals with OCD remains a challenge[19]. For example, animal models lack the human brain's structural and functional complexity, particularly in pathways comprising the prefrontal cortex[20,21].

[1]QIMR Berghofer, Brisbane, Queensland, Australia. [2]School of Physics, Faculty of Science, The University of Sydney, Sydney, New South Wales, Australia. [3]Department of Psychiatry, The University of Melbourne, Melbourne, Victoria, Australia. [4]Department of Biomedical Engineering, The University of Melbourne, Melbourne, Victoria, Australia. [5]School of Biomedical Sciences, Faculty of Health, Medicine and Behavioural Sciences, University of Queensland, Brisbane, Queensland, Australia. ✉e-mail: sebastien.naze@qimrberghofer.edu.au; luca.cocchi@qimrberghofer.edu.au

Biophysical models provide a viable way to link preclinical advances at the microscale with clinical findings at the macroscale[22–25]. Thus, models have the potential to advance knowledge on the neural basis of symptoms in people with OCD and fast-track the development of therapies by delivering testable predictions on the macroscale effects of changes in specific neural mechanisms. Accordingly, stochastic dynamic causal modelling (DCM) has shown that changes in resting-state fMRI connectivity in OCD relate to altered neural couplings between striatal and frontal cortices differing along the ventro-dorsal axis[12]. Such an approach previously facilitated the development of model-informed pilot interventions[26], leveraging recent progress in neuromodulation for depression[27]. However, DCM employs a generic attractor model to generate regional resting-state activity[28], which does not explicitly capture the complex functional interplay within and between striatal and frontal regions affected by OCD. On the other hand, recent phenomenological models provide elaborate statistical frameworks to explain state transitions in the OCD brain or behaviours, but lack an explicit mapping to the neurobiology and neuroimaging data[29,30].

Here, we develop a biophysical model of OCD frontostriatal pathology using dynamical systems theory, Bayesian inference, and a unique longitudinal dataset comprising clinical and neuroimaging information[12,16,31]. Our model was built to capture key neural processes thought to underlie deregulations in the OCD frontostriatal system. Formalising these neurobiological hypotheses into a tractable mathematical model using a top-down approach balancing biological plausibility and complexity, we predict the changes in intrinsic neural activity that could restore healthy frontostriatal functional connectivity in OCD subjects. Because of their key relevance for OCD pathophysiology[1,2], we simulate ventral and dorsal frontostriatal circuits by building on an established model summarising the evolution of excitatory and inhibitory neural activity[32,33]. The model captures essential aspects of local neural activity and cross-region coupling, explaining macroscopic patterns of resting-state functional connectivity measured via functional magnetic resonance imaging (fMRI). Critically, it provides a coarse-grained description of core neural processes known to be impacted by OCD, including changes in the basal ganglia functional microcircuits[17,34] and frontostriatal neural couplings[4,35,36]. The model incorporates key anatomical constraints linked to pathways connecting the striatum with the frontal cortex[12,37]. This mesoscale model allows the estimation of neural changes within the frontostriatal system by identifying neural mechanisms underpinning altered fMRI resting-state activity in OCD and their relationship to symptom severity.

Our work identifies neural parameters that support changes in fMRI resting-state frontostriatal connectivity in OCD compared to healthy controls. These include top-down cortico-striatal and bottom-up striato-cortical couplings and their associated drift (relaxation towards the mean activity) and volatility (variance in the neural signal). Using a formal Bayesian optimisation method, we isolate statistically relevant parameters and predict their contribution to restoring healthy frontostriatal activity. We then test those predictions using a longitudinal dataset and a digital twin paradigm. By bridging scales from neural to behavioural changes in OCD, our work advances knowledge about the mechanisms underlying obsessions and compulsions. The presented study provides a solid basis for the development of more complex models interrogating OCD frontostriatal pathology and facilitating progress towards targeted interventions aiming to restore healthy frontostriatal dynamics.

## Results

We used a clinical dataset comprising neuroimaging and psychiatric evaluations of subjects diagnosed with OCD with age-, gender- and IQ-matched healthy controls (Methods and Table 1). Figure 1 illustrates the approach. In brief, we estimated resting-state connectivity from the data and developed a computational model of the frontostriatal system optimised to reproduce functional connectivity observed in OCD and healthy controls. The optimisation step highlighted model parameters underpinning differences in OCD functional connectivity compared to controls. We then used a combinatorial approach to systematically simulate virtual interventions on the OCD model parameters, allowing the isolation of key parameters likely supporting OCD frontostriatal pathology. Predictive measures of the efficacy of the virtual interventions and the contribution of model parameters to restore healthy functional connectivity were derived from this combinatorial analysis. We validated those predictions by leveraging longitudinal data, confirming the core model parameters associated with changes in functional connectivity and their associated fluctuations in OCD symptoms over time.

### A simplified model of frontostriatal brain dynamics

Based on information regarding key brain regions and resting-state connectivity patterns linked to OCD[2,10,12,14] (Supplementary Figs. 1 and 2), we modelled the frontostriatal system through interacting ventromedial (nucleus accumbens – orbitofrontal cortex) and dorsolateral (putamen – lateral prefrontal cortex) functional circuits[38,39]. A potential concern with our approach is that the precise anatomical location of functional connectivity clusters showing a group difference may slightly differ between neuroimaging studies. For example, Harrison et al.[10] reported OCD hyperconnectivity between the nucleus accumbens and the left anteromedial orbitofrontal cortex and the anterior cingulate. However, Naze et al.[12] reported a more circumscribed hyperconnectivity comprising the nucleus accumbens and the right frontal pole. We interrogated these apparent discrepancies by re-analysing the Naze et al.[12] data using a more lenient statistical threshold ($p < 0.001$ uncorrected rather than $p_{FWE} < 0.05$). In line with Harrison et al., results from this control analysis supported the presence of increased connectivity between the nucleus accumbens and the left anteromedial orbitofrontal cortex, as well as the anterior cingulate in OCD (Supplementary Fig. 1). Likewise, the dorsolateral circuit hypoconnectivity peaks reported by Harrison et al.[10] and Naze et al.[12] are also up to 2 cm apart but align when using a less stringent threshold (Supplementary Fig. 2). Crucially, our control analyses confirmed that group changes in functional connectivity across the two studies map onto the canonical ventromedial frontostriatal circuit[13,14].

We used an established neural model[32,33] in which the net activity of each brain region, modelled by their firing rates, approximated the evolution of local inhibitory and excitatory neural populations ("Methods"). Preliminary to the analysis of coupled ventral and dorsal frontostriatal circuits and key to the understanding of the model's

## Table 1 | Sample characteristics

| | Controls (n = 45) | OCD (n = 52) |
|---|---|---|
| Age | 32.5 (8.7) | 30.2 (7.9) |
| Sex (% female) | 40% | 44% |
| Handedness (% right) | 96% | 85% |
| IQ-Full scale | 112.7 (11.3) | 106.0 (12.5) |
| Y-BOCS Total | 1.8 (3.0) | 25.3 (5.2)* |
| HAMA | 2.9 (3.1) | 19.5 (8.6) # |
| HADS Anxiety–Depression | 4.8 (3.7) – 2.2 (2.3) | 13.3 (4.7) – 8.2 (4.8) &^ |
| MADRS | 2.9 (3.5) | 19.5 (10.4) + |

Unless otherwise indicated, mean and standard deviation (S.D.) are reported. Y-BOCS: Yale-Brown Obsessive Compulsive Scale, IQ-Full scale: Wechsler Adult Intelligence Scale (WAIS-IV), MADRS: Montgomery-Asberg Depression rating scale; HAMA: Hamilton Anxiety Rating Scale; HADS: Hospital Anxiety and Depression scale. For digital twin analysis (n = 48), pre–post, *Y-BOCS 25.1 (4.8) – 20.4 (6.1); #HAMA 19.9 (9.0) – 13.4 (8.6); &HADS Anxiety 13.0 (4.7) – 11.2 (4.0); ^HADS Depression 8.0 (4.8) – 6.3 (4.2); +MADRS 19.5 (10.5) – 15.3 (10.6).

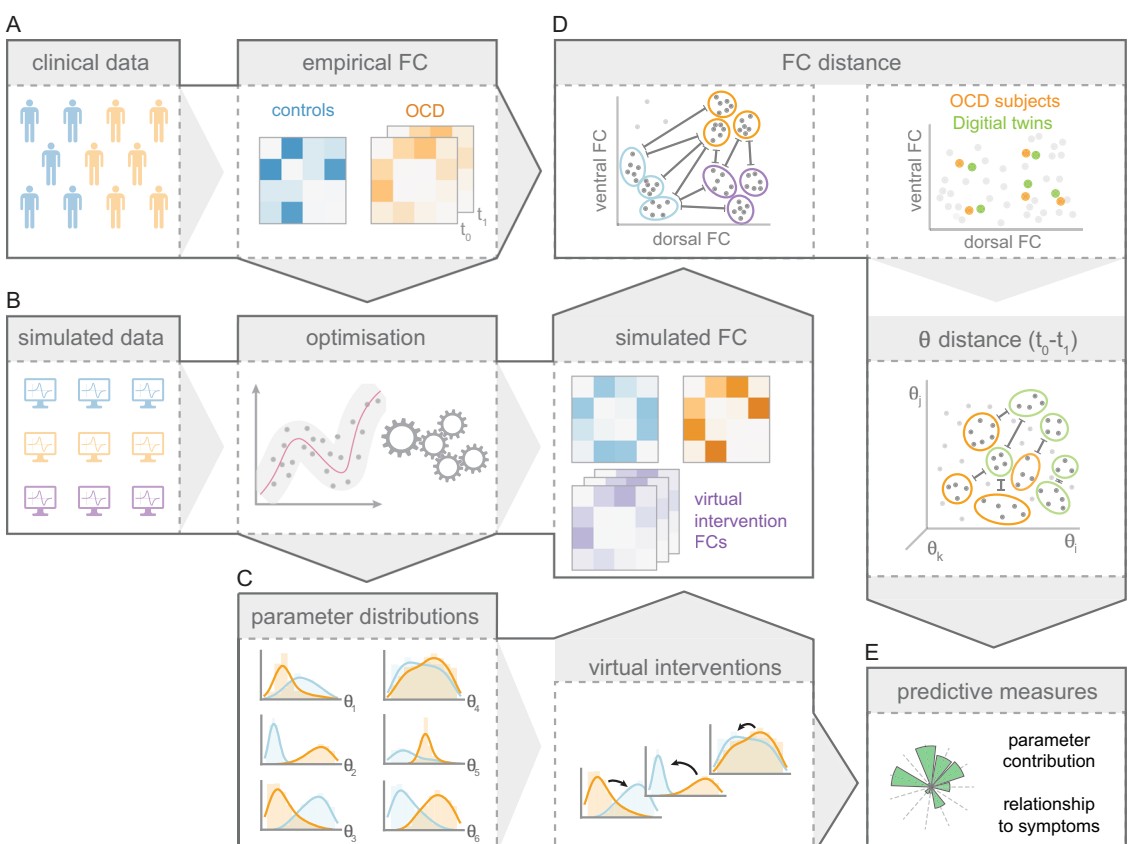

**Fig. 1 | Schematic of the analysis steps and modelling framework. A** Resting-state functional connectivity (FC) of the frontostriatal system was computed from subjects with OCD and matched healthy controls. **B** Frontostriatal resting-state neural dynamics were simulated using our computational model. The model parameters were optimised to fit the empirical FC of the two groups. **C** The optimisations produced group-specific distributions of the model parameters (denoted by **θ**) and simulated FC. Parameter differences correspond to core neural mechanisms underpinning OCD frontostriatal pathology. Virtual interventions were performed by permuting sets of model parameters from OCD subjects (orange) to distributions observed in healthy controls (blue). **D** Subjects with OCD (real data) were paired to their digital twins (simulated data) via a distance measure in FC space (Methods). **E** Predictions of model parameters that contribute the most to normalise FC were derived from the virtual interventions. These predictions were tested by assessing the relationship between changes in the isolated neural parameters, FC, and fluctuations of obsessive and compulsive symptoms over time.

dynamics, we first considered a simplified circuit composed of one striatal and one frontal region (Fig. 2A). The stability analysis of this single circuit revealed the existence of three distinct regimes: a monostable state of low firing rates in both regions, a monostable state of high firing rates in both regions, and a bistable state of coexisting low and high firing rates reflecting the potential for a dynamic switch between distinct activity states[40] (Supplementary Section I). The monostable state of low firing dominates the system for mutually inhibitory brain regions, as well as when one region is excitatory while the other is inhibitory. The monostable state of high firing is only observed when both regions exert strong mutual excitation on each other. These monostable states cannot generate the rich neural dynamics necessary to optimally process information. On the other hand, the bistable regime (Fig. 2C, D) is particularly interesting as it permits spontaneous transitions between stable activity states under minimal neural perturbations. State transitions are central to the modelling of transient dynamics supporting many neural processes, including decision-making and executive functions[41–45]. Accordingly, we introduced a stochastic coupling process ("Methods"), allowing the spontaneous shifts between dynamical states to occur. This process approximates the dynamic contribution of biological pathways promoting or inhibiting neural couplings between striatal regions and the frontal cortex (Fig. 2B). In this updated model, striato-cortical projections result from the product of the striatal activity with a dynamic coupling strength that approximates the interplay between direct and indirect pathways of the basal ganglia and its net thalamo-cortical effect. We introduced three new parameters to describe this dynamic coupling from striatal to frontal regions: mean strength, relaxation toward the mean (drift), and variance of endogenous fluctuations (volatility). The drift parameter ($\eta_{ij}$) refers to the rate at which the coupling strength returns to its mean level. Volatility ($\sigma_{ij}$) refers to the amplitude of random perturbations in the dynamic striato-cortical coupling; i.e., how much direct-indirect pathway variability is expressed by the striato-thalamic dynamics. In contrast, the functional impact of descending glutamatergic projections from the frontal region to the striatum[46] is modelled using traditional static coupling.

Based on previous genetic[47], animal[35] and clinical work[12,37,48], we hypothesised that those neural parameters can reveal how local changes in microcircuit activity drive altered frontostriatal dynamics in OCD. We quantified the transition rate between net-excitatory and net-inhibitory coupling and resulting changes in functional connectivity between frontal and striatal regions (Fig. 2E, F). When volatility $\sigma_{12}$ is zero, the dynamic coupling converges to its constant mean value, which boosts functional connectivity as long as the drift $\eta_{12}$ is non-zero. This is not surprising as, by construction, zero volatility causes constant striato-cortical coupling and higher functional connectivity relative to a more dynamic coupling between the frontal cortex and the striatum (Fig. 2E). When volatility $\sigma_{12}$ and neural drift $\eta_{12}$ are both zero, the coupling remains fixed at its set point. For non-zero volatility, the functional connectivity is relatively insensitive to changes in $\eta_{12}$

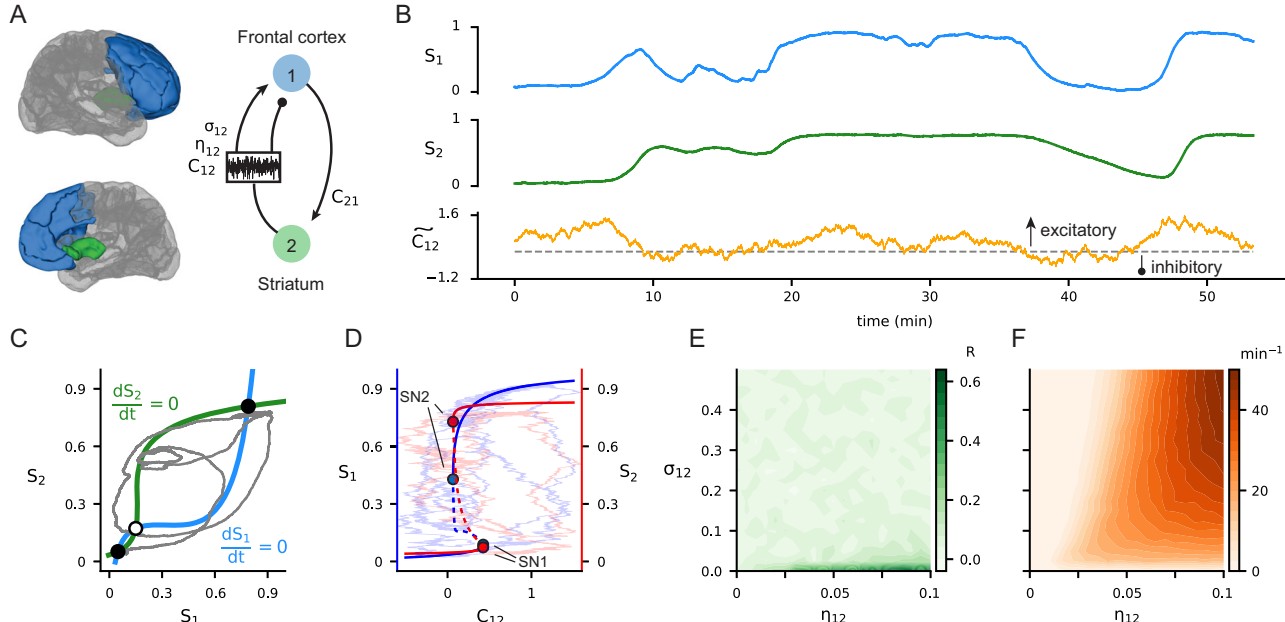

**Fig. 2 | The single circuit model adopted to study frontostriatal neural dynamics at rest. A** Schematic of the model derived from Wong & Wang[32] and Deco et al.[33], incorporating static frontostriatal ($C_{21}$) and stochastic striato-cortical ($C_{12}$, $\eta_{12}$, $\sigma_{12}$) coupling terms (Methods). **B** Time series of the model state variables $S_1$ (frontal cortex), $S_2$ (striatum), and $\widetilde{C_{12}}$ (striato-cortical projection, with sign determining excitation or inhibition). **C** State-space representation of the bistable frontal ($S_1$) and striatal ($S_2$) neural dynamics (i.e., average synaptic gating). Null-clines ($\frac{dS_1}{dt} = 0$, blue; and $\frac{dS_2}{dt} = 0$, green) intersections highlight stable fixed points (black circles, bottom left: low activity state; top right: high activity state) and an unstable fixed point (white circle). The trajectory (grey trace) is the resulting

projection of $S_1$ and $S_2$ timeseries from panel **A**. **D** Stability analysis of the model as a function of striato-cortical coupling ($C_{12}$). The two variables (cortical $S_1$ and striatal $S_2$ activity) exhibit stable (solid) and unstable (dashed) equilibria separated by saddle-node bifurcations (SN1 and SN2 circles) that demarcate the ends of the bistable region. Background trajectories correspond to the projections of time-series shown in panel A in $S_1 - C_{12}$ (blue) and $S_2 - C_{12}$ (red) spaces. **E** Functional connectivity (Pearson's R) between striatum and frontal cortex as a function of drift $\eta_{12}$ and volatility $\sigma_{12}$ parameters. **F** Transition rate (number of zero-crossings in striato-cortical coupling per minute) as a function of drift $\eta_{12}$ and volatility $\sigma_{12}$ parameters.

(Fig. 2E). Conversely, the transition rate between positive and negative coupling increases when both drift and volatility increase (Fig. 2F).

Accordingly, these processes parameterise the net effect of the excitatory-inhibitory balance in each frontostriatal circuit dynamically, influencing its stability. While the simplified single-circuit model captures these essential dynamics within a single isolated system, it cannot address the specifics of key cross-circuit interactions known to be affected in OCD[12].

## Frontostriatal changes in ventral and dorsal circuits activity in OCD

Due to the limitations of our simplified frontostriatal model, we developed a frontostriatal model capturing within- and cross-circuit interactions by coupling two single-circuit models (Fig. 3A). This model links the ventromedial and dorsolateral frontostriatal circuits through lateral cortical and striatal couplings (Fig. 3A). We used the bistable parameter regime from the simplified single-circuit model described above to inform the prior parameter distributions of this cross-circuit model. We then adopted a sequential optimisation algorithm[49] to fit activity and coupling parameters in the two inter-acting circuits to empirical data ("Methods"). Table 2 in the Methods section presents the neurophysiological interpretation of these parameters. Simulated frontostriatal functional connectivity patterns of OCD and healthy controls using parameters drawn from their respective posterior distributions closely match the empirical data, illustrating the successful modelling of both groups (Supplementary Fig. 3).

The posterior distributions of parameters confirmed a general change of frontostriatal parameter values in OCD compared to controls (Fig. 3B). This change mapped onto an increased bidirectional coupling between the nucleus accumbens and the orbitofrontal cortex

in OCD (median $C_{OA}$ 0.36 (OCD) vs. −0.03 (controls), Cohen's $d = 5.2$, two-sided Mann-Whitney U test $p_{FWE} < 0.001$; median $C_{AO}$ 0.34 vs. 0.11, $d = 2.26$, $p_{FWE} < 0.001$, $n = 1000$). Conversely, we detected an opposite coupling pattern in the dorsal circuit, with OCD having a zero-mean coupling from the putamen to the lateral prefrontal cortex (median $C_{LP}$ 0.05 vs. 0.21, $d = 1.12$, $p_{FWE} < 0.001$, $n = 1000$) and decreased posi-tive coupling from the lateral prefrontal cortex to the putamen relative to controls (median $C_{PL}$ 0.10 vs. 0.34, $d = 2.42$, $p_{FWE} < 0.001$, $n = 1000$).

Further analyses revealed a decrease in the lateral prefrontal to orbitofrontal cortices coupling (median $C_{OL}$ 0.22 vs. 0.30, $d = 0.61$, $p_{FWE} < 0.001$, $n = 1000$) and the absence of the negative coupling from the putamen to the nucleus accumbens (median $C_{AP}$ 0.02 vs. −0.26, $d = 0.53$, $p_{FWE} < 0.001$, $n = 1000$) in OCD relative to controls. Impor-tantly, OCD showed significantly decreased drift (median $\eta_{OA}$ 0.04 vs. 0.05, $d = 0.39$, $p_{FWE} < 0.001$; median $\eta_{LP}$ 0.04 vs. 0.05, $d = 0.18$, $p_{FWE} < 0.001$, $n = 1000$) and increased volatility (median $\sigma_{OA}$ 0.27 vs. 0.24, $d = 0.29$, $p_{FWE} < 0.001$; median $\sigma_{LP}$ 0.26 vs. 0.24, $d = 0.22$, $p_{FWE} < 0.001$, $n = 1000$) in both striato-frontal circuits, suggesting more rigid but less stable striato-cortical coupling dynamics.

To test for the role of the thalamus and across-circuit frontos-triatal interactions, we generated a revised model that explicitly incorporates the thalamus and couplings between the ventral and dorsal frontostriatal circuits. We also removed the assumption that glutamatergic projections always have excitatory postsynaptic effects. This more general connectivity was motivated by the fact that frontal glutamatergic neurons can modulate inhibitory striatal interneurons, resulting in net inhibitory effects locally[50]. Supple-mentary Fig. 4A provides a diagram of this supplementary model with the six regions, the updated priors, and the normative coupling parameters' posteriors for controls and OCD after running the Bayesian optimisation. Results adopting this more complex model

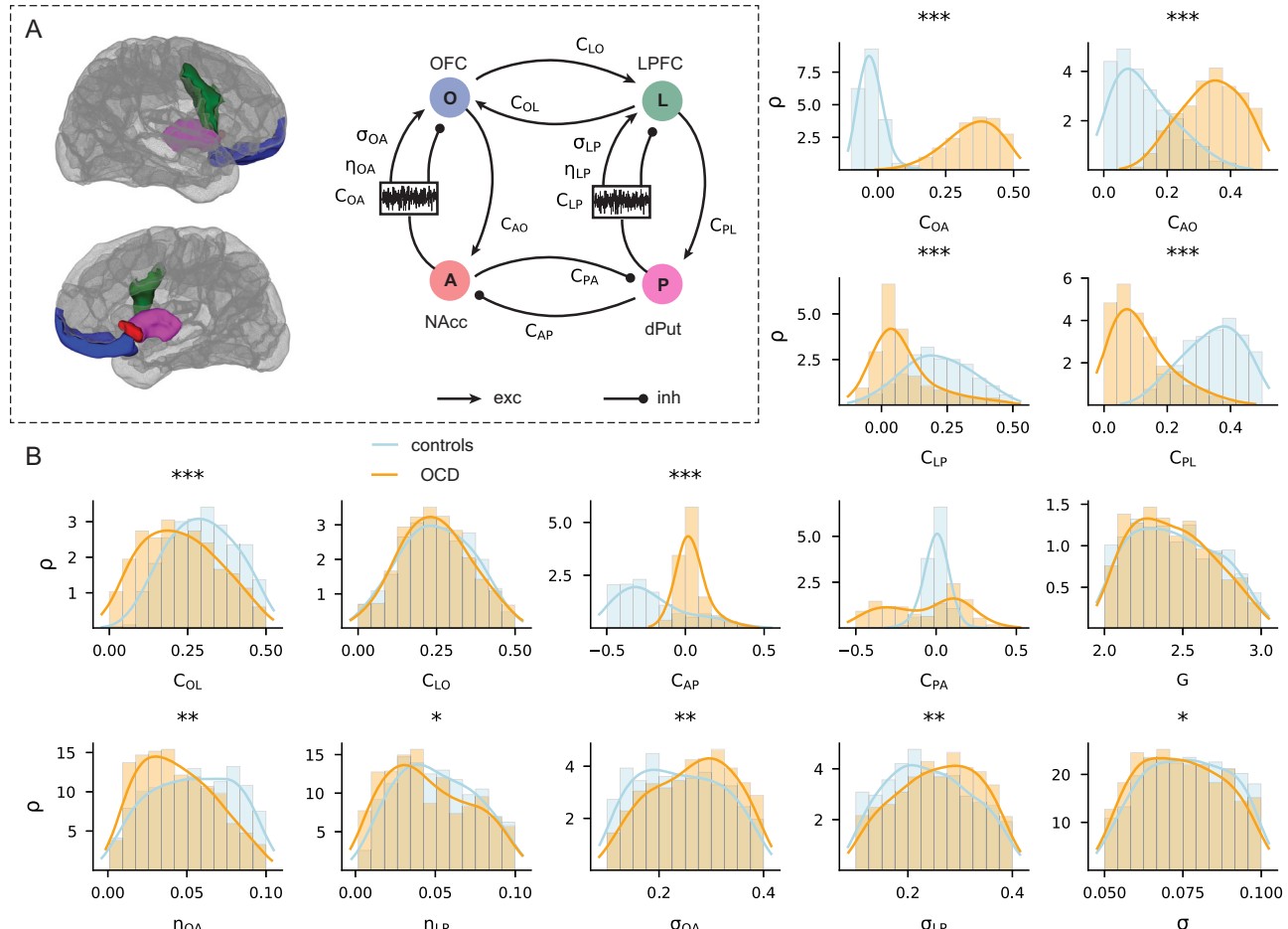

**Fig. 3 | Altered activity of coupled ventral and dorsal frontostriatal circuits in OCD. A** Schematic of the coupled model space comprising the ventral (nucleus accumbens [NAcc]-orbitofrontal cortex [OFC]) and dorsal (putamen [dPut]-lateral prefrontal cortex [LPFC]) frontostriatal circuits and their associated neural parameters (details in Table 2). Brain regions were extracted from the Schaefer et al. atlas (100 regions) for illustrative purposes. Note that $C_{XY}$, $\eta_{XY}$, and $\sigma_{XY}$ are parameters for the connection from region $Y$ to region $X$ (right panel). Global (region-unspecific) parameters do not have subscripts. C terms are coupling strengths, $\eta$ and $\sigma$ terms are drift and volatility properties of noise. **B** Posterior probability

distributions (histograms) and estimated density functions ($\rho$, solid lines, $y$-axis, values are normalised such that $\rho = \frac{a}{\sum a \times \Delta\theta}$ where $a$ is the number of values in a bin and $\Delta\theta$ is the bin width) of each model parameter ($x$-axis, parameter values) after fitting the model to controls (blue) and OCD (orange) functional connectivity from fMRI resting-state data. Mann-Whitney U-test (two-sided): * $p_{FWE} < 0.05$. Significant effects with medium and strong effect sizes are denoted by ** (Cohen's $d > 0.5$) and *** ($d > 0.8$). The detailed statistical table is provided in Supplementary Table 1.

show an OCD-specific disinhibition of the thalamus in the ventromedial circuit. Because removal of thalamic inhibition results in a net increase in cortical excitation[48,51], this finding is consistent with the increased striato-cortical coupling observed in Fig. 3 ($C_{OA}$). The lack of change in thalamo-cortical model parameters between OCD and controls also suggests that OCD-related disruptions are cortico-striatal and striato-thalamic but unlikely thalamo-cortical. In this supplementary model, cortico-striatal projections show opposite signs between the ventromedial and the dorsolateral circuits across groups. This result confirms a circuit imbalance in OCD compared to controls, further supporting that both circuits are affected anti-symmetrically. Finally, this supplementary analysis revealed that cortico-striatal and striato-thalamic ventromedial inhibition over the dorsolateral regions is exacerbated in OCD compared to controls, suggesting an overly active cross-pathway inhibition at play in OCD. While incorporating the thalamus in our model provided confirmatory evidence and additional insights into frontostriatal dysregulations in OCD, the model did not reduce the residual error of the fit to empirical data (Supplementary Fig. 4B). This finding indicates that the added biological granularity did not improve the model evidence[52], even before imposing a penalty for the increased model

complexity[53]. Thus, the remaining of the study focuses on the simplified striato-thalamo-cortical projections introduced in Fig. 3A.

## Defining key neural parameters able to restore healthy functional connectivity

We isolated the minimal changes in model parameters required to bring OCD frontostriatal functional connectivity patterns closer to those observed in healthy controls. Adopting a combinatorial approach, we first ran 1000 simulations using parameter distributions obtained from healthy controls and another 1000 simulations using parameter distributions from OCD subjects. We used these simulations to create our baseline (intervention-free) digital groups. Each group contained 20 digital cohorts of 50 simulations (virtual subjects). Next, sets of up to six parameters (out of the 11 parameters showing significant group differences, Fig. 3B) were systematically permuted between control and OCD posterior distributions to run an additional 1.5 million simulations (1000 different simulations for each of the $\sum_{n=1}^{6} \frac{11!}{n!(11-n)!} = 1485$ combinations of parameter permutations). Permutations of the various parameters can be interpreted as targeted interventions aiming to restore specific attributes of frontostriatal activity disrupted in OCD. This approach allowed us to create synthetic

**Table 2 | Bounds of uniform prior distributions for each parameter with underlying assumptions and interpretations. PFC: prefrontal cortex; OFC: orbitofrontal cortex; NAcc: nucleus accumbens**

| Parameters | Biological interpretation | Lower – upper bounds | Assumptions |
|---|---|---|---|
| $C_{OL}$ | Lateral PFC to OFC coupling strength | 0–0.5 | Cortico-cortical long-range projections are glutamatergic/excitatory[136] |
| $C_{LO}$ | OFC to lateral PFC coupling strength | | |
| $C_{OA}$ | NAcc to OFC coupling strength | − 0.5 – 0.5 | Striato-cortical mean-reversal process can favour either direct (positive) or indirect (negative) pathway[81] |
| $C_{LP}$ | Putamen to lateral PFC coupling strength | | |
| $C_{AO}$ | OFC to NAcc coupling strength | 0–0.5 | Cortico-striatal couplings are excitatory[46,137] |
| $C_{PL}$ | Lateral PFC to Putamen coupling strength | | |
| $C_{AP}$ | Putamen to NAcc coupling strength | − 0.5 – 0.5 | Striato-striatal couplings can be inhibitory or disinhibitory[81,83] |
| $C_{PA}$ | NAcc to Putamen coupling strength | | |
| $\eta_{OA}$ | NAcc to OFC ($\eta_{OA}$) and Putamen to lateral PFC ($\eta_{LP}$) drift; i.e., rate at which striato-cortical coupling strength returns to its baseline value | 0–0.1 s$^{-1}$ | The drift of the striato-cortical mean-reversal process occurs at a timescale at least 10x slower than neural dynamics (i.e., phasic dopamine release acts at a timescale of hundreds of milliseconds)[117] |
| $\eta_{LP}$ | | | |
| σ | Amplitude of neural fluctuations from processes not modelled explicitly | 0.05–0.1 s$^{-1}$ | The noise standard deviation is between 0.05 and 0.1 s$^{-1}$ |
| $\sigma_{OA}$ | NAcc to OFC ($\sigma_{OA}$) and Putamen to lateral PFC ($\sigma_{LP}$) volatility; i.e., the amplitude of random perturbations in coupling strength around its baseline value | 0.1–0.4 s$^{-1}$ | The volatility of striato-cortical mean-reversal process occurs at timescales 2.5 to 10x slower than neural dynamics (i.e., tonic dopamine release acts at timescales of a few milliseconds)[118] |
| $\sigma_{LP}$ | | | |

cohorts undergoing various virtual interventions (Fig. 4A and Supplementary Fig. 5).

We assessed the outcome of a virtual intervention by evaluating the distance between its cohorts' frontostriatal connectivity and the digital healthy controls' connectivity ("Methods"). As the assumption of normality for those distributions is not always guaranteed, we adopted a nonparametric one-sided Mann-Whitney U test to quantify the shift of functional connectivity towards the baseline healthy control cohorts (unpermuted parameters). When scaled by the number of samples, this U statistic becomes the probability that each virtual intervention improves the functional connectivity of simulated OCD cohorts (AUC, Eq.(5), Methods). Virtual interventions were sorted according to the number of permuted parameters (i.e., the number of targets $n_t$ in each intervention) used to restore frontostriatal neural dynamics. Figure 4B shows the top five virtual interventions for each number of targets $n_t$, alongside the comparison within baseline (intervention-free) cohorts.

For single-target interventions ($n_t = 1$), permuting dorsolateral cortico-striatal coupling $C_{PL}$ from OCD to healthy controls resulted in the largest restoration of the whole frontostriatal system's dynamics (AUC = 0.634, $p_{FWE}$ < 0.001; Fig. 4B). That is, if an intervention can only target one parameter, it should target the coupling from the lateral prefrontal cortex to the putamen ($C_{PL}$). Likewise, the modulation of the ventromedial striato-cortical drift ($\eta_{OA}$) also delivered a positive outcome (AUC = 0.627, $p_{FWE}$ < 0.001). Increasing the number of intervention targets ($n_t \geq 2$) improved the efficacy of the interventions logarithmically (mean AUC $\langle AUC \rangle$ > 0.65, Fig. 4B, inset), suggesting the existence of a trade-off line between intervention outcome and complexity or empirical feasibility. Notably, the combinations of parameter substitutions achieving the best intervention outcomes differ according to the number of targets $n_t$, indicating a non-trivial interplay between target points and their resulting outcomes. Thus, instead of considering the optimal neural parameters in absolute terms, the optimal target parameters are relative to the specificity of the planned intervention (e.g., DBS on the NAcc may only be able to change the coupling between this region and the frontal cortex, while pharmacological interventions can alter a broader set of parameters).

We used the dot product to quantify how changes in each model parameter contribute to restoring healthy frontostriatal connectivity ("Methods") across virtual interventions. Briefly, this approach identified which neural targets contribute the most to the overall restoration

of OCD frontostriatal dynamics across all significant interventions at the group level (Fig. 4C). For each of the best interventions per $n_t$ number of targets, the correlation between parameter changes and functional connectivity improvements are provided as regression plots in Supplementary Fig. 6. Figure 4C is an aggregate measure of those correlations across all interventions and not only the absolute best ones. Results showed that when only a few neural targets are concurrently adjusted (≤ 4), increasing dorsolateral cortico-striatal coupling ($C_{PL}$) had the highest association with group-level improvements in functional connectivity across virtual interventions. On the other hand, interventions decreasing the bidirectional ($C_{AO}$ and $C_{OA}$) coupling in the ventromedial circuit delivered the best functional outcomes across virtual interventions when using more (> 4) concurrent targets. These findings highlight different target opportunities based on the ability of a planned intervention to selectively change the relevant neural processes. In addition, results suggested that the effects of changes in cortico-cortical coupling ($C_{OL}$), striato-cortical drifts ($\eta_{OA}$, $\eta_{LP}$) and volatility ($\sigma_{OA}$, $\sigma_{LP}$) on functional connectivity are subject-specific. In other words, magnitudes and directions of change in these targets do not generalise to the whole population across interventions. As a representative example, we plotted the direction and magnitude of changes needed to restore two virtual OCD subjects' parameters to the control average values (Supplementary Fig. 7). Results highlighted that when the underlying parameter distributions (OCD versus controls) largely overlap, many subjects will have parameter changes in opposite directions to achieve restoration of healthy functional connectivity (i.e., the direction of change is subject-specific). Conversely, when the distributions are very distinct, the direction of the targeted change is largely consistent across the cohort. Overall, these findings suggest that restoring healthy frontostriatal dynamics in individuals with OCD is achievable by selectively targeting neural processes in the dorsal and ventral frontostriatal circuits.

**Testing model predictions against longitudinal changes of OCD symptoms and reward-driven impulsivity in healthy controls**
Using four-week follow-up clinical and neuroimaging data collected from the same OCD cohort[16] (Table 1), we tested the predictions generated by our virtual intervention experiment. We started by confirming the clinical relevance of our outcome measure in determining the severity of OCD symptoms based on empirical frontostriatal patterns of functional connectivity. Notably, results showed that changes

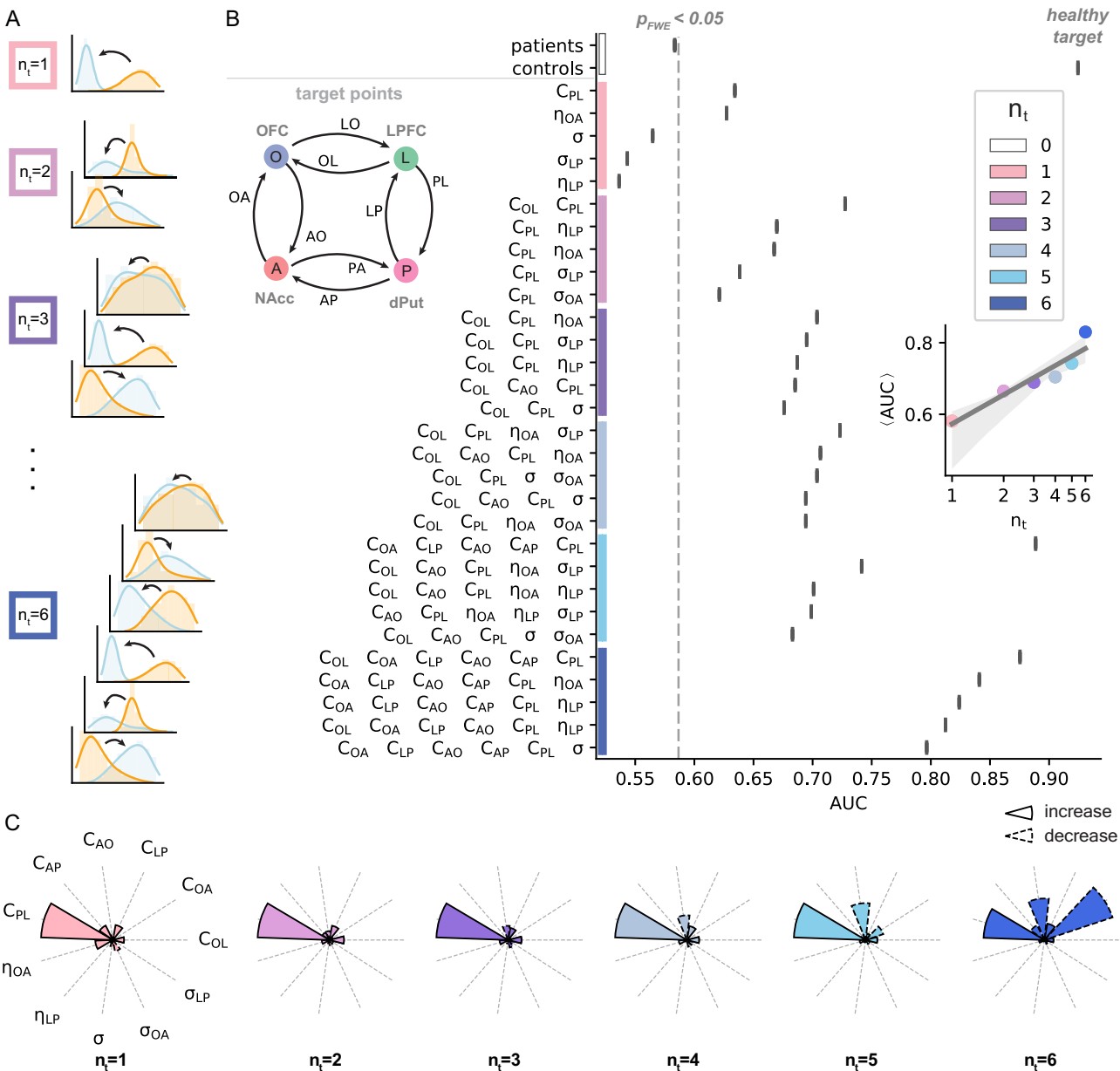

**Fig. 4 | Effect of targeted parameter changes to restore frontostriatal brain dynamics in OCD. A** Schematic of the virtual intervention approach: Sets of $n_t = 1 \ldots 6$ posterior distribution(s) were systematically permuted between parameter sets for OCD and healthy controls, simulating the restoration of healthy parameter ranges in OCD. The statistical outcome of each intervention was estimated using the difference in distance between the simulated healthy functional connectivities (virtual control cohorts) and the simulated OCD connectivities computed *after* the neural parameters were permuted (virtual interventions, Fig. 1C, D, Methods, and Supplementary Fig. 5). **B** Statistical outcome of the targeted virtual interventions. The five most effective interventions to normalise changes in OCD functional connectivity for each number of targets $n_t$ (permuted parameters listed on the *y*-axis and highlighted by colour bands) are shown. The area under the receiver operating characteristic curve (AUC) is derived from the one-sided Mann-Whitney U statistic and denotes the probability that a virtual intervention improves frontostriatal functional connectivity (sample size $n = 400$). $n_t = 0$ corresponds to the baseline OCD and controls simulated functional connectivity. The vertical dashed line indicates the AUC at which $p_{FWE} = 0.05$. Average AUC $\langle AUC \rangle$ per number of intervention ($n_t$) indicates a logarithmic scaling (inset). The grey shaded area corresponds to a CI of 95% estimated by bootstrapping with 1000 resamples. **C** Dot-product between normalised parameter changes and the improvement of functional connectivity across statistically significant virtual interventions ($p_{FWE} < 0.05$), indicating the association between a modulation of parameter (increased or decreased values) and the resulting improvement in functional connectivity. The colour code relating to the number of targets $n_t$ is shared across panels.

in OCD patterns of frontostriatal connectivity towards those observed in healthy controls related to improved OCD symptoms across time ($r = 0.35$, $p = 0.016$, $n = 48$, Fig. 5A). Next, we paired the initial and follow-up functional connectivity patterns of each OCD participant to the closest baseline and post-intervention outputs of our simulations (Fig. 1D). This approach allowed us to create "digital twins" of OCD subjects and probe hidden neural processes associated with the observed changes in symptom severity over time. The pairing was based on the minimal Euclidean distance in functional connectivity space between the empirical values of a given individual and their closest simulations. We quantified the relationship between behavioural improvements in individuals with OCD and their digital twins' changes in parameters via their dot-product. We observed the strongest relationship in the cortico-striatal couplings of ventromedial

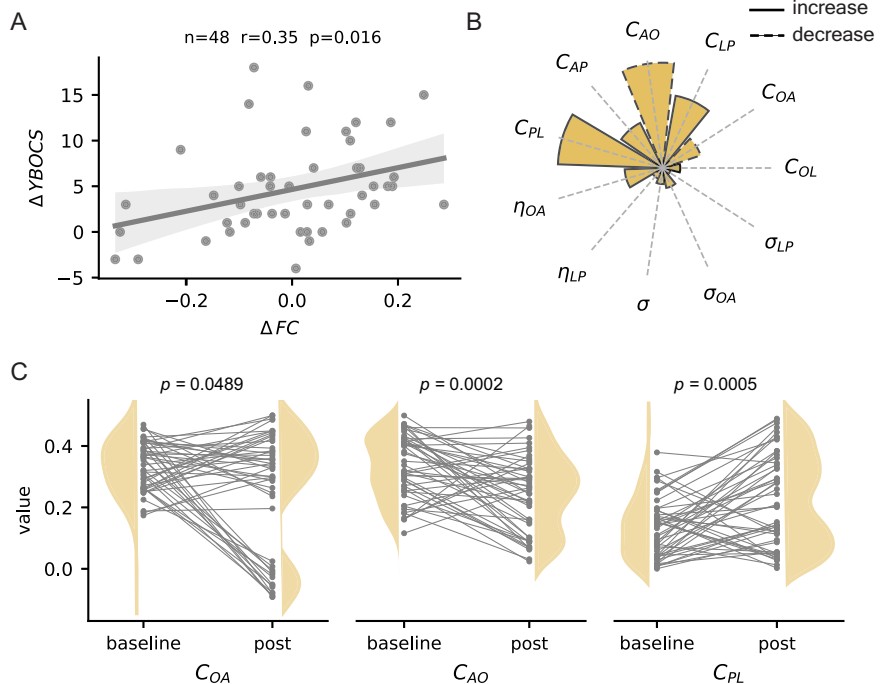

**Fig. 5 | Digital twin analysis on the core neural mechanisms supporting changes in frontostriatal functional connectivity and OCD symptoms over time.**
**A** Changes in resting-state functional connectivity (FC) and OCD symptoms' severity (Y-BOCS: Yale-Brown Obsessive-Compulsive Scale) over time (Δ baseline *minus* follow-up) were linearly correlated (R: Pearson's correlation; grey shaded area corresponds to a CI of 95% estimated by bootstrapping with 1000 resamples). **B** Dot-product between digital twin normalised parameter changes (increase (solid) or decrease (dashed) from initial to follow-up sessions) and the Y-BOCS changes across OCD subjects. This measure quantifies the correspondence of hidden neural processes to functional improvement in empirical data. **C** Distributions of model parameters in digital twins of OCD subjects at baseline and follow-up (post). These parameters showed a significant time effect (two-sided paired Wilcoxon rank test, sample size $n = 48$). Source data are provided as a Source Data file.

($C_{AO}$) and dorsolateral ($C_{PL}$) circuits (Fig. 5B). Raw parameter differences in OCD digital twins between initial (baseline) and follow-up (post) appointments further supported this finding (Fig. 5C, $C_{AO}$ and $C_{PL}$, $p_{FWE} < 0.05$; $C_{AO}$ paired Wilcoxon rank test (two-sided), $n = 48$). Results further suggest a decrease in the ventromedial striato-cortical coupling ($C_{OA}$, $p_{uncorrected} < 0.05$). Collectively, these results highlight the importance of modulating the neural couplings in both ventromedial and dorsolateral circuits to restore healthier resting-state functional connectivity in the OCD frontostriatal system.

It has been suggested that selective serotonin reuptake inhibitors (SSRI) medication can change frontostriatal circuits activity[54]. In a complementary analysis, we show that those results are unlikely driven by medication status (Supplementary Fig. 8), and that the observed functional connectivity changes in the ventromedial and dorsolateral circuits are not directly associated to other clinical measures besides OCD (Supplementary Fig. 9).

Our digital twin approach was further exploited to assess an expected association between the most impactful ventral frontostriatal model parameters defining OCD ($C_{OA}$) and individual differences in reward impulsivity in a healthy population[55,56]. This complementary analysis was conducted by adopting an independent sample of healthy controls from the Human Connectome Project (HCP)[57] ("Methods"). After generating synthetic data from our healthy control model parameters (Fig. 3B, blue distributions), we paired each HCP subject to their closest digital neighbours in frontostriatal functional connectivity space. Next, associations between the estimated ventromedial striato-cortical coupling parameter $C_{OA}$ and reaction time in the delay-discounting tasks were evaluated. This task was chosen because human[58] and preclinical studies[59,60] showed that task execution engages a brain network comprising the nucleus accumbens and the orbitofrontal cortex. Results showed a correlation between

behavioural performance and the ventromedial coupling parameter $C_{OA}$ (Pearson's $R = 0.09$, $p = 0.004$, $n = 1015$), suggesting that our digital twins can inform on frontostriatal neural parameters underlying behaviours beyond OCD.

## Discussion

The main objective of this study was to identify neural mechanisms underpinning spontaneous changes in frontostriatal activity supporting obsessions and compulsions in OCD. Our biophysical model and computer simulations provided a unique opportunity to link OCD-induced changes in neural processes suggested by preclinical[4,17,61] and post-mortem[62] studies with altered macroscopic patterns of brain activity observed in individuals using neuroimaging techniques[10,63]. Together with a unique longitudinal dataset comprising individuals with OCD and healthy controls[12,16], our modelling work suggests that distinct changes in neural variability and cross-regional coupling play a key role in OCD frontostriatal pathophysiology. We further found that the bidirectional reduction of neural coupling between the nucleus accumbens to the orbitofrontal cortex, combined with increased couplings between the lateral prefrontal cortex and the dorsal putamen, supports the improvement of obsessions and compulsions over time. These findings confirm[64] and progress the understanding of the neural mechanisms relating key diagnostic and dimensional features of OCD, offering potential targets for precision brain systems therapy to be probed in clinical trials.

Studies using animal models of OCD have suggested altered patterns of excitatory and inhibitory neural activity in the ventral striatum[17] and frontal brain regions[65]. These variations in local neural activity have been linked with altered cortico-striatal synaptic transmission[4,62], possibly reflecting an activity imbalance between the direct and indirect pathways of the basal ganglia[66]. Specifically, altered

outputs of the ventral striatum due to changes in the dynamics between fast-spiking striatal interneurons and medium spiny projection neurons may disrupt activity in the ventral frontostriatal system and cause the emergence of impulsive behaviour and OCD symptoms[17,59]. This hypothesis aligns with the current results, showing greater neural coupling between the ventral striatum (nucleus accumbens) and the orbitofrontal cortex in OCD. Accordingly, our findings suggest that the coupling between the ventral striatum and orbitofrontal cortex may explain variability in reward-based impulsive behaviour[67,68].

Computational work focusing on obsessions and compulsions is relatively scarce. Early neural network models[69] were based on other psychiatric disorders, such as schizophrenia[70–72]. While these models simulated behavioural measures captured by cognitive tasks with reasonable success, they did not assess the brain basis of behaviour. The next generation of models were informed from contemporary discoveries concerning the basal ganglia microcircuitry and the dopaminergic system[73]. These models focused on neural mechanisms supporting reinforcement learning[74], serial processing[75], and action selection[76–78] and were applied to gain insights on Parkinson's disease or addiction (see a review by Gillies et al.[79]). Advances in modelling approaches allowed us to consider larger cortico-striato-thalamo-cortical circuits[80–83] and permitted a better integration of core findings from animal models of OCD[84]. More recent modelling has implicated a role for serotonergic, dopaminergic and glutamatergic dysregulations in OCD[85]. Moreover, the in-silico study of brain circuits can provide insights to guide the development of new therapeutic interventions and pharmacological neuromodulation[86]. Our current modelling work extends on previous studies by (i) providing mechanistic insight into recent neuroimaging findings on altered frontostriatal functional connectivity in OCD; (ii) using recent advances in parameter estimation to infer distributions of biophysical model parameters, rather than point estimates; (iii) systematically exploring the efficacy of a range of potential interventions to restore healthy neural parameter values, considering interventions both individually and working in combination; (iv) showing that a digital twin framework can be used to probe hidden neural processes associated with changes in functional connectivity; and (v) linking changes in symptom severity to changes in neural parameters. Moreover, the presented framework is made publicly accessible for future development, including model extensions with additional brain network regions, and applications to other clinical conditions.

Capitalising on unique longitudinal clinical and neuroimaging data, we confirmed a linear relation between changes in frontostriatal functional connectivity and changes in OCD symptom severity over time. This relationship appeared when regressing OCD symptom severity against functional connectivity in both ventromedial (NAcc-OFC) and dorsolateral (Putamen-LPFC) circuits. This is in line with previous reports from preclinical[35], neuroimaging[10], and clinical work[6] suggesting a relation between the considered frontostriatal circuits and OCD symptoms. By relating those changes to hidden neural parameters, our findings suggest that targeted therapeutic interventions have the highest chances of succeeding when they concurrently reduce neural coupling within the ventral circuit and enhance top-down coupling within the dorsolateral circuit. This hypothesis is supported by evidence from neurosurgical studies changing the activity of the nucleus accumbens and normalising its interplay with frontal activity, improving symptoms[6,8,87]. However, while deep-brain stimulation targeting the ventral circuit in OCD patients can improve clinical outcomes, 20–25% relapse within two years[88]. The current results suggest that the lack of concurrent interventions targeting the dorsal circuit may hinder a stronger and sustained clinical improvement following DBS treatment, refraining the frontostriatal system from fully reaching and stabilising in a healthy state. Targeted cognitive interventions[89,90] or non-invasive neuromodulation approaches[91] may

thus complement DBS interventions on the ventral circuit by recovering the influence of lateral frontal regions to the putamen's activity. This proposition is in line with studies showing that cognitive-behavioural therapy in individuals with OCD[89,90] normalises functional connectivity in the dorsolateral frontostriatal circuit[92].

In addition to the advances mentioned above, our study provides a computational tool to interrogate the neural underpinnings of OCD and behaviours linked to frontostriatal circuits' activity. Combining neuroimaging data with mathematical models and numerical analysis offers a promising approach to bypassing the intrinsic limitations of human experiments[22]. The present work directly illustrates this point by providing insights into disease mechanisms, including changes in the directed neural coupling between distinct frontostriatal brain regions in OCD. Moreover, we provide a virtual mapping of potential intervention targets and their associated functional outcomes. This knowledge helps to frame the investigative scope of expensive and time-consuming proof-of-concept clinical trials[26].

The interpretation and translational value of the virtual mapping presented here require some caution. Fluctuations of the BOLD signal emerge from complex physiological and metabolic processes that are not yet fully understood[93], including the signalling from excitatory and inhibitory neurons and other non-neuronal cells[94–96]. While changes in the level of synaptic activity do not account for all relevant mechanisms, the modelled non-linear responses of the observed hemodynamics has been widely validated[97] and shown to generate realistic resting-state BOLD timeseries with associated functional connectivity patterns[22,33]. Future modelling work on OCD could integrate other metabolic and synaptic processes contributing to the neurovascular coupling and the BOLD signal[24,98]. Also, our outcomes measure does not encompass the feasibility of the interventions beyond sorting them by the number of targets. It will be important to develop an outcome measure informed by operational difficulty, which may be patient-specific. In addition, while we identified the most efficient intervention targets at the group level, the current framework can easily be adapted to inform the development of personalised interventions[99]. As we illustrated, given an individual OCD patient's frontostriatal functional connectivity, one can use our framework to generate a digital twin to compare this subject's model parameters to the distribution of control parameters. Connections showing the largest deviation from the controls' values could be targeted by clinical interventions to lean towards healthier frontostriatal dynamics. We here note that the reported peak location of the ventromedial hyper-connectivity and dorsolateral hypoconnectivity differ across studies (see Supplementary Figs. 1 and 2) and most likely subjects. However, these functional clusters belong to the same canonical networks thought to support different symptomatic dimensions of OCD[2,3,13,14]. Future studies are required to reliably define prefrontal regions underlying OCD and its symptom severity. Since symptom severity in our OCD cohort only moderately changed over time (Y-BOCS mean = −4.7 points over four weeks)[16] our framework will benefit from additional data with larger changes in symptoms. Nevertheless, using the current training data, the proposed "digital twin" analysis can readily extract hidden neural parameters related to OCD from new clinical data. This can facilitate the extraction of intervention targets to enhance the accuracy of future treatments.

In summary, we extended an established biophysical model[32,33] and exploited unique neuroimaging and clinical data[12,16] to generate a parsimonious account of the neural mechanisms underpinning altered frontostriatal brain activity supporting obsessive and compulsive behaviour characteristic of OCD. By blending computational modelling, neuroimaging, and clinical data, our work bridges the gap between knowledge gathered from preclinical and clinical studies. We used a validated neural mass model explaining resting-state dynamics[33,100] rather than the generic noise-driven exponential decay model utilised in dynamic causal modelling (DCM for fMRI[28,101]). We

added to this model a stochastic process abstracting the dynamics of the direct-indirect pathway interactions in striato-thalamo-cortical circuits relevant to OCD. Our findings support and advance knowledge on the opposing functional changes characterising the activity of the ventral and dorsal frontostriatal circuits in OCD, suggesting new neural targets for research on obsessive thinking and compulsive behaviours.

# Methods

## Participants

Fifty-two individuals diagnosed with OCD for at least 12 months, showing moderate to severe symptoms (total Y-BOCS score greater than 14), were recruited across Australia (Table 1). A board-certified psychiatrist confirmed a primary diagnosis of OCD and assessed comorbid symptoms of anxiety and depression using validated clinical tools. This OCD cohort was compared to an age, gender, and handedness matched group of healthy controls ($n = 45$). Exclusion criteria included a history of psychotic disorders, suicide attempts, manic episodes, seizures, neurological disorders, traumatic head injuries, substance abuse disorders, and contraindications to MRI. All participants were between 18 and 50 years of age and had stable pharmaceutical treatment. The study was approved by the Human Research Ethics Committee of QIMR Berghofer (P2253) and the relevant governance offices (HREC/16/QRBW/265). Written informed consent was obtained from all participants. Participants did not receive monetary compensation for enrolling in the study but were offered reimbursement for associated travel expenses.

These longitudinal data were collected in the context of a clinical trial probing the effect of transcranial magnetic stimulation (TMS) to normalise frontostriatal functional connectivity and improve OCD symptoms (ACTRN12616001687482). Results from this trial were negative, showing that the adopted brain stimulation protocol did not result in differential changes in symptoms, frontal activity, and functional brain connectivity between sham and active interventions. However, participants included in the trial showed an overall reduction in OCD symptoms from baseline to follow-up.

## Data acquisition and pre-processing

Neuroimaging recordings were acquired on a 3T MRI scanner equipped with a 64-channel head coil (Herston Imaging Research Facility, Brisbane, Australia).

Structural (T1) and whole-brain fMRI images were acquired with the following parameters: *T1:* voxel size = 1 mm³, TR = 1900 ms, TE = 2.98 ms, 256 slices, flip angle = 9°; *fMRI:* voxel size = 2 mm³, TR = 810 ms, acceleration factor = 8, TE = 30 ms, flip angle = 53°, field of view = 212 mm, 72 slices. Eyes-open resting-state recordings lasted about 12 min (880 volumes). Anterior-to-posterior and posterior-to-anterior field maps were also collected.

Functional brain images were pre-processed using a combination of fMRIprep (v. 20.2.1)[102] and FMRIB's ICA-based X-noiseifier (ICA-FIX). Image pre-processing included skull stripping, correction for susceptibility distortions, co-registration to the anatomical image, slice timing, ICA-FIX[103], resampling to a standard space (MNI152Nlin2009-cAsym), detrending, global signal regression, temporal filtering (0.01–0.1 Hz), scrubbing (framewise displacement threshold of 0.5 mm) and 8 mm spatial smoothing[12,31].

Time series were extracted from frontostriatal regions of interest (right hemisphere, based on previous observations showing larger functional differences in this hemisphere in OCD compared to controls[10,12]). Striatal regions were defined by previous studies[10,104], while cortical regions correspond to areas that showed significant resting-state FC changes between OCD and healthy subjects[12]. For the complementary analysis of healthy impulsive decision-making using data from the Human Connectome Project (HCP)[57], we used the Schaefer 2018 atlas (100 cortical regions)[105] combined with the Melbourne subcortical atlas (scale 1)[106]. Frontostriatal ROIs were

extracted from the right hemisphere labels corresponding to nucleus accumbens (Nac), putamen (Put), orbitofrontal cortex (OFC_1), and lateral prefrontal cortex (PFCl_1).

## Model of bidirectional frontostriatal circuit

**Model description.** We used a mathematical model of neural activity originally developed to explain perceptual decision-making[32] and further simplified to model resting-state fMRI brain dynamics[33]. The choice of this model was motivated by its extensive use in the modelling of brain dynamics underlying resting-state fMRI[33,107,108], its derivation from the interplay between local neural excitation-inhibition[109], and its original application to study cortico-basal ganglia circuit mechanisms[110]. All of those model ingredients are relevant to the pathophysiology of OCD and the imaging modalities of our empirical dataset. The equations of the reduced Wong-Wang model[33] are:

$$\dot{S}_i = -\frac{S_i}{\tau_S} + (1 - S_i)\gamma H(x_i) + \nu_i, \tag{1}$$

$$H(x_i) = \frac{ax_i - b}{1 - \exp(-d(ax_i - b))}, \tag{2}$$

$$x_i = wJ_N S_i + GJ_N \sum_j C_{ij} S_j + I_0, \tag{3}$$

where $S_i$ denotes the average synaptic gating of population $i$ with a relaxation timescale of $\tau_S = 100$ ms;

$H(\cdot)$ is a nonlinear population's average synaptic gating to firing rate transfer function, and $\nu_i(t)$ is a Gaussian white noise process with variance $\langle \nu_i(t)^2 \rangle = \sigma^2 = 0.01 \, \text{s}^{-2}$. Couplings within and between populations are scaled in time by a constant $1/\gamma \approx 1560$ ms, and in amplitude by constants $w = 0.9$ and $G = 2.5$, respectively, with global factor $J_N = 0.2609$ nA. Cross-population coupling is defined by the connectivity matrix $C$ wherein entry $C_{ij}$ denotes the coupling strength from population $j$ to population $i$. Default parameters are summarised in Supplementary Table 2. Parameters $a, b, d, \gamma, w$ and $I_0$ were based on their prior literature estimates and lack of hypothesis for them varying in OCD. Numerical simulations were performed in Python (version 3.9.16) using an Euler-Maruyama integration method with time step $dt = 0.01$ ms.

To calibrate the model in a parameter space relevant to the pathophysiology of OCD, we conducted a numerical stability analysis of the model using two bidirectionally coupled neural populations (simplified model, details in Supplementary Materials Section I). As obsessions and compulsions are believed to emerge from imbalanced activity in frontostriatal circuits[3,14], understanding the elements that trigger transitions between low and high neural activity in the modelled brain areas is a necessary stepping stone to understand a system of multiple coupled sub-circuits.

The two-population system with frontal and striatal regions (Fig. 2) exhibits a bistable regime for mutually excitatory couplings, with the coexistence of a baseline activity state and a high-activity state. Each state is attracting and stable, indicating that the observed state of the system depends only on initial conditions under fixed coupling. A bifurcation analysis using one of the coupling strengths as a free parameter reveals a pair of saddle-node bifurcations in each variable $S_1$ and $S_2$, which delimit the bistability regime (see Supplementary Section I). Without noise, spontaneous transitions between each stable branch could only occur through these saddle-node bifurcation points.

**Modelling the striato-cortical projections.** Cortico-striatal projections originate mainly from pyramidal neurons in Layer V and have direct glutamatergic synapses with striatal neurons (inhibitory medium spiny neurons, the largest population in the striatum, and both

fast-spiking and low-threshold interneurons)[46,111-113]. The static coupling $C$ described in the section above is an adequate model parameter for these cortico-striatal projections.

In humans, striatal activity is conveyed to thalamic nuclei via the *globus pallidus* and the *substantia nigra compacta*, in what has been conceptualised as the direct and indirect pathways of the basal ganglia[114,115] (Supplementary Fig. 10). The net effect of striatal activations on the thalamus is context-dependent[39] and can be excitatory (disinhibitory) or inhibitory. The thalamus then relays the information to the corresponding cortical regions[116] through monosynaptic glutamatergic projections that are assumed excitatory. The activity of striato-thalamo-cortical circuits is modulated by dopamine on timescales ranging from a few milliseconds to several hundreds of milliseconds[117,118]. In line with this knowledge, our model treats the net effects of striato-thalamo-cortical connections through a noise-affected striato-cortical influence that can be excitatory or inhibitory depending on the moment-by-moment relative contributions of the different pathways. In practice, we implemented these time-varying couplings $\widetilde{C}_{ij}$ as mean-reverting stochastic processes (Ornstein-Uhlenbeck processes[119]):

$$\dot{\widetilde{C}}_{ij} = -\eta_{ij}\left(\widetilde{C}_{ij} - C_{ij}\right) + \upsilon_{ij}, \qquad (4)$$

where $C_{ij}$ is the mean coupling strength from population $j$ to population $i$, $\eta_{ij}$ is the relaxation rate toward this mean (drift), and $\upsilon_{ij}(t)$ is a Gaussian white noise process with variance $\left\langle \upsilon_{ij}(t)^2 \right\rangle = \sigma_{ij}^2$ (volatility). Equation(3) thus becomes $x_i = wJ_NS_i + GJ_N\sum_j\widetilde{C}_{ij}S_j + I_0$.

**System simplifications: from local dynamics to global connectivity in the striato-cortical projections.** Using insights from our two-population model, we built a four-population model (Fig. 3A) implicating activity in two circuits—one ventral and one dorsal—known to be disrupted in OCD[10,12-14]. Supplementary Fig. 10 summarises the biological simplifications made in our model. Since functional coupling can occur without direct anatomical connectivity, we allowed cross-coupling between brain circuit regions to also capture indirect functional paths not explicitly modelled. For example, because there is topographic mapping along the ventromedial-dorsolateral axis from the frontal cortex to the striatum[120], and from the striatum back to the cortex via the globus pallidus[121] and the thalamus nuclei[122-124], we abstracted a complex cortico-striato-pallido-thalamo-cortical circuit into a simpler striato-cortical circuit. Our approach allowed to capture these complex cortico-striato-pallido-thalamo-cortical circuits via a simpler striato-cortical circuit containing only regions showing a significant group difference in functional connectivity[12]. Nevertheless, to assess contributions of the thalamus in a confirmatory analysis, we created a six-populations model by including a thalamic nucleus between the striatal regions and the cortex in each pathway (Supplementary Fig. 4). In this network, striato-thalamic projections are modelled with the stochastic couplings introduced above, while thalamo-cortical and cortico-striatal couplings remain non-dynamic. We also allowed cortico-striatal and thalamo-cortical couplings to be inhibitory, to include scenarios where glutamatergic neurons mainly project post-synaptically to inhibitory neurons rather than pyramidal cells[125] (see section below about priors of the optimisation).

**Hemodynamic model of neural activity.** To compare our model neural activity $S$ to neuroimaging data, we applied a hemodynamic filter converting neural firing rates to blood-oxygenated level-dependent responses[97]. This conversion, known as the Balloon-Windkessel model, is achieved via a nonlinear dynamical system describing the normalised deoxyhemoglobin content, normalised blood inflow,

resting oxygen extraction fraction, and normalised blood volume. We used an existing model implementation[126], with all state equations and biophysical parameters taken from the original publication[97].

## Functional connectivity and assessment of the effects of the virtual interventions

Functional connectivity (FC) between the four regions of interest was calculated using the Pearson correlation between average fMRI time-series extracted from each region. This resulted in six FC values per subject (or simulation). A cohort's FC ($FC_p^{coh}$) is the set of FC of all subjects in the cohort across each edge $p$ (six edges). The distance between two cohorts A and B in FC space was computed as the sum of the Wasserstein distances[127] across all edges (i.e., $d(A,B) = \sum_{p=1}^{6}W\left(A_p, B_p\right)$). For the synthetic dataset, drawing parameters from the posterior distributions defined via the above optimisation, we ran 1000 simulations (each simulation corresponding to a virtual subject) for each group (OCD and controls). Next, we divided the virtual subjects from each group into 20 cohorts of 50 subjects. We decided the size of these cohorts based on the size of our empirical dataset (45 healthy controls and 52 subjects with OCD, see Table 1). We refer to these cohorts as reference virtual cohorts. The dissimilarity between these reference cohorts was calculated using a distance metric in functional connectivity space ($d(A,B)$, Supplementary Fig. 5).

In the restoration analysis, for each virtual intervention, we generated an additional set of 1000 simulations (i.e., generating a new set of virtual subjects). As performed for the reference cohorts, we divided the resulting virtual subjects into 20 cohorts of 50 subjects. For these simulations, the specific parameters targeted by the intervention are drawn from the controls' posterior distributions, while the other parameters remained drawn from the OCD posterior distributions. This framework allowed us to explicitly test for the effect of restoring OCD-related parameters to control values. We quantified the effect of an intervention (i.e., functional improvement) using a distance metric in functional connectivity space. Specifically, we calculated the distance between the reference control cohorts and the virtual intervention cohorts ($d(A,B')$, Supplementary Fig. 5), resulting in a sample of $20 \times 20 = 400$ distances. This vector $\boldsymbol{d}(post, baseline_{hc})$ of size 400, reflects the distribution of FC distances between the virtual intervention cohorts (of a specific intervention) and the simulated control cohorts. Likewise, $\boldsymbol{d}(baseline_{OCD}, baseline_{hc})$ is the distributions of distances between the simulated OCD before any intervention and the simulated controls.

We assessed the outcome of a virtual intervention with a non-parametric approach. We used the one-sided Mann-Whitney U test to quantify the improvement in the distance $\boldsymbol{d}(post, baseline_{hc})$ relative to the starting point $\boldsymbol{d}(baseline_{OCD}, baseline_{hc})$, thereby assessing how virtual interventions change the control-to-OCD distance in connectivity space relative to the initial group distance. This is motivated by the focus on finding those interventions that improve FC (decrease of distance to healthy controls FC) rather than seeking all changes, including deleterious ones. We quantify the effect size for each intervention using the area under the receiving operating characteristic curve (AUC), which scales the U-statistic by the number of samples in each group, resulting in the probability of $\boldsymbol{d}(post, baseline_{hc})$ being smaller than $\boldsymbol{d}(baseline_{OCD}, baseline_{hc})$[128], i.e., that the virtual intervention results in an improvement of functional connectivity.

Here,

$$AUC = \frac{U}{n_1 n_2}, \qquad (5)$$

where $n_1 = n_2 = 400$ samples. AUC < 0.5 indicates a negative outcome, i.e., the intervention increases the distance between OCD and controls, while AUC > 0.5 indicates a positive outcome, i.e., the distance between OCD and controls decreases.

To account for the possibility that finite sample sizes may hinder group comparisons, we calculated a null statistic assessing how far away healthy controls are from OCD individuals at baseline versus the variability inherent in the healthy control population, using the distributions of distances $d(baseline_{OCD}, baseline_{hc})$ and $d(baseline_{hc'}, baseline_{hc})$ where $hc$ and $hc'$ refer to different simulated cohorts of healthy controls. Likewise, we derived a null statistic taking into account variability in the OCD population from the distribution of distances $d(baseline_{OCD}, baseline_{hc})$ and $d(baseline_{OCD'}, baseline_{hc})$ where $OCD$ and $OCD'$ refers to different simulated cohorts of OCD subjects.

This virtual intervention approach mitigates the overfitting problem inherent to small datasets. By creating 20 virtual cohorts of 50 virtual subjects, we capture finite sample size effects inherent in our data, rather than generating arbitrarily large ensembles of simulations to obtain arbitrarily precise parameter estimates. Moreover, we focus on distributions of parameters rather than point estimates. To further avoid oversimplification, we also systematically investigated a wide range of scenarios from single parameter changes to combinations of parameters describing more complex interventions.

To relate the contribution of changes in each parameter θ to the virtual intervention outcome, we quantified the relationships between changes in the parameters' values and distance to healthy functional connectivity. First, virtual intervention parameter values were Z-score normalised with respect to their baseline distributions (Fig. 3B, orange distributions). The difference between pre and post-virtual intervention normalised parameter values was then retained for each of the 400 combinations of pre and post-pairings. This resulted in a vector $\theta^z_{pre-post}$, for each virtual intervention. Likewise, the difference between pre and post-intervention functional connectivity forms a vector $d_{pre-post} = d(baseline_{OCD}, baseline_{hc}) - d(post, baseline_{hc})$. We then used the dot-product between $\theta^z_{pre-post}$ and $d_{pre-post}$ across *all* virtual interventions, showing a statistically significant difference (AUC > 0.533, $p_{FWE}$ < 0.05) to quantify the relationship between changes in the parameters' values and improvement in functional connectivity. To give the reader an intuition of this measure, we provide the linear regression plots between those two variables $\theta^z_{pre-post}$ and $d_{pre-post}$ *for the best interventions per number of targets* in Supplementary Fig. 6. The final parameter contribution using the dot-product is an aggregate measure of those regressions across *all* virtual interventions involving $n_t$ number of targets, rather than for only the *best intervention per number of targets*.

### Optimisation using Approximate Bayesian Computation

We used a Bayesian optimisation framework[129] to fit simulated neural dynamics to real data (Supplementary Materials, Section II). The adopted sequential Monte-Carlo algorithm (SMC[49,130]) is an advanced iteration of the variational Bayes approach used in Dynamic Causal Modelling (DCM[28,131]). Specifically, this approach alleviates the Laplace assumption, which treats posterior distributions as necessarily Gaussian[132,133]. To summarise SMC, model parameters θ were first sampled from a prior uniform distribution (Table 2), and the summary statistics (mean and variance) of functional connectivity from simulated time series were compared to the summary statistics observed in real data, using the root mean squared error fitness function. Sequentially, parameters from the best-fitting simulations were used to create new sets of parameters $\hat{\theta}$ using a perturbation kernel inspired by evolutionary algorithms until a target tolerance of error was reached (target $\epsilon$ = 0.01). This method has been shown to outperform other optimisation algorithms for problems when the likelihood function is unknown[49,134]. The fitting of the model against resting-state functional connectivity patterns from healthy controls ($N$ = 45) and subjects with OCD ($N$ = 53) provided posterior distributions of model parameters best matching each group (OCD and controls). These posterior distributions were computed using a kernel density

estimated from the $N$ = 1000 best simulations at the final stage of the optimisation (Supplementary Materials, Section II).

Note that while we assume cortico-cortical and cortico-striatal projections are glutamatergic and therefore excitatory in nature, the net effect of such excitatory projections also depends on the post-synaptic target. Accordingly, in regions with a large proportion of GABAergic targets (e.g., medium spiny neurons in the striatum[135]), the net effect of excitatory projections can become largely inhibitory.

### Digital twin analysis

We paired simulations of functional connectivity (FC) to empirical FC estimated from baseline and follow-up (four weeks post baseline) data collected in individuals with clinical OCD[16]. The pairing was performed using the minimal Euclidean distance in frontostriatal FC (Fig. 1D) between subjects with OCD and their digital neighbours. The empirical baseline (pre) FC was paired to intervention-free simulations ($n$ = 1,000,000), while the follow-up (post) empirical connectivity (which includes both intervention-free and post-intervention subjects) was paired to the combined intervention-free and post-intervention simulated FCs ($n$ = 2,450,000). Our empirical cohort of individuals with OCD showed an overall improvement of symptoms across the four weeks between baseline and follow-up appointments (mean − 4.7 points, 5.0 SD, in Y-BOCS scores)[16]. Four individuals with OCD were excluded from this analysis because of missing data at follow-up, insufficient length of recording after head motion correction, or excessive signal distortion in the frontal regions. The relationship between the improvement of symptoms in the empirical subjects and the normalised parameter changes in their digital twins was assessed using the dot product between the two variables. This measure is related to a Pearson correlation, but does not apply a zero-mean centreing to the variables prior to their piecewise multiplication in order to preserve the interpretation of the sign of the transformations (initial vs follow-up values, i.e., increase or decrease in parameters, and improvement or worsening of symptoms). In the complementary analysis using the independent HCP dataset[55], HCP subjects FCs ($n$ = 1015) were paired to our controls' simulations ($n$ = 1.000.000) using the minimal Euclidean distance in frontostriatal FC between HCP subjects and their digital neighbours.

### Reporting summary

Further information on research design is available in the Nature Portfolio Reporting Summary linked to this article.

## Data availability

Pre-processed functional connectivity and behavioural scores data are publicly available at https://doi.org/10.6084/m9.figshare.26310616 and are attached as Source Data in the Supplementary Materials. De-identified participant data for research purposes are available on request only due to restrictions laid out in the study's participant consent forms and QIMR Berghofer data-protection policies. Data will be made available after a data-sharing agreement with QIMR Berghofer has been signed. The timeframe for reaching an agreement may vary depending on the requesting institution and the contract clauses. Source data are provided in this paper.

## Code availability

The full code is available on GitHub at www.github.com/sebnaze/OCD-Modeling, with its detailed documentation at https://ocd-modeling.readthedocs.io. A Docker container with all the necessary dependencies pre-installed and a demo of the model is included in this website. This allows the computation of digital twins on any given dataset. We also provide a lightweight cloud demo with a graphical user interface to facilitate the initial exploration of the model parameters. The code has been deposited in the CodeOcean platform under https://doi.org/10.24433/CO.3954416.v1.

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

## Acknowledgements

This work was supported by the Australian NHMRC (2001283 and 2027597, L.C. and S.N.). A.Z. and L.J.H. were supported by research fellowships from the NHMRC (1118153 and 1194070, respectively). The authors thank Michael Breakspear for contributing to the clinical data used to test the validity of our model predictions.

## Author contributions

C.R., C.V.H., S.S., B.B. and L.C. collected the data. L.J.H. and S.N. processed the data. S.N. developed the model, wrote the computer code and ran all analyses. P.S.L. and L.C. acquired the funding. J.A.R. and L.C. supervised the study. S.N. and L.C. wrote the manuscript. L.J.H., P.S.L., A.Z., and J.A.R. provided critical review and edits of the manuscript. All authors reviewed the manuscript and provided consent of authorship.

## Competing interests

L.J.H., C.R., B.B., A.Z. and L.C. are involved in a clinical neuromodulation centre (Queensland Neurostimulation Centre, QNC, as trading for Australia Brain Foundation). L.J.H., A.Z. and L.C. are not paid by QNC, while B.B. (psychiatrist) and C.R. (technician) received monetary compensation for their work with QNC. This centre had no role in this study. L.C. served as a co-inventor on a patent application that covers neuroimaging-based personalised TMS. He is also involved in the development of imaging-based personalised TMS for depression with ANT Neuro. The provisional patent and ANT Neuro products are not directly related to this work. The remaining authors declare no competing interests.
