## [Transparent Peer Review file · Nature Communications]

Mechanisms and interventions promoting healthy frontostriatal dynamics in obsessive-compulsive disorder

Corresponding Author: Dr Sebastien Naze

Version 0:

Reviewer comments:

Reviewer #1

(Remarks to the Author)

General Comments:

The manuscript offers a significant contribution by linking computational modeling with clinical data to understand frontostriatal dynamics in OCD. By integrating neuroimaging data with a biophysical model, the study proposes a dynamic dysregulation mechanism of the frontostriatal circuit and validates the model's predictions through intervention experiments. Therefore, I feel that the work is a valuable contribution. However, there are some major concerns which should be addressed.

Major Comments:

- 1.The bistability findings are interesting but only presented in the supplementary materials. Consider including some of these results in the main text.
- 2.There are many coupling parameters mentioned, but their meanings are unclear. Could you provide a table explaining what each parameter, such as COA and CAO, represents and its biological significance?
- 3.Could you clarify how these parameters (η, σ) relate to neural activity stability or volatility? For example, does η represent the speed at which neuronal activity returns to baseline? Does σ indicate random perturbations in neuronal activity? Please explain how these parameters contribute to neural instability and the worsening of OCD symptoms.
- 4.What is the relationship between the Pearson correlation coefficient (R) and the specific parameters in the model? For example, how do these parameters affect COA and CPL, and why?
- 5.The methods section lacks clarity on how local neural activity leads to global connectivity changes. Consider providing a clearer process description, perhaps with a flowchart to improve the readers' understanding of the model.
- 6.The results section discusses various parameters, but it is unclear which parameter has the greatest impact on OCD. Could you identify the most influential parameter and discuss interventions targeting it?
- 7.The discussion section would benefit from a clearer comparison with existing research to highlight the novelty of this work.
- 8.For interesting reader, computational modeling studies on OCD should be extended, such as these references:
[1]Yin L, et al. Unveiling serotonergic dysfunction of obsessive-compulsive disorder on prefrontal network dynamics: a computational perspective. *Cereb Cortex*. 2024 Jun 4;34(6):bhae258. doi: 10.1093/cercor/bhae258.
[2]Yin L, et al.A computational network dynamical modeling for abnormal oscillation and deep brain stimulation control of obsessive-compulsive disorder. *Cogn Neurodyn*. 2023 Oct;17(5):1167-1184. doi: 10.1007/s11571-022-09858-3.

Minor Comments:

- 9.Lines 25-26: The sentence tenses are inconsistent, with multiple tenses used within one sentence.
- 10.The reference formatting is inconsistent throughout the paper, which affects readability.You should double-check.
- 11.The colors are too light, making it difficult to distinguish between the control and OCD groups in Clo and Col in Figure 2. Increase the color contrast and add axis labels to clarify the biological significance of the parameters and neural couplings.
- 12.Figure 3 is overly complex, and it is hard to get the main message. Consider simplifying with two-parameter plots or other formats to show the intervention effects of multiple parameter combinations.
- 13.The table formatting is inconsistent throughout the manuscript.
- 14.In Figure 1B and C of the supplementary materials, it would be helpful to define what S1 and S2 represent in the legend. Also, do the blue and green lines represent $dS1/dt = 0$ and $dS2/dt = 0$? This is unclear from the figure. Additionally, the dashed line in panel C is not prominent, and adding annotations to the figure would improve comprehension.

Conclusion:

Overall, the manuscript presents valuable insights, but addressing these concerns would significantly strengthen its clarity.

(Remarks on code availability)
Code is OK !

Reviewer #2

(Remarks to the Author)

This is an interesting study which combined neuroimaging and behavioural data with computational modelling, focused on investigating the neural mechanisms driving imbalances in frontostriatal circuit activity in OCD patients. The results found that neural variability and cross-regional coupling play a key role in OCD frontostriatal pathophysiology. And they also found bidirectionally decreasing spontaneous neural coupling in the ventromedial circuit while concurrently increasing dorsolateral cortico-striatal coupling delivers the highest functional improvements in OCD. These findings will advance our understanding of the underlying neural mechanisms of OCD and demonstrates significant potential for future clinical translation. However, I have several reservations regarding the manuscript in its current form.

Introduction

1. The authors mentioned that the role of inhibitory balance in microcircuit dynamics and the use of biophysical models cannot fully explained why these existing mechanisms and models cannot be directly applied. It should further clarify "why existing mechanisms cannot be directly used in clinical applications" and "how the biophysical model specifically overcomes these limitations."
2. The basis for selecting the "core parameters" in the biophysical model is not clearly explained. Why did select the core parameters? and how is related to OCD-related neural coupling changes demonstrated? It is recommended to add more details on model validation and the theoretical or experimental basis for selecting these core parameters.

Methods

1. It is unclear about the standards for constructing virtual data. Could the authors provide a more detailed explanation of the process of constructing virtual data based on real data, and how to avoid the problem of over-simplification?

Results

1. The result plots in Figure 2B should be adjusted to match the order of the statistical results in the text, which might enhance clarity and make the presentation more directly.
2. In the results section, it is mentioned that the impact of changes in cortico-cortical coupling (COL), striato-cortical drifts (η_{OA} , η_{LP}) and volatility (σ_{OA} , σ_{LP}) on functional connectivity is subject-specific and may not be generalizable. Would provide additional explanation or detailing how this subject-specific effect was identified?

Discussion

1. The study initially demonstrated that the dorsolateral cortico-striatal coupling CPL consistently plays a key role in reducing the difference in functional connectivity between OCD patients and healthy individuals across various target combinations. However, in the subsequent statistical analysis regarding symptom improvement, it was found that the original parameters of CPL in the digital twin model did not show a significant difference between baseline and follow-up. It would be beneficial to discuss this finding in the discussion section, and add more implications for symptom improvement?

(Remarks on code availability)

Reviewer #3

(Remarks to the Author)

This article explores a computational model of frontostriatal dynamics in obsessive compulsive disorder(OCD) that is used to explain a possible mechanism leading to those dynamics, compare them to those seen in controls, and suggest possible interventions that would normalize those dynamics based on the computational model. Empirical data leading to the model is based on a double blind sham controlled study of orbitofrontal continuous theta burst TMS and its effect on resting state functional connectivity conducted by the same group, that has been previously published in two widely cited journals. Dynamical systems theory and Bayesian inference are used to develop the model and its potential clinical applications. The article is well written and has numerous strengths including a large sample size, confirmatory evidence of the models fit to empirical data using longitudinal data and thoughtful extrapolation of existing computational models to frontostriatal circuitry in OCD. However, there are major concerns about the failure of the model to capture the highly distributed and complex nature of the circuit in question and therefore the authors conclusion that the model can be used to guide circuit based neuromodulatory treatment. The authors have not adequately delineated the limitations of the model or the empirical data on which it is based in the paper.

Some of these limitations are listed below.

- 1) Frontostriatal connectivity findings in OCD differ significantly across studies. For example the 2009 Harrison paper found hyperconnectivity in the L anteromedial OFC as well as the anterior cingulate with the nucleus accumbens(NA) as opposed to the r medial frontal pole and NA in the current study. The Harrison et al studies showed a correlation with YBOCS severity while the current study did not. The 2009 Harrison study found hypofrontality in a frontal region that was significantly anterior to the current study while the finding in the current study is located several millimeters behind the temporal pole. Although 9 patients who were med free are included the authors do not reference other papers showing that frontostriatal hyperconnectivity in OCD may be related to SRI treatment. The lack of reproducibility of the findings across studies may in part be due to grouped data. Though the authors claim that their framework could be easily adapted to inform individual and

personalized interventions it is left unclear how this could be accomplished. The model depends on connectivity data that is consistent and reproducible across studies or individual subjects.

2) The frontal cortex is a massively distributed system that contains several complex interacting networks. It is likely that integration across frontostriatal loops takes place not only in frontal cortex but also at the level of the striatum and thalamus. The computational model fails to explicitly capture the complex functional interplay with and between striatal and frontal regions and their modulation by other large scale networks. The authors state that the model implicitly incorporates thalamic connections but this is likely to be an unfounded assumption.

3) Dynamic causal modeling informs us about effective connectivity but does little to inform us about the actual neural processes underlying the hyperconnectivity. The assumption that regional bold activity is primarily related to an excitatory-inhibitory balance of neural firing rates in a given region has been increasingly challenged. Accumulating evidence points to the importance of presynaptic changes as driving regional metabolic rates and these may not be correlated with firing rates. The assumption that glutamatergic projections always have an excitatory effect on a given region is also not clear due to the density of synaptic interaction with both inhibitory as well as excitatory post synaptic neurons

4) As the authors point out the model could be used to describe many conditions where there is hypofrontality of DLPFC RSFC and hyperfrontality of mPFC with RSFC. The patient population used in this study (Hearne et al 2023) had significant differences between OCD patients and Controls in terms of depressive symptoms (MADRS) and anxiety (HADS). Although this may be unavoidable due to the high rates of comorbidity, it would be helpful to have the concurrent psychiatric diagnoses reiterated in this paper given that this may be relevant for neurocircuitry findings. It would also be relevant to comment on whether concurrent depressive symptoms were accounted for in the models (although this may limit ability to detect change). The discussion could benefit from a discussion of the issues related to this as well as how the findings of frontostriatal hyperconnectivity in OCD compare to other studies of depression and anxiety disorders

Due to the limitations noted above some of the authors conclusions are inadequately supported by the data. For example, it is unclear if the model strikes an optimal balance between biologic plausibility and complexity. The claim that the study "advances knowledge of neural mechanisms underpinning the pathophysiology of OCD and informs the development of new precision treatments" is not supported by the model in this reviewers opinion.

(Remarks on code availability)

Reviewer #4

(Remarks to the Author)

(Remarks on code availability)

Version 1:

Reviewer comments:

Reviewer #1

(Remarks to the Author)

The Authors have addressed the Reviewer's concerns with sufficient details. The current version is suitable for publication in NC.

(Remarks on code availability)

Reviewer #3

(Remarks to the Author)

The authors have made a herculean effort to address this reviewers concerns. They have succeeded in articulating limitations of the model and have added numerous additional analyses that speak to the issues related to the empirical data on which the model is based. They have tempered their conclusions on how the model could inform new circuit based treatments. Overall a most impressive revision that in my opinion justifies publication. A couple of additional questions/concerns that if addressed would further strengthen the manuscript in my opinion.

1) Additional data using a less conservative statistical threshold has addressed inconsistencies between the Harrison et al and Naze et al data sets in terms of the hyperconnectivity of the anterior OFC and ACC. However it remains unclear why the main finding of the Naze et al study in the R frontal pole was not also found in the Harrison study given the less conservative analysis in the Harrison study.

2) While there is substantial evidence supporting RSFC differences in the ventromedial circuit, the data supporting hypoconnectivity in the resting state in the caudal VLPFC/insula found in the Naze study is less clear. A review of the Robbins Shepard and Von Heuvel papers seemed to substantiate a larger swath of mostly DLPFC caudate hypoconnectivity. The caudal VLPFC/opercular.anterior insula area is difficult it terms of low SNR and is an area that

appears to be at the intersection of the salience language and premotor areas. Clarification of which areas have been proven to be hypofunctional in other studies in the lateral prefrontal cortex and what role this are might play in the genesis of OCD symptoms would be useful additions.

(Remarks on code availability)

Reviewer #4

(Remarks to the Author)

(Remarks on code availability)

REVIEWER COMMENTS

We thank the reviewers for their constructive feedback. We carefully considered all comments and addressed them below. The response is formatted as follows:

Reviewers' comments are presented without indentation.

R0: our response is in bold characters with indentation and “changes made in the manuscript are in blue”. Note that citations in this rebuttal use (author, date) format with a dedicated reference section for ease of reading. The manuscript uses indexed items following the Nature Communications reference format.

Reviewer #1 (Remarks to the Author)

General Comments:

The manuscript offers a significant contribution by linking computational modeling with clinical data to understand frontostriatal dynamics in OCD. By integrating neuroimaging data with a biophysical model, the study proposes a dynamic dysregulation mechanism of the frontostriatal circuit and validates the model's predictions through intervention experiments. Therefore, I feel that the work is a valuable contribution.

We thank the reviewer for this positive appraisal of our work.

However, there are some major concerns which should be addressed.

Major Comments:

1. The bistability findings are interesting but only presented in the supplementary materials. Consider including some of these results in the main text.

R1-1: We agree that the bistability findings are important and were not emphasized in the original manuscript. Thus, we have moved Supplementary Fig. 1 to the main text (now Figure 2) and updated the Results section accordingly (p.5, line 152):

“The monostable state of low firing dominates the system for mutually inhibitory brain regions, as well as when one region is excitatory while the other is inhibitory. The monostable state of high firing is only observed when both regions exert strong mutual excitation on each other. These monostable states cannot generate rich neural dynamics necessary to optimally process information. On the other hand, the bistable regime (~~Supplementary~~ Figure 2B-C) is particularly interesting as it permits spontaneous transitions between stable activity states under minimal neural perturbations. State transitions are central to the modelling of transient dynamics supporting many neural processes, including decision-making and executive functions (Rabinovich et al. 2008; Durstewitz and Deco 2008; Wang 2008; Freyer et al. 2012; Roberts, Friston, and Breakspear 2017). Accordingly, we introduced a stochastic coupling process (Methods), allowing the spontaneous shifts between dynamical states to occur. This process

approximates the dynamic contribution of biological pathways promoting or inhibiting neural couplings between striatal regions and the frontal cortex (Supplementary Figure 2D).”

2. There are many coupling parameters mentioned, but their meanings are unclear. Could you provide a table explaining what each parameter, such as COA and CAO, represents and its biological significance?

R1-2: To better highlight the meaning of the various model’s parameters we added a “Biological interpretation” column to Table 2:

Table 2: Bounds of uniform prior distributions for each parameter with underlying assumptions and interpretations. PFC: prefrontal cortex; OFC: orbitofrontal cortex; NAcc: nucleus accumbens.

Parameters	Biological interpretation	Lower – upper bounds	Assumptions
C_{OL}	Lateral PFC to OFC coupling strength	0 – 0.5	Cortico-cortical long-range projections are glutamatergic/excitatory (Jirsa and McIntosh 2007)
C_{LO}	OFC to lateral PFC coupling strength		
C_{OA}	NAcc to OFC coupling strength	-0.5 – 0.5	Striato-cortical mean-reversal process can favour either direct (positive) or indirect (negative) pathway (Tomkins et al. 2014)
C_{LP}	Putamen to lateral PFC coupling strength		
C_{AO}	OFC to NAcc coupling strength	0 – 0.5	Cortico-striatal couplings are excitatory (Reiner et al. 2010; Shepherd 2013)
C_{PL}	Lateral PFC to Putamen coupling strength		
C_{AP}	Putamen to NAcc coupling strength	-0.5 – 0.5	Striato-striatal couplings can be inhibitory or disinhibitory (Spreizer et al. 2017; Tomkins et al. 2014)
C_{PA}	NAcc to Putamen coupling strength		
η_{OA}	NAcc to OFC (η_{OA}) and Putamen to lateral PFC (η_{LP}) drift; i.e., rate at which striato-cortical coupling strength returns to its baseline value	0 – 0.1 s ⁻¹	The drift of the striato-cortical mean-reversal process occurs at a timescale at least 10x slower than neural dynamics (i.e., phasic dopamine release acts at timescale of hundreds of milliseconds) (Berke 2018)
η_{LP}			
σ	Amplitude of neural fluctuations from processes not modelled explicitly	0.05 – 0.1 s ⁻¹	The noise standard deviation is between 0.05 and 0.1 s ⁻¹

σ_{OA}	NAcc to OFC (σ_{OA}) and Putamen to lateral PFC (σ_{LP}) volatility; i.e., the amplitude of random perturbations in coupling strength around its baseline value	$0.1 - 0.4 \text{ s}^{-1}$	The volatility of striato-cortical mean-reversal process occurs at timescales 2.5 to 10x slower than neural dynamics (i.e., tonic dopamine release acts at timescales of a few milliseconds (Liu, Goel, and Kaeser 2021))
σ_{LP}			

3. Could you clarify how these parameters (η, σ) relate to neural activity stability or volatility? For example, does η represent the speed at which neuronal activity returns to baseline? Does σ indicate random perturbations in neuronal activity? Please explain how these parameters contribute to neural instability and the worsening of OCD symptoms.

R1-3: We apologize for the lack of clarity. To address this problem, we revised the text as follows (Results, p.5, line 163):

“In this updated model, striato-cortical projections result from the product of the striatal activity with a dynamic coupling strength that approximates the interplay between direct and indirect pathways of the basal ganglia and its net thalamo-cortical effect. We introduced three new parameters to describe this dynamic coupling from striatal to frontal regions: mean strength, relaxation toward the mean (drift), and variance of endogenous fluctuations (volatility). The drift parameter (η_{ij}) refers to the rate at which the coupling strength returns to its mean level. Volatility (σ_{ij}) refers to the amplitude of random perturbations in the dynamic striato-cortical coupling; i.e., how much direct-indirect pathway variability is expressed by the striato-thalamic dynamics. In contrast, the functional impact of descending glutamatergic projections from the frontal region to the striatum (Shepherd, 2013) is modelled using traditional static coupling.

Based on previous genetic (Mattheisen et al. 2015), animal (Ahmari et al. 2013) and clinical work (e.g., (Naze et al. 2023; Fettes, Schulze, and Downar 2017; Peters, Dunlop, and Downar 2016)), we hypothesized that those neural parameters can reveal how local changes in microcircuit activity drive altered frontostriatal dynamics in OCD.”

4. What is the relationship between the Pearson correlation coefficient (R) and the specific parameters in the model? For example, how do these parameters affect COA and CPL, and why?

R1-4: We performed additional analyses using the simplified two-population model to address the Reviewer’s comment. Specifically, we varied the parameters η_{12} (drift) and σ_{12} (volatility) (Figure 2E-F) and assessed the resulting changes in functional connectivity (Pearson’s R). This additional analysis clarifies how changes in neural parameters supporting OCD frontostriatal deregulations link to modulations in functional connectivity. The following changes have been made in the Results section (p.6, line 193):

“We quantified the transition rate between net-excitatory and net-inhibitory coupling and resulting changes in functional connectivity between frontal and striatal regions (Figure 2E-F). When volatility σ_{12} is zero, the dynamic coupling converges to its constant mean value which boosts functional connectivity as long as the drift η_{12} is non-zero. This is not surprising as, by construction, zero volatility causes constant striato-cortical coupling and higher functional

connectivity relative to a more dynamic coupling between the frontal cortex and the striatum (Figure 2E). When volatility σ_{12} and neural drift η_{12} are both zero, the coupling remains fixed at its set point. For non-zero volatility, the functional connectivity is relatively insensitive to changes in η_{12} (Figure 2E). Conversely, the transition rate between positive and negative coupling increases when both drift and volatility increase (Figure 2F).

Accordingly, these processes parameterise the net effect of the excitatory-inhibitory balance in each frontostriatal circuit dynamically, influencing its stability.”

Supplementary Figure 2: The single circuit model adopted to study frontostriatal neural dynamics at rest. **A.** Schematic of the model derived from Wong & Wang (2002) and Deco et al. (2013), incorporating static frontostriatal (C_{21}) and stochastic striato-cortical (C_{12} , η_{12} , σ_{12}) coupling terms (Methods). **B.** Time series of the model state variables S_1 (frontal cortex), S_2 (striatum), and \widetilde{C}_{12} (striato-cortical projection, with sign determining excitation or inhibition). **C.** State-space representation of the bistable frontal (S_1) and striatal (S_2) neural dynamics (i.e., average synaptic gating). Nullclines ($\frac{dS_2}{dt} = 0$, blue; and $\frac{dS_1}{dt} = 0$, green) intersections highlight stable fixed points (black circles, bottom left: low activity state; top right: high activity state) and an unstable fixed point (white circle). The trajectory (grey trace) is the resulting projection of S_1 and S_2 timeseries from panel A. **D.** Stability analysis of the model as a function of striato-cortical coupling (C_{12}). The two variables (cortical S_1 and striatal S_2 activity) exhibit stable (solid) and unstable (dash-dotted) equilibria separated by saddle-node bifurcations (SN1 and SN2 circles) that demarcate the ends of the bistable region. Background trajectories correspond to the projections of timeseries shown in panel A in $S_1 - C_{12}$ (blue) and $S_2 - C_{12}$ (red) spaces. **E.** Functional connectivity (Pearson’s R) between striatum and frontal cortex as a function of drift η_{12} and volatility σ_{12} parameters. **F.** Transition rate (number of zero-crossings in striato-cortical coupling per minute) as a function of drift η_{12} and volatility σ_{12} parameters.

5. The methods section lacks clarity on how local neural activity leads to global connectivity changes. Consider providing a clearer process description, perhaps with a flowchart to improve the readers’ understanding of the model.

R1-5: Thank you for the feedback on this important point. Accordingly, we added a section describing the links between local activity and connectivity in the macroscopic basal ganglia-frontal system and provide a new diagram highlighting the rationale of our simplified model:

Methods (p.17, line 659):

iii) System simplifications: from local dynamics to global connectivity in the striato-cortical projections.

Using insights from our two-population model, we built a four-population model (Figure 3A) implicating activity in two circuits—one ventral and one dorsal—known to be disrupted in OCD (Harrison et al. 2009; van den Heuvel et al. 2016; Shephard et al. 2021). Supplementary Figure 10 summarizes the biological simplifications made in our model. Since functional coupling can occur without direct anatomical connectivity, we allowed cross coupling between brain circuit regions to also capture indirect functional paths not explicitly modelled. For example, because there is topographic mapping along the ventromedial-dorsolateral axis from the frontal cortex to the striatum (Klaus et al. 2017), and from the striatum back to the cortex via the globus pallidus (Bertino et al. 2020) and the thalamus nuclei (Sidibe et al. 1997; Mitchell and Chakraborty 2013; Kenwood, Kalin, and Barbas 2022), we abstracted a complex cortico-striato-pallido-thalamo-cortical circuit into a simpler striato-cortical circuit. Our approach allowed to capture these complex cortico-striato-pallido-thalamo-cortical circuits via a simpler striato-cortical circuit containing only regions showing a significant group difference in functional connectivity (Naze et al. 2023).

“(…), our model treats the net effects of striato-thalamo-cortical connections through a noise-affected driven striato-cortical influence that can be excitatory or inhibitory depending on the moment-by-moment relative contributions of the different pathways”.

Supplementary Figure 10: Diagram of the basal ganglia circuitry and network simplifications. The frontal cortex can be coarsely functionally partitioned into a ventromedial (affective) circuit and a dorsolateral (cognitive) circuit (van den Heuvel et al. 2016; Shephard et al. 2021). The basal ganglia encompass the striatum, the globus pallidus (GP), the substantia nigra (SN) and the subthalamic nucleus (STN) (left panel). The net effect of the basal ganglia projections to the cortex via the thalamus has been conceptualized as the direct (disinhibitory, blue) and indirect (inhibitory, pink) pathways. Marked

changes in functional connectivity in the ventromedial circuit (in orange) and the dorsolateral circuit (in green) were previously identified in OCD using neuroimaging (Naze et al. 2023). The whole system (left panel) is simplified in our network model (right panel), where striato-pallido-thalamo-cortical projections and the interplay between direct and indirect pathways are reduced to a dynamic coupling with both excitatory and inhibitory effects. OFC: orbitofrontal cortex, mPFC/IPFC: medial / lateral prefrontal cortex, NAcc: nucleus accumbens, GPe/GPi: globus pallidus externa/interna, MDmc/MDpc: magnocellular/parvocellular mediolateral thalamus.

6. The results section discusses various parameters, but it is unclear which parameter has the greatest impact on OCD. Could you identify the most influential parameter and discuss interventions targeting it?

R1-6: This feedback helped us to revise the results section, emphasizing the impact of the interventions with respect to their feasibility.

Our results show that the parameter substitutions achieving the best intervention outcomes differ according to the number of neural targets. If changing only one parameter, the single greatest effect comes from the lateral PFC to Putamen coupling, C_{PL} . This parameter also performs well in interventions targeting multiple parameters. However, the “best” set of parameters ultimately depends on any given therapeutic approach. For example, Deep Brain Stimulation (DBS) can target a restricted number of parameters (e.g., coupling between NAcc and prefrontal cortex (Figeo et al. 2013)) while less specific pharmacological interventions could impact a broader set of parameters. Therefore, the “best” set of parameters for a DBS intervention is not the same as the one to be considered for a pharmacological treatment. Our work provides information on what neural mechanism should be targeted to normalise frontostriatal activity in OCD given the specificity of the selected approach.

We revised the text in the Results section to highlight the above considerations:

p.10, line 328: “That is, if an intervention can only target one parameter, it should target the coupling from the lateral prefrontal cortex to the putamen (C_{PL}).”

p.10, line 334: “Notably, the combinations of parameter substitutions achieving the best intervention outcomes differ according to the number of targets n_t indicating a non-trivial interplay between target points and their resulting outcomes. Thus, instead of considering the optimal neural parameters in absolute terms, the optimal target parameters are relative to the specificity of the planned intervention (e.g., DBS on the NAcc may only be able to change the coupling between this region and the frontal cortex while pharmacological interventions can alter a broader set of parameters).”

p.10, line 349: “Results showed that when only a few neural targets are concurrently adjusted (≤ 4), increasing dorsolateral cortico-striatal coupling (C_{PL}) had the highest association with group-level improvements in functional connectivity across virtual interventions. On the other hand, interventions decreasing the bidirectional (C_{AO} and C_{OA}) coupling in the ventromedial circuit delivered the best functional outcomes across virtual interventions when using more (>4) concurrent targets. These findings highlight different target opportunities based on the ability of a planned intervention to selectively change the relevant neural processes.”

7. The discussion section would benefit from a clearer comparison with existing research to highlight the novelty of this work.

R1-7: In line with the reviewer's comment, we added the following text in the Discussion (p.12, line 449):

“Computational work focusing on obsessions and compulsions is relatively scarce. Early neural network models (Rumelhart and McClelland, 1986) were based on other psychiatric disorders such as schizophrenia (Cohen and Servan-Schreiber 1992; Stein and Hollander 1994; Ownby 1998). While these models simulated behavioural measures captured by cognitive tasks with reasonable success, they did not assess the brain basis of behaviour. The next generation of models were informed from contemporary discoveries concerning the basal ganglia microcircuitry and the dopaminergic system (Steiner and Tseng 2016). These models focused on neural mechanisms supporting reinforcement learning (Joel et al. 2002), serial processing (Dominey 1995), and action selection (Gurney et al. 2001a; 2001b; Humphries and Gurney 2002) and were applied to gain insights on Parkinson's disease or addiction (see a review by Gillies et al. (Gillies and Arbuthnott 2000)). Advances in modelling approaches allowed us to consider larger cortico-striato-thalamo-cortical circuits (Eliasmith et al. 2012; Tomkins et al. 2014; Kozloski 2016; Spreizer et al. 2017) and permitted a better integration of core findings from animal models of OCD (Gurney, Humphries, and Redgrave 2015). More recent modelling has implicated a role for serotonergic, dopaminergic and glutamatergic dysregulations in OCD (Yin et al. 2023). Moreover, the in-silico study of brain circuits can provide insights to guide the development of new therapeutic interventions and pharmacological neuromodulation (Yin et al. 2024). Our current modelling work extends on previous work by (i) providing mechanistic insight into recent neuroimaging findings on altered frontostriatal functional connectivity in OCD; (ii) using recent advances in parameter estimation to infer distributions of biophysical model parameters, rather than point estimates; (iii) systematically exploring the efficacy of a range of potential interventions to restore healthy neural parameter values, considering interventions both individually and working in combination; (iv) showing that a digital twin framework can be used probe hidden neural processes associated with changes in functional connectivity; and (v) linking changes in symptom severity to changes in neural parameters.”

8. For interesting reader, computational modeling studies on OCD should be extended, such as these references:

[1] Yin L, et al. Unveiling serotonergic dysfunction of obsessive-compulsive disorder on prefrontal network dynamics: a computational perspective. *Cereb Cortex*. 2024 Jun 4;34(6):bhae258. doi: 10.1093/cercor/bhae258.

[2] Yin L, et al. A computational network dynamical modeling for abnormal oscillation and deep brain stimulation control of obsessive-compulsive disorder. *Cogn Neurodyn*. 2023 Oct;17(5):1167-1184. doi: 10.1007/s11571-022-09858-3

R1-8: Thank you for highlighting these interesting papers, we now cite these (see previous response R1-7).

Minor Comments:

9. Lines 25-26: The sentence tenses are inconsistent, with multiple tenses used within one sentence.

R1-9: We have reworded the sentence (Abstract, p.1, line 27): “We found that bidirectionally decreasing spontaneous neural coupling in the ventromedial circuit while concurrently increasing dorsolateral cortico-striatal coupling delivered the highest functional improvements in OCD”

10. The reference formatting is inconsistent throughout the paper, which affects readability. You should double-check.

R1-10: We apologize for this inaccuracy. We have now homogenized the references following the Nature Communications formatting guidelines.

11. The colors are too light, making it difficult to distinguish between the control and OCD groups in Clo and Col in Figure 2. Increase the color contrast and add axis labels to clarify the biological significance of the parameters and neural couplings.

R1-11: Thank you for pointing this visualization issue. We increased the contrast of several details (e.g., histograms). In the figure caption, we also clarified that the y-axis labels refer to the normalized density of each parameter value.

Figure 3: Altered activity of coupled ventral and dorsal frontostriatal circuits in OCD. A. Schematic of the coupled model space comprising the ventral (nucleus accumbens [NAcc]-orbitofrontal cortex [OFC]) and dorsal (putamen [dPut]-lateral prefrontal cortex [LPFC]) frontostriatal circuits and their associated neural parameters (details in Table 2). Brain regions were extracted from the Schaefer et al. atlas (100

regions) for illustrative purposes. Note that C_{XY} , η_{XY} , and σ_{XY} are parameters for the connection from region Y to region X (right panel). Global (region unspecific) parameters do not have subscripts. C terms are coupling strengths, η and σ terms are drift and volatility properties of noise. **B.** Posterior probability distributions (histograms) and estimated density functions (y-axis, ρ , solid lines; values are normalised such that $\rho = \frac{a}{\sum_n a \times \Delta\theta}$ where a is the number of values in a bin and $\Delta\theta$ is the bin width) of each model parameter (x-axis, parameter values) after fitting the model to controls (blue) and OCD (orange) functional connectivity from fMRI resting-state data. Mann-Whitney U-test (two-sided): * $p_{FWE} < 0.05$. Significant effects with medium and strong effect size are denoted by ** (Cohen's $d > 0.5$) and *** ($d > 0.8$).

12. Figure 3 is overly complex, and it is hard to get the main message. Consider simplifying with two-parameter plots or other formats to show the intervention effects of multiple parameter combinations.

R1-12: We appreciate the feedback. To highlight how a parameter or set of parameters are associated with the resulting intervention efficacy, we added Supplementary Figure 6. This new figure facilitates the link between the representation of the statistical outcome of each intervention (Fig 4B) and the summary contribution of each parameter (Fig 4C). In addition to the new figure, the following text has been changed:

Methods (p.20, line 757): “We then used the dot-product between $\theta_{pre-post}^z$ and $d_{pre-post}$ across *all* virtual interventions showing a statistically significant difference ($AUC > 0.533$, $p_{FWE} < 0.05$) to quantify the relationship between changes in the parameters’ values and improvement in functional connectivity. To give the reader an intuition of this measure, we provide the linear regression plots between those two variables $\theta_{pre-post}^z$ and $d_{pre-post}$ for the best interventions per number of targets in Supplementary Figure 6. The final parameter contribution using the dot-product is an aggregate measure of those regressions across *all* virtual interventions involving n_t number of targets, rather than for only the *best intervention per number of targets.*”

Results (p.10, line 342):

“We used the dot-product to quantify how changes in each model parameter contribute to restoring healthy frontostriatal connectivity (Methods) across virtual interventions. Briefly, this approach identified which neural targets contribute the most to the overall restoration of OCD frontostriatal dynamics across *all significant* interventions at the group level (Figure 3C). For each of the best interventions per n_t number of targets, the correlation between parameter changes and functional connectivity improvements are provided as regression plots in Supplementary Figure 6. Figure 3C is an aggregate measure of those correlations across *all* interventions and not only the absolute best ones.”

Supplementary Figure 6: Associations between parameter values and intervention efficacies. Z-score normalized parameter values (θ^Z , where θ represents the parameter name; x-axis) against intervention efficacy (difference in distance to healthy controls in functional connectivity space, ΔFC ; y-axis). Numbers of targets n_t are indicated by the same color code as Figure 4). Only the best intervention (highest AUC) is shown for each number of targets.

In addition, we restructured Figure 4 (originally Figure 3), highlighting the control benchmark.

Figure 4: Effect of targeted parameter changes to restore frontostriatal brain dynamics in OCD. A. Schematic of the virtual intervention approach: Sets of $n_t = 1 \dots 6$ posterior distribution(s) were systematically permuted between parameter sets for OCD and healthy controls, simulating the restoration of healthy parameter ranges in OCD. The statistical outcome of each intervention was estimated using the difference in distance between the simulated healthy functional connectivities (virtual control cohorts) and the simulated OCD connectivities computed *after* the neural parameters were permuted (virtual interventions, Figure 1C-D, Methods, and Supplementary Figure 5). **B.** Statistical outcome of the targeted virtual interventions. The five most effective interventions to normalise changes in OCD functional connectivity for each number of targets n_t (permuted parameters listed on the y-axis and highlighted by colour bands) are shown. The area under the receiver operating characteristic curve (AUC) is derived from the Mann-Whitney U statistic and denotes the probability that a virtual intervention improves frontostriatal functional connectivity. $n_t = 0$ corresponds to the baseline OCD and controls simulated functional connectivity. The vertical dashed line indicates the AUC at which $p_{FWE} = 0.05$. Average AUC $\langle AUC \rangle$ per number of intervention (n_t) indicates a logarithmic scaling (inset). **C.** Dot-product between normalised parameter changes and the improvement of functional connectivity

across statistically significant virtual interventions ($p_{\text{FWE}} < 0.05$), indicating the association between a modulation of parameter (increased or decreased values) and the resulting improvement in functional connectivity. The colour code relating to the number of targets n_t is shared across panels.

13. The table formatting is inconsistent throughout the manuscript.

R1-13: Thank you for noticing this. We have standardized the Table formatting in the revised version of the manuscript.

14. In Figure 1B and C of the supplementary materials, it would be helpful to define what S_1 and S_2 represent in the legend. Also, do the blue and green lines represent $dS_1/dt = 0$ and $dS_2/dt = 0$? This is unclear from the figure. Additionally, the dashed line in panel C is not prominent, and adding annotations to the figure would improve comprehension.

R1-14: We edited the figure and legend accordingly:

Supplementary Figure 2: The single circuit model adopted to study frontostriatal neural dynamics at rest. **A.** Schematic of the model derived from Wong & Wang (2002) and Deco et al. (2013), incorporating static frontostriatal (C_{21}) and stochastic striato-cortical (C_{12} , η_{12} , σ_{12}) coupling terms (Methods). **B.** Time series of the model state variables S_1 (frontal cortex), S_2 (striatum), and \widetilde{C}_{12} (striato-cortical projection, with sign determining excitation or inhibition). **C.** State-space representation of the bistable frontal (S_1) and striatal (S_2) neural dynamics (i.e., average synaptic gating). Nullclines ($\frac{dS_1}{dt} = 0$, blue; and $\frac{dS_2}{dt} = 0$, green) intersections highlight stable fixed points (black circles, bottom left: low activity state; top right: high activity state) and an unstable fixed point (white circle). The trajectory (grey trace) is the resulting projection of S_1 and S_2 timeseries from panel A. **D.** Stability analysis of the model as a function of striato-cortical coupling (C_{12}). The two variables (cortical S_1 and striatal S_2 activity) exhibit stable (solid) and unstable (dash-dotted) equilibria separated by saddle-node bifurcations (SN1 and SN2 circles) that demarcate the ends of the bistable region. Background trajectories correspond to the projections of timeseries shown in panel A in $S_1 - C_{12}$ (blue) and $S_2 - C_{12}$ (red) spaces. **E.** Functional connectivity (Pearson's R) between striatum and frontal cortex as a function of drift η_{12} and volatility σ_{12}

parameters. **F.** Transition rate (number of zero-crossings in striato-cortical coupling per minute) as a function of drift η_{12} and volatility σ_{12} parameters.

Conclusion:

Overall, the manuscript presents valuable insights, but addressing these concerns would significantly strengthen its clarity.

R1-15: We appreciate the constructive feedback. Addressing the comments has greatly improved the clarity of the manuscript.

Reviewer #1 (Remarks on code availability):

Code is OK !

R1-16: Thank you for looking at the code.

Reviewer #2 (Remarks to the Author)

This is an interesting study which combined neuroimaging and behavioural data with computational modelling, focused on investigating the neural mechanisms driving imbalances in frontostriatal circuit activity in OCD patients. The results found that neural variability and cross-regional coupling play a key role in OCD frontostriatal pathophysiology. And they also found bidirectionally decreasing spontaneous neural coupling in the ventromedial circuit while concurrently increasing dorsolateral cortico-striatal coupling delivers the highest functional improvements in OCD. These findings will advance our understanding of the underlying neural mechanisms of OCD and demonstrates significant potential for future clinical translation. However, I have several reservations regarding the manuscript in its current form.

We thank the reviewer for this positive appraisal of our work and constructive feedback.

Introduction

1. The authors mentioned that the role of inhibitory balance in microcircuit dynamics and the use of biophysical models cannot fully explained why these existing mechanisms and models cannot be directly applied. It should further clarify “why existing mechanisms cannot be directly used in clinical applications” and “how the biophysical model specifically overcomes these limitations.”

R2-1: Thank you for pointing out this lack of clarity. We have revised the text to better emphasize the challenges of translating knowledge on microscale deregulations in animal models of OCD onto clinical applications, and to highlight how our approach can facilitate progress in understanding the neural basis of OCD and the development of new therapies:

Introduction (p.2, line 47):

“Preclinical work suggests a key role for inhibitory balance in striatal microcircuit dynamics and its coupling to prefrontal activity (Burguière et al. 2013; Vollweiler et al. 2023). However, directly translating these preclinical results to individuals with OCD remains a challenge (Ahmari 2016). For example, animal models lack the human brain’s structural and functional complexity, particularly in pathways comprising the prefrontal cortex (Pang et al. 2022; Smaers et al. 2017). Biophysical models provide a viable way to link preclinical advances at the microscale with clinical findings at the macroscale (Breakspear 2017; Shine 2021; Dutta et al. 2023; H. E. Wang et al. 2023). Thus, models have the potential to advance knowledge on the neural basis of symptoms in people with OCD and fast-track the development of therapies by delivering testable predictions on the macroscale effects of changes in specific neural mechanisms. Accordingly, stochastic dynamic causal modelling (DCM) has shown that changes in resting-state fMRI connectivity in OCD relate to altered neural couplings between striatal and frontal cortices differing along the ventro-dorsal axis (Naze et al. 2023). Such an approach previously facilitated the development of model-informed pilot interventions (Gollo, Roberts, and Cocchi 2017), leveraging recent progress in neuromodulation for depression (Cash et al. 2021). However, DCM employs a generic attractor model to generate regional resting-state activity (Friston et al., 2014), which does not explicitly capture the complex functional interplay within and between striatal and frontal regions affected by OCD. On the other hand, recent phenomenological models provide elaborate statistical frameworks to explain state transitions in the OCD brain or

behaviours, but lack an explicit mapping to the neurobiology and neuroimaging data (Fradkin et al. 2020; Sato, Shimomura, and Morita 2023).”

Introduction (p.2, line 67):

“Our model was built to capture key neural processes known to underpin deregulations in the OCD frontostriatal system. Formalising these neurobiological hypotheses into a tractable mathematical model using a top-down approach balancing biological plausibility and complexity, we predict the changes in intrinsic neural activity that could restore healthy frontostriatal functional connectivity in OCD subjects. Because of their key relevance for OCD pathophysiology (Robbins, Vaghi, and Banca 2019; Stein et al. 2019), we simulate ventral and dorsal frontostriatal circuits by building on an established model summarising the evolution of excitatory and inhibitory neural activity (Wong and Wang 2006; Deco et al. 2013). The model captures [...]. Critically, it provides a coarse-grained description of core neural processes known to be impacted by OCD, including changes in the basal ganglia functional microcircuits (Burguière et al. 2013; Piantadosi et al. 2024) and frontostriatal neural couplings (Welch et al. 2007; Ahmari et al. 2013; Mondragón-González, Schreiweis, and Burguière 2024). The model incorporates key anatomical constraints linked to pathways connecting the striatum with the frontal cortex (Fettes et al. 2017; Naze et al. 2023). This mesoscale model allows the estimation of neural changes within the frontostriatal system, by identifying neural mechanisms underpinning altered fMRI resting-state activity in OCD and their relationship to symptom severity.”

2. The basis for selecting the "core parameters" in the biophysical model is not clearly explained. Why did select the core parameters? and how is related to OCD-related neural coupling changes demonstrated? It is recommended to add more details on model validation and the theoretical or experimental basis for selecting these core parameters.

R2-2: We appreciate the Reviewer’s comment, which points to some foundational assumptions about our model choice and unclear terminology. We now clarified in the Introduction that “core neural processes” (former “core parameters”) refer to well-documented neurophysiological processes suggested to be affected in OCD. These processes include local neural interactions balancing inhibition and disinhibition in the basal ganglia (Graybiel and Rauch 2000; Ahmari and Rauch 2022) and defined cortico-basal ganglia projections (Sidibe et al. 1997; Klaus et al. 2017; Bertino et al. 2020). On the other hand, “core parameters” only refer to the “statistically relevant parameters” of the model. We have now clarified that this means parameters showing significant difference between OCD and healthy controls in our model after running the Bayesian optimisation.

Introduction (p.3, line 87):

“Using a formal Bayesian optimisation method, we isolate statistically relevant parameters and predict their contribution to restoring healthy frontostriatal activity. We then test those predictions using a longitudinal dataset and a digital twin paradigm.”

Further, we elaborated on the rationale behind our model simplification, and added a new Supplementary Figure:

Supplementary Figure 10: Diagram of the basal ganglia circuitry and network simplifications. The frontal cortex can be coarsely functionally partitioned into a ventromedial (affective) circuit and a dorsolateral (cognitive) circuit (van den Heuvel et al. 2016; Shephard et al. 2021). The basal ganglia encompass the striatum, the globus pallidus (GP), the substantia nigra (SN) and the subthalamic nucleus (STN) (left panel). The net effect of the basal ganglia projections to the cortex via the thalamus has been conceptualized as the direct (disinhibitory, blue) and indirect (inhibitory, pink) pathways. Marked changes in functional connectivity in the ventromedial circuit (in orange) and the dorsolateral circuit (in green) were previously identified in OCD using neuroimaging [Naze et al., 2023]. The whole system (left panel) is simplified in our network model (right panel), where striato-pallido-thalamo-cortical projections and the interplay between direct and indirect pathways are reduced to a dynamic coupling with both excitatory and inhibitory effects. OFC: orbitofrontal cortex, mPFC/IPFC: medial / lateral prefrontal cortex, NAcc: nucleus accumbens, GPe/GPi: globus pallidus externa/interna, MDmc/MDpc: magnocellular/parvocellular mediolateral thalamus.

The following text in the Results has also been added (p.5, line 165):

“We introduced three ~~new~~ new parameters to describe the dynamic coupling from striatal to frontal regions: mean strength, relaxation toward the mean (drift), and variance of endogenous fluctuations (volatility). The drift parameter (η_{ij}) refers to the rate at which the coupling strength returns to its mean level. Volatility (σ_{ij}) refers to the amplitude of random perturbations in the dynamic striato-cortical coupling; i.e., how much direct-indirect pathway variability is expressed by the striato-thalamic dynamics. In contrast, the functional impact of descending glutamatergic projections from the frontal region to the striatum (Shepherd, 2013) is modelled using traditional static coupling.

Based on previous genetic (Mattheisen et al. 2015), animal (Ahmari et al. 2013) and clinical work (Naze et al. 2023; Fettes, Schulze, and Downar 2017; Peters, Dunlop, and Downar 2016), we hypothesize that those neural parameters explain how local changes in microcircuit activity drive altered frontostriatal dynamics in OCD. (...)

Accordingly, these processes parameterize the net effect of the excitatory-inhibitory balance in frontostriatal circuits, influencing the stability of the whole striato-cortical system (Figure 2).”

Finally, the results section (p.8, line 286) has been changed as follows:

“Next, sets of up to six parameters (out of the 11 parameters showing significant group differences, Figure 3B) were systematically permuted between control and OCD posterior distributions to run an additional 1.5 million simulations (1000 different simulations for each of the $\sum_{n=1}^6 \frac{11!}{n!(11-n)!} = 1485$ combinations of parameter permutations).”

And in the Methods section we specified (p.16, line 614):

“Parameters a, b, d, γ, w and I_0 were based on their prior literature estimates and lack of hypotheses for them varying in OCD.”

Methods

1. It is unclear about the standards for constructing virtual data. Could the authors provide a more detailed explanation of the process of constructing virtual data based on real data, and how to avoid the problem of over-simplification?

R2-3: As requested, we clarified the approach we used to generate and compare virtual data. Specifically, the revised text now emphasizes that virtual data were created using statistical sampling, explaining how the adopted approach avoided oversimplification. Below, we detail the changes made to the Methods. We also added a new Supplementary Figure to facilitate the understanding of our approach.

Methods (p.18, line 693):

“For the synthetic dataset, drawing parameters from the posterior distributions defined via the above optimisation, we ran 1000 simulations (each simulation corresponding to a virtual subject) for each group (OCD and controls). Next, we divided the virtual subjects from each group into 20 cohorts of 50 subjects. We decided the size of these cohorts based on the size of our empirical dataset. We refer to these cohorts as reference virtual cohorts. The dissimilarity between these reference cohorts was calculated using a distance metric in functional connectivity space ($d(A, B)$, Supplementary Figure 5).

In the restoration analysis, for each virtual intervention, we generated an additional set of 1000 simulations (i.e., generating a new set of virtual subjects). As performed for the reference cohorts, we divided the resulting virtual subjects into 20 cohorts of 50 subjects. For these simulations, the specific parameters targeted by the intervention are drawn from the controls’ posterior distributions, while the other parameters remained drawn from the OCD posterior distributions. This framework allowed us to explicitly test for the effect of restoring OCD-related parameters to control values. We quantified the effect of an intervention (i.e., functional improvement) using a distance metric in functional connectivity space. Specifically, we calculated the distance between the reference control cohorts and the virtual intervention cohorts ($d(A, B')$, Supplementary Figure 5), resulting in a sample of $20 \times 20 = 400$ distances. This vector $\mathbf{d}(post, baseline_{hc})$ of size 400 reflects the distribution of FC distances between the virtual intervention cohorts (of a specific intervention) and the simulated control cohorts. Likewise, $\mathbf{d}(baseline_{OCD}, baseline_{hc})$ is the distribution of distances between the simulated OCD before any intervention and the simulated controls.

We assessed the outcome of a virtual intervention with a nonparametric approach. We used the one-sided Mann-Whitney U test to quantify the improvement in the distance $d(post, baseline_{hc})$ relative to the starting point $d(baseline_{OCD}, baseline_{hc})$, thereby assessing how virtual interventions change the control-to-OCD distance in connectivity space relative to the initial group distance. This is motivated by the focus on finding those interventions that improve FC (decrease of distance to healthy controls FC) rather than seeking all changes including deleterious ones. We quantify the effect size for each intervention using the area under the receiving operating characteristic curve (AUC), which scales the U-statistic by the number of samples in each group, resulting in the probability of $d(post, baseline_{hc})$ being smaller than $d(baseline_{OCD}, baseline_{hc})$ (Mason and Graham 2002), i.e., that the virtual intervention results in an improvement of functional connectivity. Here,

$$AUC = \frac{U}{n_1 n_2}, \quad (5)$$

where $n_1 = n_2 = 400$ samples. $AUC < 0.5$ indicates a negative outcome, i.e., the intervention increases the distance between OCD and controls, while $AUC > 0.5$ indicates a positive outcome, i.e., the distance between OCD and control decreases.”

Supplementary Figure 5: Generation of virtual cohorts and evaluation of virtual interventions. A. Parameters are drawn from OCD (orange) and control (blue) posterior distributions to create 2000 reference virtual subjects, 1000 in each group. Virtual interventions are modelled by drawing from the reference control group distributions for the parameters targeted by the intervention and drawing from the reference distributions of the OCD group for parameters not targeted by the intervention. 1000 virtual subjects are generated to create the virtual intervention cohorts. **B.** For each group, the 1000 virtual subjects are separated into 20 cohorts of 50 subjects. We computed functional connectivity (FC) distances between all controls and OCD cohorts [$d(A, B)$; i.e., *reference*]; and all controls and virtual interventions cohorts [$d(A, B')$; i.e., *intervention*]. **C.** The distribution of FC distances between *reference*

(orange) and *intervention* (purple) is compared (decrease in the distance implies functional improvement). **D.** The efficacy of the intervention is statistically quantified using a Mann-Whitney U test between *reference* and *intervention* distributions. **E.** Normalization of the U statistic by the number of samples leads to the AUC for which scores above 0.5 denote functional improvement.

Methods section (p.19, line 742):

“This virtual intervention approach mitigates the overfitting problem inherent to small datasets. By creating 20 virtual cohorts of 50 virtual subjects, we capture finite sample size effects inherent in our data, rather than generating arbitrarily large ensembles of simulations to obtain arbitrarily precise parameter estimates. Moreover, we focus on distributions of parameters rather than point estimates. To further avoid oversimplification, we also systematically investigated a wide range of scenarios from single parameter changes to combinations of parameters describing more complex interventions.”

Results

1. The result plots in Figure 2B should be adjusted to match the order of the statistical results in the text, which might enhance clarity and make the presentation more directly.

R2-4: Thank you for this suggestion, it improved the readability of the figure (see below).

Figure 3: Altered activity of coupled ventral and dorsal frontostriatal circuits in OCD. **A.** Schematic of the coupled model space comprising the ventral (nucleus accumbens [NAcc]-orbitofrontal cortex [OFC]) and dorsal (putamen [dPut]-lateral prefrontal cortex [LPFC]) frontostriatal circuits and their associated neural parameters (details in Table 2). **Brain regions were extracted from the Schaefer et al. atlas (100**

regions) for illustrative purposes. Note that C_{XY} , η_{XY} , and σ_{XY} are parameters for the connection from region Y to region X (right panel). Global (region unspecific) parameters do not have subscripts. C terms are coupling strengths, η and σ terms are drift and volatility properties of noise. **B.** Posterior probability distributions (histograms) and estimated density functions (y-axis, ρ , solid lines; values are normalised such that $\rho = \frac{a}{\sum_n a \times \Delta\theta}$ where a is the number of values in a bin and $\Delta\theta$ is the bin width) of each model parameter (x-axis, parameter values) after fitting the model to controls (blue) and OCD (orange) functional connectivity from fMRI resting-state data. Mann-Whitney U-test (two-sided): * $p_{FWE} < 0.05$. Significant effects with medium and strong effect size are denoted by ** (Cohen's $d > 0.5$) and *** ($d > 0.8$).

2. In the results section, it is mentioned that the impact of changes in cortico-cortical coupling (COL), striato-cortical drifts (η_{OA} , η_{LP}) and volatility (σ_{OA} , σ_{LP}) on functional connectivity is subject-specific and may not be generalizable. Would provide additional explanation or detailing how this subject-specific effect was identified?

R2-5: The Reviewer raised an important point regarding how our results speak to individual differences. We would like to apologize for the lack of clarity, re-reading the manuscript made us realize that we did not sufficiently elaborate on this point. To clarify we now provide a Supplementary Figure with the following explanatory text:

Results (p.10, line 355):

“In addition, results suggested that the effects of changes in cortico-cortical coupling (C_{OL}), striato-cortical drifts (η_{OA} , η_{LP}) and volatility (σ_{OA} , σ_{LP}) on functional connectivity are subject-specific. In other words, magnitudes and directions of change in these targets do not generalize to the whole population across interventions. As a representative example, we plotted the direction and magnitude of changes needed to restore two virtual OCD subjects' parameters to the control average values (Supplementary Figure 7). Results highlighted that when the underlying parameter distributions (OCD versus controls) largely overlap, many subjects will have parameter changes in opposite directions to achieve restoration of healthy functional connectivity (i.e., the direction of change is subject-specific). Conversely, when the distributions are very distinct, the direction of the targeted change is largely consistent across the cohort.”

Supplementary Figure 7: Individualised intervention framework. Digital twin parameters (vertical coloured lines; green: OCD #01, purple: OCD #31) are displayed alongside the control group’s average (black vertical line). Arrows indicate the direction (arrowhead) and amplitude (vector length) of the targeted changes in specific neural parameters to restore healthy neural dynamics at the individual level. Background distributions are posteriors of OCD (orange) and controls (blue) as per Figure 3 (main text).

Discussion

1. The study initially demonstrated that the dorsolateral cortico-striatal coupling C_{PL} consistently plays a key role in reducing the difference in functional connectivity between OCD patients and healthy individuals across various target combinations. However, in the subsequent statistical analysis regarding symptom improvement, it was found that the original parameters of C_{PL} in the digital twin model did not show a significant difference between baseline and follow-up. It would be beneficial to discuss this finding in the discussion section, and add more implications for symptom improvement?

R2-6: We thank the reviewer for the careful reading of the paper. The comment prompted us to double-check the analyses and detect an error in the digital twin pairing at follow-up (*post*) saved in our pipeline. Note that initial (*baseline*) twins were not affected. Consequently, the original figure did not accurately reflect the parameters linked to the empirical OCD data. We now rectified the error and updated Figure 5 accordingly with the updated set of digital twins. The change of each parameter in relation to symptom improvement ($\Delta YBOCS$) reflects the dominant contributions of frontostriatal couplings in the ventromedial and the dorsolateral circuits to behavioural improvement (Fig 5B). The longitudinal raw parameter differences confirm significant changes in both top-down (C_{AO}) and bottom-up (C_{OA}) couplings in the ventromedial circuit (Fig 5C). These updated findings include the increase of lateral PFC coupling to Putamen (C_{PL}) highlighted in the original results.

Updated Figure 5 and results section:

Figure 5: Digital twin analysis on the core neural mechanisms supporting changes in frontostriatal functional connectivity and OCD symptoms over time. **A.** Changes in resting-state functional connectivity (FC) and OCD symptoms' severity (Y-BOCS: Yale-Brown Obsessive-Compulsive Scale) over time (Δ , baseline *minus* follow-up) were linearly correlated. **B.** Dot-product between digital twin normalised parameter changes (increase (solid) or decrease (dashed) from initial to follow-up sessions) and the Y-BOCS changes across OCD subjects. This measure quantifies the correspondence of hidden neural processes to functional improvement in empirical data. **C.** Distributions of model parameters related to the digital twins of OCD subjects at baseline and follow-up (post). These parameters showed a significant time effect (two-sided paired Wilcoxon rank test: * $p_{\text{uncorrected}} < 0.05$, ** $p_{\text{FWE}} < 0.05$).

Result section (p.11, line 384): “We observed the strongest relationship in the cortico-striatal couplings of ventromedial (C_{AO}) and dorsolateral (C_{PL}) circuits (Figure 5B). Raw parameter differences in OCD digital twins between initial (baseline) and follow-up (post) appointments further supported this finding (Figure 5C, C_{AO} and C_{PL} , $p_{\text{FWE}} < 0.05$; paired Wilcoxon rank test (two-sided), $n = 48$). Results further suggest a decrease in the ventromedial striato-cortical coupling (C_{OA} , $p_{\text{uncorrected}} < 0.05$). Collectively, these results highlight the importance of modulating the neural couplings in both ventromedial and dorsolateral circuits to restore healthier resting-state functional connectivity in the OCD frontostriatal system.”

Reviewer #3 (Remarks to the Author)

This article explores a computational model of frontostriatal dynamics in obsessive compulsive disorder(OCD) that is used to explain a possible mechanism leading to those dynamics, compare them to those seen in controls, and suggest possible interventions that would normalize those dynamics based on the computational model. Empirical data leading to the model is based on a double blind sham controlled study of orbitofrontal continuous theta burst TMS and its effect on resting state functional connectivity conducted by the same group, that has been previously published in two widely cited journals. Dynamical systems theory and Bayesian inference are used to develop the model and its potential clinical applications. The article is well written and has numerous strengths including a large sample size, confirmatory evidence of the models fit to empirical data using longitudinal data and thoughtful extrapolation of existing computational models to frontostriatal circuitry in OCD. However, there are major concerns about the failure of the model to capture the highly distributed and complex nature of the circuit in question and therefore the authors conclusion that the model can be used to guide circuit based neuromodulatory treatment. The authors have not adequately delineated the limitations of the model or the empirical data on which it is based in the paper.

R3.1a: We thank the reviewer for the thoughtful feedback on our work. The reviewer raised important points regarding the construction of our model and its limitations that require clarification. We carefully considered and addressed these concerns in the following pages but would like to start by offering some general considerations.

We decided to use a top-down modelling approach aimed at simplifying biological complexity to interrogate brain circuits and neural mechanisms known to be impacted by OCD pathology. This hypothesis-driven approach has been proven valuable to explain core neural mechanisms supporting macroscopic network dynamics that remain hidden from experimental approaches due to technological limitations and biological complexity (Breakspear 2017; Sip et al. 2023; Munn et al. 2024). Accordingly, our work did not attempt to build a detailed “in-silico” representation of the human frontostriatal system but a model that contains key neurophysiological ingredients suspected to cause OCD frontostriatal pathology. Without strong rationales and hypotheses, increasing the model’s complexity induces a factorial growth of the parameter space that makes results difficult to interpret and translate. Thus, while our parsimonious model does not describe the full complexity of frontostriatal microcircuitry, it does permit linking changes in microscale mechanisms to macroscale observations. Moreover, our work provides a necessary initial platform for developing and comparing more complex models. To facilitate such future modelling work, we provide a well-documented, self-contained code and a GUI enabling the initial exploration of neural parameters’ effects on macroscale neural dynamics.

To address this general concern, we added the following text in the Introduction and toned down the conclusion

Introduction (p.2, line 65):

“Here, we develop a biophysical model of OCD frontostriatal pathology using dynamical systems theory, Bayesian inference, and a unique longitudinal dataset comprising clinical and neuroimaging information (Cocchi et al. 2023; Hearne et al. 2023; Naze et al. 2023). Our model was built to capture key neural processes thought to underpin deregulations in the OCD frontostriatal system. Formalising these neurobiological hypotheses into a tractable

mathematical model allows us to predict the changes in intrinsic neural activity that could restore healthy frontostriatal functional connectivity in OCD subjects.”

Discussion (p.14, line 539):

“Our findings support and advance knowledge on the opposing functional changes characterizing the activity of the ventral and dorsal frontostriatal circuits in OCD, suggesting new neural targets for research on obsessive thinking and compulsive behaviours ~~and precision neurotherapeutics.~~”

Regarding limitations of the model and empirical data, and the extent to which our results support our conclusions on possible treatment strategies (be they stimulation-based or pharmacological or cognitive-behavioural), we address these points in the detailed responses below.

Some of these limitations are listed below.

1) Frontostriatal connectivity findings in OCD differ significantly across studies. For example the 2009 Harrison paper found hyperconnectivity in the L anteromedial OFC as well as the anterior cingulate with the nucleus accumbens(NA) as opposed to the r medial frontal pole and NA in the current study.

R3-1b: There is strong converging evidence from animal and human studies supporting distinct ventromedial and dorsolateral activity changes in OCD (e.g., review of (Robbins et al. 2019)). Both Harrison et al. (2009) and Naze et al. (2023) capture this important functional dissociation. While some differences in the precise location of statistically thresholded brain clusters have been reported, a circuit-based analysis provides a consistent picture of frontostriatal changes in OCD (e.g., (van den Heuvel et al. 2016; Shephard et al. 2021). Also, please see our additional analyses detailed below). In general, a circuit-based interrogation of changes in functional brain connectivity has facilitated the reconciliation of apparent discrepancies in the localization of regional changes in mental disorders like depression (Cash et al. 2023; Segal et al. 2023). As such, those approaches have shown that heterogeneous loci are generally embedded within common functional circuits and systems.

In line with the above, we assessed the impact of statistical thresholding to explain across-study heterogeneity. Moreover, we tested if the detected changes mapped onto the same frontostriatal circuits. Specifically, we re-analysed the Naze et al. (2023) data, allowing for a more lenient statistical threshold ($p < 0.001$ uncorrected). In line with the findings of Harrison et al. (2009), results revealed higher functional connectivity between the NAcc and (i) the left anteromedial OFC and (ii) the bilateral anterior cingulate cortex. These results confirm that changes in functional connectivity across the two studies map onto the canonical ventromedial frontostriatal circuit. This functional circuit comprises the NAcc, the anteromedial OFC, the medial frontal pole, and the ACC (Supplementary Figure 1, panel C). To address this important comment, the following text and Supplementary Figures have been added:

Results (p.4, line 126):

“Based on information regarding key brain regions and resting-state connectivity patterns linked to OCD (Harrison et al., 2009; Robbins et al. 2019; Shephard et al. 2021; Naze et al. 2023) (Supplementary Figures 1 and 2), we modelled the frontostriatal system through interacting ventromedial (nucleus accumbens – orbitofrontal cortex) and dorsolateral (putamen – lateral

prefrontal cortex) functional circuits (Voorn et al. 2004; Pennartz et al. 2009). A potential concern with our approach is that the precise anatomical location of functional connectivity clusters showing a group difference may slightly differ between neuroimaging studies. For example, Harrison et al. (2009) reported OCD hyperconnectivity between the nucleus accumbens and the left anteromedial orbitofrontal cortex and the anterior cingulate. However, Naze et al. (2023) showed a more circumscribed hyperconnectivity comprising the nucleus accumbens and the right frontal pole. We interrogated these apparent discrepancies by re-analysing the Naze et al. (2023) data using a more lenient statistical threshold ($p < 0.001$ uncorrected rather than $p_{FWE} < 0.05$). In line with Harrison et al. (2009), results from this control analysis supported the presence of increased connectivity between the nucleus accumbens and the left anteromedial orbitofrontal cortex, as well as the anterior cingulate in OCD (Supplementary Figure 1).”

Discussion (p.14, line 520): “We here note that the reported peak location of the ventromedial hyperconnectivity and dorsolateral hypoconnectivity differ across studies (Supplementary Figures 1 and 2), and most likely subjects. However, these functional clusters belong to the same canonical networks thought to support different OCD symptom dimensions (van den Heuvel et al. 2016; Shephard et al. 2021; Robbins, Banca, and Belin 2024).”

Supplementary Figure 1: Comparison of nucleus accumbens (NAcc) functional connectivity maps between Harrison et al. (2009) and Naze et al. (2023). A. Analyses on the dataset adopted for the current study (Naze et al, 2023) showed that NAcc-frontal hyperconnectivity at rest is highest in the right medial orbitofrontal cortex (OFC) in OCD compared to controls ($|T| > 4.2$, $***p_{FWE} < 0.05$, at MNI $x=28$, $y=60$, $z=-2$). Adopting a more lenient threshold revealed higher functional connectivity in OCD also in the left medial OFC ($**p < 0.001$; uncorrected, $x=-24$, $y=60$, $z=8$) and the anterior cingulate cortex (ACC,

* $p < 0.01$; $x = -6$, $y = 38$, $z = 22$). **B.** Side-by-side comparisons between the left hemisphere hyperconnectivity in OCD relative to controls observed in Harrison et al. ($p_{FDR} < 0.05$, in red) and Naze et al. ($p < 0.001$, same color-code as panel A) show strong spatial overlap. **C.** Cortical regions showing significant functional connectivity with the NAcc ($p_{FWE} < 0.05$, see Naze et al., 2023). These maps were obtained using a large normative sample (HCP 1080, (Van Essen et al. 2013)). Results highlight that the frontal clusters showing higher functional connectivity between the NAcc and the frontal cortex in OCD across the two studies (panel A and B) map onto the same functional circuit. Coordinates are in the MNI space.

The Harrison et al studies showed a correlation with YBOCS severity while the current study did not.

R3-1c: The reviewer is correct. However, as mentioned above, the implication of the functional circuits of interest for OCD phenomenology is suggested by converging lines of evidence that span from preclinical (Ahmari et al. 2013) to clinical work (Figeo et al. 2013). In this context, our current data showed a relationship between connectivity and Y-BOCS scores when considering both ventromedial and dorsolateral circuits functional connectivity (Fig 5A). Thus, our results are in line with previous work and support the conclusion that combined changes in the resting state activity of the two considered frontostriatal circuits are linked to the expression of OCD symptoms.

To address this comment, we made the following changes to the text (Discussion, p.13, line 473):

“Capitalising on unique longitudinal clinical and neuroimaging data, we confirmed a linear relation between changes in frontostriatal functional connectivity and changes in OCD symptom severity over time. This relationship appeared when regressing OCD symptoms severity against functional connectivity in both ventromedial (NAcc-OFC) and dorsolateral (Putamen-LPFC) circuits. This is in line with previous reports from preclinical (Ahmari et al. 2013), neuroimaging (Harrison et al. 2009), and clinical work (Figeo et al. 2013) suggesting a relation between the considered frontostriatal circuits and OCD symptoms.”

The 2009 Harrison study found hypofrontality in a frontal region that was significantly anterior to the current study while the finding in the current study is located several millimeters behind the temporal pole.

R3-1d: Our previous study (Naze et al. 2023) reported reduced connectivity between the dorsal Putamen and a lateral prefrontal cluster centered at MNI $x = 53$, $y = 13$, $z = 19$ (going anteriorly up to $y = 18$). This statistically thresholded cluster is indeed 2 cm (+/- 5 mm) more posterior than the clusters reported by Harrison et al. (2009) at [$x = -43$, $y = 32$, $z = 7$] and [$x = 51$, $y = 33$, $z = 8$]. However, there are some important methodological differences between Naze et al. (2023) and Harrison et al. (2009) to consider. Naze et al. masked group comparisons using a frontostriatal circuits map estimated using an independent dataset. Contrary to the approach adopted by Harrison et al., this approach avoided the inflation of type I error. Additionally, Naze et al. used a different statistical threshold for the analysis (i.e., $p_{FWE} < 0.05$ versus $p_{FDR} < 0.05$ in Harrison et al. (2009)). By maintaining the appropriate (independent) masking approach, but relaxing the cluster statistics ($p < 0.005$ $T = 2.0$), our additional analyses also showed “hypofrontality” in frontal clusters overlapping with the one observed by Harrison et al. (2009) (see new Supplementary Figure 2 below). Furthermore, in line with our previous response (R3-1b), both the anterior and the posterior clusters reported in the two studies belong to the same dorsolateral frontostriatal circuit (see new Supplementary Figure 2, panel D).

To fully address the reviewer’s comment, we have made the following changes in the text (shaded fonts are changes already introduced above in R3-1b) and added a new Supplementary Figure:

Results (p.4, line 126):

“Based on information regarding key brain regions and resting-state connectivity patterns linked to OCD [Harrison et al., 2009; Robbins et al., 2019; Shephard et al., 2021; Naze et al., 2023] (Supplementary Figures 1 and 2), we modelled the frontostriatal system through interacting ventromedial (nucleus accumbens – orbitofrontal cortex) and dorsolateral (putamen – lateral prefrontal cortex) functional circuits [Voorn et al, 2004; Pennartz et al., 2009]. A potential concern is that the precise anatomical location of functional connectivity clusters showing a group difference may slightly differ between neuroimaging studies. For example, Harrison et al. (2009) reported OCD hyperconnectivity between the nucleus accumbens and the left anteromedial orbitofrontal cortex as well as the anterior cingulate. However, Naze et al. (2023) showed a more circumscribed hyperconnectivity comprising the nucleus accumbens and the right frontal pole. We interrogated these apparent discrepancies by re-analysing the Naze et al. (2023) data using a more lenient statistical threshold ($p < 0.001$ uncorrected rather than $p_{FWE} < 0.05$). Results supported the presence of increased connectivity between the nucleus accumbens and the left anteromedial orbitofrontal cortex, as well as the anterior cingulate in OCD (Supplementary Figure 1). Likewise, the dorsolateral circuit hypoconnectivity peaks reported by Harrison et al. (2009) and Naze et al. (2023) are also up to 2 cm apart, but align when using a less stringent threshold (Supplementary Figure 2). Crucially, our control analyses confirmed that group changes in functional connectivity across the two studies maps onto the canonical ventromedial frontostriatal circuit (van Heuvel et al. 2016; Shephard et al., 2021).”

Supplementary Figure 2: Comparison of dorsal putamen (dPut) functional connectivity maps between Harrison et al. (2009) and Naze et al. (2023). **A.** Results from analyses of our dataset (Naze et al., 2023) show hypoconnectivity between the dorsal putamen and the right lateral prefrontal cortex in OCD relative to controls ($|T| > 3.5$, $***p_{FWE} < 0.05$, MNI $x=53, y=13, z=19$). Hypoconnectivity between the putamen and more anterior frontal cluster is also observed when more lenient (uncorrected) statistical thresholds were applied ($**p < 0.001$ for the cluster in the left hemisphere ($x=-38, y=44, z=26$) and $*p < 0.005$ for the cluster in the right hemisphere ($x=42, y=38, z=26$)). **B.** Side-by-side comparisons between the results of Harrison et al. (2009) (green, $p < 0.001$) and Naze et al. (2023) (same color-code as in panel A) highlight a strong overlap between the frontal clusters showing reduced resting-state functional connectivity with the dorsal putamen. **C.** Cortical regions showing significant ($p_{FWE} < 0.05$) functional connectivity with the dorsal putamen. These maps are obtained using a large normative sample (HCP 1080 (Van Essen et al. 2013)). The results highlight that the frontal clusters showing reduced functional connectivity between the dorsal putamen and the frontal cortex in OCD across the two studies (panels A and B) map onto the same functional circuit. Coordinates are in the MNI space.

Although 9 patients who were med free are included the authors do not reference other papers showing that frontostriatal hyperconnectivity in OCD may be related to SRI treatment.

R3-1e: We thank the reviewer for this comment. Accordingly, we added the following clarifying text and performed a new analysis on the (lack of an) effect of medication status (note that the relation to other clinical measures will be addressed in R3-4 below):

Results section (p.12, line 404): “It has been suggested that selective serotonin reuptake inhibitors (SSRI) medication can change frontostriatal circuits activity (Saxena et al. 1999). In a complementary analysis, we show that those results are unlikely driven by medication status (Supplementary Figure 8), and that the observed functional connectivity changes in the ventromedial and dorsolateral circuits are not directly associated to other clinical measures besides OCD (Supplementary Figure 9).”

Supplementary Figure 8: Medicated *versus* unmedicated OCD subjects do not display clear differences in frontostriatal parameters and functional connectivity across time. A-B. Parameter differences ($\Delta\theta$) in OCD digital twins between initial and follow-up assessments. Medicated (blue) and unmedicated (orange) subjects do not show statistically significant differences in any model parameter. C. Frontostriatal functional connectivity between medicated (blue) and unmedicated (orange) OCD subjects. NAcc: nucleus accumbens, dPut: dorsal putamen, OFC: orbitofrontal cortex, LPFC: lateral prefrontal cortex.

The lack of reproducibility of the findings across studies may in part be due to grouped data. Though the authors claim that their framework could be easily adapted to inform individual and personalized interventions it is left unclear how this could be accomplished. The model depends on connectivity data that is consistent and reproducible across studies or individual subjects.

R3-1f: The comment pertaining to a perceived lack of reproducibility in the imaging data and brain-behaviour relationships has been addressed above (R3-1b, R3-1c and R3-1d). Here, we address the comment regarding how our framework could assist in personalising therapeutic interventions. Specifically, we provide a concrete example of how it could be used:

Results (p.10, line 358):

“As a representative example, we plotted the direction and magnitude of changes needed to restore two virtual OCD subjects’ parameters to the control average values (Supplementary Figure 7). Results highlighted that when the underlying parameter distributions (OCD versus controls) largely overlap, many subjects will have parameter changes in opposite directions to achieve restoration of healthy functional connectivity (i.e., the direction of change is subject-specific). Conversely, when the distributions are very distinct, the direction of the targeted change is largely consistent across the cohort.”

Supplementary Figure 7: Individualised intervention framework. Digital twin parameters (vertical coloured lines; green: OCD #01, purple: OCD #31) are displayed alongside the control group’s average (black vertical line). Arrows indicate the direction (arrowhead) and amplitude (vector length) of the targeted changes in specific neural parameters to restore healthy neural dynamics at the individual level. Background distributions are posteriors of OCD (orange) and controls (blue) as per Figure 3 (main text).

Discussion (p.14, line 514):

“Additionally, while we identified the most efficient intervention targets at the group level, the current framework can easily be adapted to inform the development of personalised interventions (Hollunder et al. 2024). As we illustrated, given an individual OCD patient’s frontostriatal functional connectivity, one can use our model to generate a digital twin to compare this subject’s model parameters to the distribution of control parameters. Connections

showing the largest deviation to the controls' values could be targeted by clinical interventions to lean towards healthier frontostriatal dynamics.”

2) The frontal cortex is a massively distributed system that contains several complex interacting networks. It is likely that integration across frontostriatal loops takes place not only in frontal cortex but also at the level of the striatum and thalamus. The computational model fails to explicitly capture the complex functional interplay with and between striatal and frontal regions and their modulation by other large scale networks. The authors state that the model implicitly incorporates thalamic connections but this is likely to be an unfounded assumption.

R3-2: We agree with the reviewer that integration within and across frontostriatal loops does not only take place in the frontal cortex but also in the striatum and thalamus. For this reason, our model explicitly includes reciprocal connections in the striatum (couplings C_{PA} and C_{AP}). However, for reasons of parsimony (as addressed above in R3.1a), the model did not explicitly incorporate the thalamus, instead we opted to model the net effect of striato-thalamo-cortical connections. Nevertheless, to directly address the reviewer's concerns we generated a new model incorporating the thalamus, including all cross-circuit connections. Adopting this new model, we re-performed the optimization while relaxing the original prior indicating that glutamatergic projections were necessarily excitatory (by doing so, we are also addressing the following comment, R3-3).

The following text has been added to motivate the new model (Results, p.7, line 255): “To test for the role of the thalamus and across-circuit frontostriatal interactions, we generated a revised model that explicitly incorporates the thalamus and couplings between the ventral and dorsal frontostriatal circuits. We also removed the assumption that glutamatergic projections always have excitatory post-synaptic effects. This more-general connectivity was motivated by the fact that frontal glutamatergic neurons can modulate inhibitory striatal interneurons, resulting in net inhibitory effects locally (Paraskevopoulou, Herman, and Rosenmund 2019). Supplementary Figure 4A provides a diagram of this new model with the six regions, the updated priors, and the normative coupling parameters' posteriors for controls and OCD after running the Bayesian optimization.”

Results from the model explicitly including a thalamus and integration across frontostriatal loops further localized the connections disrupted in OCD and identified changes in functional interplay between the ventromedial and dorsolateral circuits.

Results (p.8, line 262): “Results adopting this more-complex model show an OCD-specific disinhibition of the thalamus in the ventromedial circuit. Because removal of thalamic inhibition results in net increase in cortical excitation (Deniau and Chevalier 1985; Peters, Dunlop, and Downar 2016), this finding is consistent with the increased striato-cortical coupling observed in Figure 3 (C_{OA}). The lack of change in thalamo-cortical model parameters between OCD and controls also suggests that OCD-related disruptions are cortico-striatal and striato-thalamic, but unlikely thalamo-cortical. In this new model, cortico-striatal projections show opposite signs between the ventromedial and the dorsolateral circuits across groups. This result confirms a circuits imbalance in OCD compared to controls, further supporting that both circuits are affected antisymmetrically. Finally, this new analysis revealed that cortico-striatal and striato-

thalamic ventromedial inhibition over the dorsolateral regions is exacerbated in OCD compared to controls, suggesting an overly active cross-pathway inhibition at play in OCD.”

Despite the valuable confirmation and addition to the original results, the model including the thalamus did not improve the fit to empirical data (Supplementary Figure 4B). This highlights that adding biological complexity to our model did not improve its explanatory power. In fact, penalizing for this increased complexity (Penny et al. 2010; Daunizeau, David, and Stephan 2011) will ultimately result in a reduced explanatory power relative to the original model. We added the following text:

Results (p.8, line 272): “While incorporating the thalamus in our model provided confirmatory evidence and additional insights into frontostriatal dysregulations in OCD, the model did not reduce the residual error of the fit to empirical data (Supplementary Figure 4B). This finding indicates that the added biological granularity did not improve the model evidence (Penny et al. 2010), even before imposing a penalty for the increased model complexity (Daunizeau, David, and Stephan 2011), therefore the remaining of the study focuses on the simplified striato-thalamo-cortical projections introduced in Figure 3A.”

Supplementary Figure 4: Model extension with thalamic nuclei and relaxed priors confirms changes in neural coupling in and within the OCD ventromedial and dorsolateral circuits but fails to improve accuracy. **A.** Illustration of the fronto-striato-thalamo-cortical model with all cross-pathway (but the thalamo-thalamic (Swanson, Sporns, and Hahn 2019)) connections. For this model, we removed priors on the nature of cortical and thalamic efferents being only excitatory. Differences in couplings between controls and OCD are indicated by a change in arrow size (magnitude) and arrow ends (sign) as indicated in the legend. **B.** The root mean square error of the fitting to empirical data after inference (i.e., the inverse of model evidence) for the original model (main text, N = 4 regions, blue) and the relaxed model with thalamic nodes (N = 6 regions, orange). OFC: orbitofrontal cortex, mPFC/IPFC: medial / lateral

prefrontal cortex, NAcc: nucleus accumbens, GPe/GPi: globus pallidus externa/interna, MDmc/MDpc: magnocellular/parvocellular mediolateral thalamus.

Regarding modulation by other large-scale networks beyond those with strong priors for involvement in OCD, please note that assessing the influence of various other brain networks on frontostriatal circuits is beyond the scope of this study. However, our work provides a solid basis to facilitate future studies addressing this question.

Discussion (p.13, line 469):

“Moreover, the presented framework is made publicly accessible for future development, including model extensions with additional brain network regions, and applications to other clinical conditions.”

3) Dynamic causal modeling informs us about effective connectivity but does little to inform us about the actual neural processes underlying the hyperconnectivity. The assumption that regional bold activity is primarily related to an excitatory-inhibitory balance of neural firing rates in a given region has been increasingly challenged. Accumulating evidence points to the importance of presynaptic changes as driving regional metabolic rates and these may not be correlated with firing rates. The assumption that glutamatergic projections always have an excitatory effect on a given region is also not clear due to the density of synaptic interaction with both inhibitory as well as excitatory post synaptic neurons

R3-3: We would like to clarify that we *did not* adopt DCM in the current work. Our model critically differs from DCM, which for fMRI uses a phenomenological model (i.e., it is not a biophysical model as it does not explicitly model neurophysiological parameters). However, the Bayesian optimization framework used to fit the model parameters is not dissimilar to the Bayesian methods used in DCM analyses. To makes these important points clearer, we have revised the following text:

Methods section (p.20, line 769):

“The adopted sequential Monte-Carlo algorithm (SMC (Toni et al. 2008; Klinger and Hasenauer 2017)) is an advanced iteration of the variational Bayes approach used in Dynamic Causal Modelling (DCM (Friston, Harrison, and Penny 2003; Friston et al. 2014). Specifically, this approach alleviates the Laplace assumption, which treats posterior distributions as necessarily Gaussian (Friston et al. 2007; Friston et al. 2016).”

Discussion (p.14, line 535):

“We used a validated neural mass model explaining resting-state dynamics (Deco et al. 2013; 2014) rather than the generic noise-driven exponential decay model utilised in dynamic causal modelling (DCM for fMRI, (Friston, Harrison, and Penny 2003; Friston et al. 2014)). We added to this model a stochastic process abstracting the dynamics of the direct-indirect pathway interactions in striato-thalamo-cortical circuits relevant to OCD.”

On the origins of regional BOLD activity, we agree with the reviewer’s statement that metabolic rates are not only driven by firing rates but also presynaptic activity and noticed a mislabelling of the model’s variable S as firing rate in the manuscript that we corrected in both the Methods and Figure 2 caption:

Methods (p.16, line 602):

“The equations of the reduced Wong-Wang model (Deco et al., 2013) are:

$$\dot{S}_i = -\frac{S_i}{\tau_S} + (1 - S_i)\gamma H(x_i) + v_i, \quad (1)$$

$$H(x) = \frac{ax - b}{1 - \exp(-d(ax - b))}, \quad (2)$$

$$x_i = wJ_N S_i + GJ_N \sum_j C_{ij} S_j + I_0, \quad (3)$$

where S_i denotes the average synaptic gating of population i with a relaxation timescale of $t_S = 100$ ms;

$H(\cdot)$ is a nonlinear population’s average synaptic gating to firing rate transfer function, and $v_i(t)$ is a Gaussian white noise process with variance $\langle v_i(t)^2 \rangle = \sigma^2 = 0.01 \text{ s}^{-2}$.”

Results (p.6, line 175):

Figure 1: The single circuit model adopted to study frontostriatal neural dynamics at rest. **A.** Schematic of the model derived from Wong & Wang (2002) and Deco et al. (2013), incorporating static frontostriatal (C_{21}) and stochastic striato-cortical (C_{12} , η_{12} , σ_{12}) coupling terms (Methods). **B.** Time series of the model state variables S_1 (frontal cortex), S_2 (striatum), and \tilde{C}_{12} (striato-cortical projection, with sign determining excitation or inhibition). **C.** State-space representation of the bistable frontal (S_1) and striatal (S_2) neural dynamics (i.e., average synaptic gating). Nullclines ($\frac{dS_1}{dt} = 0$, blue; and $\frac{dS_2}{dt} = 0$, green) intersections highlight stable fixed points (black circles, bottom left: low activity state; top right: high activity state) and an unstable fixed point (white circle). The trajectory (grey trace) is the resulting projection of S_1 and S_2 timeseries from panel A. **D.** Stability analysis of the model as a function of striato-cortical coupling (C_{12}). The two variables (cortical S_1 and striatal S_2 activity) exhibit stable (solid

and unstable (dash-dotted) equilibria separated by saddle-node bifurcations (SN1 and SN2 circles) that demarcate the ends of the bistable region. Background trajectories correspond to the projections of timeseries shown in panel A in S1 –C12 (blue) and S2 –C12 (red) spaces. **E. Functional connectivity** (Pearson’s R) between striatum and frontal cortex as a function of drift η_{12} and volatility σ_{12} parameters. **F. Transition rate** (number of zero-crossings in striato-cortical coupling per minute) as a function of drift η_{12} and volatility σ_{12} parameters.

We also highlighted the current challenges in understanding the physiological sources of the fMRI signal, emphasizing that it is still an area of active research:

Discussion (p.14, line 505):

“Fluctuations of the BOLD signal emerge from complex physiological and metabolic processes that are not yet fully understood (Mishra et al. 2021), including the signalling from excitatory and inhibitory neurons and other non-neuronal cells (Hall et al. 2016; Attwell et al. 2010; Howarth, Mishra, and Hall 2021). While changes in the level of synaptic activity do not account for all relevant mechanisms, the modelled non-linear responses of the observed hemodynamics has been widely validated (Friston et al. 2000) and shown to generate realistic resting-state BOLD timeseries with associated functional connectivity patterns (Deco et al. 2013; Breakspear 2017). Future modelling work on OCD could integrate other metabolic and synaptic processes contributing to the neurovascular coupling and the BOLD signal (Mathias et al. 2018; Dutta et al. 2023).”

Finally, the reviewer mentioned that the assumption that glutamatergic projections always have an excitatory effect on a given region is unfounded due to the density of synaptic interaction with both inhibitory and excitatory postsynaptic neurons. We clarified this point by the following changes in the text.

Methods (p.21, line 788):

“Note that while we assume cortico-cortical and cortico-striatal projections are glutamatergic and therefore excitatory in nature, the net effect of such excitatory projections also depends on the post-synaptic target. Accordingly, in regions with a large proportion of GABAergic targets (e.g., medium spiny neurons in the striatum (Koós and Tepper 1999)), the net effect of excitatory projections can become largely inhibitory.”

Moreover, as mentioned in the above response (R3-2), we re-performed the optimization utilizing a new model with relaxed priors, whereby glutamatergic projections can have inhibitory effects, i.e., negative couplings.

Methods (p.18, line 671):

“Nevertheless, to assess contributions of the thalamus in a confirmatory analysis, we created a six-populations model by including a thalamic nucleus between the striatal regions and the cortex in each pathway (Supplementary Figure 4). In this network, striato-thalamic projections are modeled with the stochastic couplings introduced above, while thalamo-cortical and cortico-striatal couplings remain non-dynamic. We also allowed cortico-striatal and thalamo-cortical couplings to be inhibitory, to include scenarios where glutamatergic neurons mainly

project post-synaptically to inhibitory neurons rather than pyramidal cells (Geisler and Wise 2008) (see section below about priors of the optimisation).”

We have seen that thalamo-cortical projections are not affected by this modification, not strongly differentiating OCD from controls. Cortico-striatal projections, however, showed differences that we now report.

Results, p.8, line 267:

“In this new model, cortico-striatal projections show opposite signs between the ventromedial and the dorsolateral circuits across both OCD and controls. Noticeably, those signs are reversed between the two conditions: excitatory ventromedial cortico-striatal coupling in controls becomes inhibitory in OCD, and inhibitory dorsolateral cortico-striatal coupling in controls becomes excitatory in OCD. This confirms an imbalance across those circuits in OCD compared to controls, and further supports that both circuits are affected antisymmetrically.”

4) As the authors point out the model could be used to describe many conditions where there is hypofrontality of DLPFC RSFC and hyperfrontality of mPFC with RSFC. The patient population used in this study (Hearne et al 2023) had significant differences between OCD patients and Controls in terms of depressive symptoms (MADRS) and anxiety (HADS). Although this may be unavoidable due to the high rates of comorbidity, It would be helpful to have the concurrent psychiatric diagnoses reiterated in this paper given that this may be relevant for neurocircuitry findings. It would also be relevant to comment on whether concurrent depressive symptoms were accounted for in the models (although this may limit ability to detect change). The discussion could benefit from a discussion of the issues related to this as well as how the findings of frontostriatal hyperconnectivity in OCD compare to other studies of depression and anxiety disorders

R3-4: As requested, we have reiterated that our clinical sample comprises individuals with a primary diagnosis of OCD (confirmed by a psychiatrist-led interview). The revised text also highlights that a psychiatrist assessed comorbid symptoms of anxiety and depression. Accordingly, we changed the text and expanded Table 1 to provide a more comprehensive description of the sample.

Methods (p.14, line 546):

“Fifty-two individuals diagnosed with OCD for at least 12 months showing moderate to severe symptoms (total Y-BOCS score greater than 14) were recruited across Australia (Table 1). A board-certified psychiatrist confirmed a primary diagnosis of OCD and assessed comorbid symptoms of anxiety and depression using validated clinical tools.”

Table 1: Sample characteristics.

	Controls (n=45)	OCD (n=52)
Age	32.5 (8.7)	30.2 (7.9)
Sex (% female)	40%	44%
Handedness (% right)	96%	85%
IQ-Full scale	112.7 (11.3)	106.0 (12.5)
Y-BOCS Total	1.8 (3.0)	25.3 (5.2)*
HAMA	2.9 (3.1)	19.5 (8.6) #

HADS Anxiety–Depression	4.8 (3.7) – 2.2 (2.3)	13.3 (4.7) – 8.2 (4.8) &^
MADRS	2.9 (3.5)	19.5 (10.4) +

Unless otherwise indicated, mean and standard deviation (S.D.) are reported. Y-BOCS: Yale-Brown Obsessive Compulsive Scale, IQ-Full scale: Wechsler Adult Intelligence Scale (WAIS-IV), MADRS: Montgomery-Asberg Depression rating scale; HAMA: Hamilton Anxiety Rating Scale; HADS: Hospital Anxiety and Depression scale. For digital twin analysis (n=48), pre–post, *Y-BOCS 25.1 (4.8) – 20.4 (6.1); #HAMA 19.9 (9.0) – 13.4 (8.6); &HADS Anxiety 13.0 (4.7) – 11.2 (4.0); ^HADS Depression 8.0 (4.8) – 6.3 (4.2); +MADRS 19.5 (10.5) – 15.3 (10.6).

We also performed additional control analyses assessing the possible relationships between changes in symptom severity measured by validated anxiety and depression scales and the change of frontostriatal functional connectivity in OCD. Details regarding these new control analyses have been added in the text with a supplementary figure:

Supplementary Figure 9: Absence of association between improvement in frontostriatal functional connectivity and other comorbid clinical measures. Improvement in frontostriatal functional connectivity (ΔFC) is calculated as the difference of FC distances (in FC space across ventromedial and dorsolateral circuits) to mean healthy controls' FC for each OCD subjects (at initial *minus* follow-up values). Likewise, improvements in clinical measures are calculated as the difference between symptoms at baseline and at follow-up. HAMA: Hamilton Anxiety Rating Scale; MADRS: Montgomery-Asberg Depression rating scale; HADS: Hospital Anxiety (Anx) and Depression (Dep) scale.

Results (p.12, line 405):

“In a complementary analysis, we show that those results are unlikely driven by medication status (Supplementary Figure 8), and that the observed functional connectivity changes in the ventromedial and dorsolateral circuits are not directly associated to other clinical measures besides OCD (Supplementary Figure 9).”

Discussion (p.12, line 433): “These findings confirm (Apergis-Schoute et al. 2018) and progress the understanding of the neural mechanisms relating key diagnostic and dimensional features of OCD, offering new targets for precision brain systems therapy to be probed in clinical trials.”

Due to the limitations noted above some of the authors conclusions are inadequately supported by the data. For example, it is unclear if the model strikes an optimal balance between biologic plausibility and complexity. The claim that the study “advances knowledge of neural mechanisms underpinning the

pathophysiology of OCD and informs the development of new precision treatments” is not supported by the model in this reviewer’s opinion.

R3-5: We sincerely hope that the extensive additional analyses and revision of the text have resolved the reviewer’s concerns. Regarding the specific example mentioned in this comment, the text has been revised to present a more nuanced view of our contribution, removing mentions of optimal balance between biological plausibility and complexity. We now highlight that our model represents a necessary step to develop more complex models that use neuroimaging modalities to interrogate the neural basis of altered macroscale circuit dynamics linked to the expression of OCD symptoms. Accordingly, the last paragraph of the Introduction has been changed as follows.

Introduction (p.3, line 90):

“By bridging scales from neural to behavioural changes in OCD, our work advances knowledge about the mechanisms underlying obsessions and compulsions. The presented study provides a solid basis for the development of more complex models interrogating OCD frontostriatal pathology and facilitating the progress towards new targeted interventions aiming to restore healthy frontostriatal dynamics.”

Importantly, to facilitate future investigations and ease the development of our model by colleagues, we made considerable efforts to provide to the community a self-contained execution environment reproducing all aspects of our study (i.e., a docker container with all necessary metadata and pre/post processing analysis). This environment was positively assessed by Reviewer #1 and is linked to an extensive online documentation. Further, we generated a cloud demo that includes a graphical user interface (GUI). This GUI allows less computationally oriented users to explore our model’s dynamics and generate digital twins linked to any given datasets. We updated the Data and Code availability section of the manuscript accordingly (p.22, line 816):

Data and code availability

The full code is available on Github at www.github.com/sebnaze/OCD-Modeling, with its detailed documentation at <https://ocd-modeling.readthedocs.io>. A docker container with all the necessary dependencies pre-installed and a demo of the model is included in this website. This allows the computation of digital twins on any given dataset. We also provide a lightweight cloud demo with a graphical user interface to facilitate the initial exploration of the model parameters. Pre-processed functional connectivity and behavioural scores are included in the repository. De-identified participant data for research purposes are available on request. Data can only be shared on request due to restrictions laid out in the study’s participant consent forms and QIMR Berghofer data-protection policies.

Reviewer #4 (Remarks to the Author):

Thank you for providing your joint feedback with another reviewer, it helped us to greatly improve the manuscript.

References

- Abramowitz, Jonathan S., Shannon M. Blakey, Lillian Reuman, and Jennifer L. Buchholz. 2018. "New Directions in the Cognitive-Behavioral Treatment of OCD: Theory, Research, and Practice." *Behavior Therapy* 49 (3): 311–22. <https://doi.org/10.1016/j.beth.2017.09.002>.
- Ahmari, Susanne E. 2016. "Using Mice to Model Obsessive Compulsive Disorder: From Genes to Circuits." *Neuroscience, Animal Models of Neuropsychiatric Disease*, 321 (May):121–37. <https://doi.org/10.1016/j.neuroscience.2015.11.009>.
- Ahmari, Susanne E., and Scott L. Rauch. 2022. "The Prefrontal Cortex and OCD." *Neuropsychopharmacology* 47 (1): 211–24.
- Ahmari, Susanne E., Timothy Spellman, Neria L. Douglass, Mazen A. Kheirbek, H. Blair Simpson, Karl Deisseroth, Joshua A. Gordon, and René Hen. 2013. "Repeated Cortico-Striatal Stimulation Generates Persistent OCD-Like Behavior." *Science* 340 (6137): 1234–39. <https://doi.org/10.1126/science.1234733>.
- Apergis-Schoute, Annemieke M., Bastiaan Bijleveld, Claire M. Gillan, Naomi A. Fineberg, Barbara J. Sahakian, and Trevor W. Robbins. 2018. "Hyperconnectivity of the Ventromedial Prefrontal Cortex in Obsessive-Compulsive Disorder." *Brain and Neuroscience Advances* 2 (January):2398212818808710. <https://doi.org/10.1177/2398212818808710>.
- Attwell, David, Alastair M. Buchan, Serge Charpak, Martin Lauritzen, Brian A. MacVicar, and Eric A. Newman. 2010. "Glial and Neuronal Control of Brain Blood Flow." *Nature* 468 (7321): 232–43. <https://doi.org/10.1038/nature09613>.
- Berke, Josh. 2018. "What Does Dopamine Mean?" *Nature Neuroscience* 21 (6): 787–93. <https://doi.org/10.1038/s41593-018-0152-y>.
- Bertino, Salvatore, Gianpaolo Antonio Basile, Alessia Bramanti, Giuseppe Pio Anastasi, Angelo Quartarone, Demetrio Milardi, and Alberto Cacciola. 2020. "Spatially Coherent and Topographically Organized Pathways of the Human Globus Pallidus." *Human Brain Mapping* 41 (16): 4641. <https://doi.org/10.1002/hbm.25147>.
- Breakspear, Michael. 2017. "Dynamic Models of Large-Scale Brain Activity." *Nature Neuroscience* 20 (3): 340–52. <https://doi.org/10.1038/nn.4497>.
- Burguière, Eric, Patricia Monteiro, Guoping Feng, and Ann M. Graybiel. 2013. "Optogenetic Stimulation of Lateral Orbitofronto-Striatal Pathway Suppresses Compulsive Behaviors." *Science (New York, N.Y.)* 340 (6137): 1243–46. <https://doi.org/10.1126/science.1232380>.
- Cash, Robin FH, Veronika I. Müller, Paul B. Fitzgerald, Simon B. Eickhoff, and Andrew Zalesky. 2023. "Altered Brain Activity in Unipolar Depression Unveiled Using Connectomics." *Nature Mental Health* 1 (3): 174–85.
- Cash, Robin FH, Anne Weigand, Andrew Zalesky, Shan H. Siddiqi, Jonathan Downar, Paul B. Fitzgerald, and Michael D. Fox. 2021. "Using Brain Imaging to Improve Spatial Targeting of Transcranial Magnetic Stimulation for Depression." *Biological Psychiatry* 90 (10): 689–700.
- Cocchi, Luca, Sebastien Naze, Conor Robinson, Lachlan Webb, Saurabh Sonkusare, Luke J. Hearne, Genevieve Whybird, et al. 2023. "Effects of Transcranial Magnetic Stimulation of the Rostromedial Prefrontal Cortex in Obsessive–Compulsive Disorder: A Randomized Clinical Trial." *Nature Mental Health* 1 (8): 555–63. <https://doi.org/10.1038/s44220-023-00094-0>.
- Cohen, Jonathan D., and David Servan-Schreiber. 1992. "Context, Cortex, and Dopamine: A Connectionist Approach to Behavior and Biology in Schizophrenia." *Psychological Review* 99 (1): 45.
- Daunizeau, Jean, Olivier David, and Klaas E. Stephan. 2011. "Dynamic Causal Modelling: A Critical Review of the Biophysical and Statistical Foundations." *Neuroimage* 58 (2): 312–22.
- Deco, Gustavo, Patric Hagmann, Anthony G. Hudetz, and Giulio Tononi. 2014. "Modeling Resting-State Functional Networks When the Cortex Falls Asleep: Local and Global Changes." *Cerebral Cortex (New York, N.Y.: 1991)* 24 (12): 3180–94. <https://doi.org/10.1093/cercor/bht176>.

- Deco, Gustavo, Adrián Ponce-Alvarez, Dante Mantini, Gian Luca Romani, Patric Hagmann, and Maurizio Corbetta. 2013. "Resting-State Functional Connectivity Emerges from Structurally and Dynamically Shaped Slow Linear Fluctuations." *Journal of Neuroscience* 33 (27): 11239–52.
- Deniau, J. M., and G. Chevalier. 1985. "Disinhibition as a Basic Process in the Expression of Striatal Functions. II. The Striato-Nigral Influence on Thalamocortical Cells of the Ventromedial Thalamic Nucleus." *Brain Research* 334 (2): 227–33. [https://doi.org/10.1016/0006-8993\(85\)90214-8](https://doi.org/10.1016/0006-8993(85)90214-8).
- Dominey, Peter F. 1995. "Complex Sensory-Motor Sequence Learning Based on Recurrent State Representation and Reinforcement Learning." *Biological Cybernetics* 73 (3): 265–74. <https://doi.org/10.1007/BF00201428>.
- Durstewitz, Daniel, and Gustavo Deco. 2008. "Computational Significance of Transient Dynamics in Cortical Networks." *The European Journal of Neuroscience* 27 (1): 217–27. <https://doi.org/10.1111/j.1460-9568.2007.05976.x>.
- Dutta, Shrey, Kartik K. Iyer, Sampsa Vanhatalo, Michael Breakspear, and James A. Roberts. 2023. "Mechanisms Underlying Pathological Cortical Bursts during Metabolic Depletion." *Nature Communications* 14 (1): 4792. <https://doi.org/10.1038/s41467-023-40437-0>.
- Eliasmith, Chris, Terrence C. Stewart, Xuan Choo, Trevor Bekolay, Travis DeWolf, Yichuan Tang, and Daniel Rasmussen. 2012. "A Large-Scale Model of the Functioning Brain." *Science* 338 (6111): 1202–5.
- Fettes, Peter, Laura Schulze, and Jonathan Downar. 2017. "Cortico-Striatal-Thalamic Loop Circuits of the Orbitofrontal Cortex: Promising Therapeutic Targets in Psychiatric Illness." *Frontiers in Systems Neuroscience* 11. <https://www.frontiersin.org/articles/10.3389/fnsys.2017.00025>.
- Figeé, Martijn, Judy Luigjes, Ruud Smolders, Carlos-Eduardo Valencia-Alfonso, Guido van Wingen, Bart de Kwaasteniet, Mariska Mantione, et al. 2013. "Deep Brain Stimulation Restores Frontostriatal Network Activity in Obsessive-Compulsive Disorder." *Nature Neuroscience* 16 (4): 386–87. <https://doi.org/10.1038/nn.3344>.
- Folloni, Davide. 2022. "Ultrasound Neuromodulation of the Deep Brain." *Science* 377 (6606): 589–589. <https://doi.org/10.1126/science.add4836>.
- Fradkin, Isaac, Casimir Ludwig, Eran Eldar, and Jonathan D. Huppert. 2020. "Doubting What You Already Know: Uncertainty Regarding State Transitions Is Associated with Obsessive Compulsive Symptoms." *PLOS Computational Biology* 16 (2): e1007634. <https://doi.org/10.1371/journal.pcbi.1007634>.
- Freyer, Frank, James A. Roberts, Petra Ritter, and Michael Breakspear. 2012. "A Canonical Model of Multistability and Scale-Invariance in Biological Systems." *PLOS Computational Biology* 8 (8): e1002634. <https://doi.org/10.1371/journal.pcbi.1002634>.
- Friston, K. J., L. Harrison, and W. Penny. 2003. "Dynamic Causal Modelling." *NeuroImage* 19 (4): 1273–1302. [https://doi.org/10.1016/S1053-8119\(03\)00202-7](https://doi.org/10.1016/S1053-8119(03)00202-7).
- Friston, Karl J., Joshua Kahan, Bharat Biswal, and Adeel Razi. 2014. "A DCM for Resting State fMRI." *Neuroimage* 94:396–407.
- Friston, Karl J., Vladimir Litvak, Ashwini Oswal, Adeel Razi, Klaas E. Stephan, Bernadette C. M. van Wijk, Gabriel Ziegler, and Peter Zeidman. 2016. "Bayesian Model Reduction and Empirical Bayes for Group (DCM) Studies." *NeuroImage* 128 (March):413–31. <https://doi.org/10.1016/j.neuroimage.2015.11.015>.
- Friston, Karl, Jérémy Mattout, Nelson Trujillo-Barreto, John Ashburner, and Will Penny. 2007. "Variational Free Energy and the Laplace Approximation." *NeuroImage* 34 (1): 220–34. <https://doi.org/10.1016/j.neuroimage.2006.08.035>.
- Gillies, Andrew, and Gordon Arbuthnott. 2000. "Computational Models of the Basal Ganglia." *Movement Disorders* 15 (5): 762–70. [https://doi.org/10.1002/1531-8257\(200009\)15:5<762::AID-MDS1002>3.0.CO;2-2](https://doi.org/10.1002/1531-8257(200009)15:5<762::AID-MDS1002>3.0.CO;2-2).
- Gollo, Leonardo L., James A. Roberts, and Luca Cocchi. 2017. "Mapping How Local Perturbations Influence Systems-Level Brain Dynamics." *NeuroImage, Functional Architecture of the Brain*, 160 (October):97–112. <https://doi.org/10.1016/j.neuroimage.2017.01.057>.

- Graybiel, A. M., and S. L. Rauch. 2000. "Toward a Neurobiology of Obsessive-Compulsive Disorder." *Neuron* 28 (2): 343–47. [https://doi.org/10.1016/s0896-6273\(00\)00113-6](https://doi.org/10.1016/s0896-6273(00)00113-6).
- Gurney, K., T. J. Prescott, and P. Redgrave. 2001a. "A Computational Model of Action Selection in the Basal Ganglia. I. A New Functional Anatomy." *Biological Cybernetics* 84 (6): 401–10. <https://doi.org/10.1007/PL00007984>.
- . 2001b. "A Computational Model of Action Selection in the Basal Ganglia. II. Analysis and Simulation of Behaviour." *Biological Cybernetics* 84 (6): 411–23. <https://doi.org/10.1007/PL00007985>.
- Gurney, Kevin N., Mark D. Humphries, and Peter Redgrave. 2015. "A New Framework for Cortico-Striatal Plasticity: Behavioural Theory Meets In Vitro Data at the Reinforcement-Action Interface." *PLOS Biology* 13 (1): e1002034. <https://doi.org/10.1371/journal.pbio.1002034>.
- Hall, Catherine N., Clare Howarth, Zebulun Kurth-Nelson, and Anusha Mishra. 2016. "Interpreting BOLD: Towards a Dialogue between Cognitive and Cellular Neuroscience." *Philosophical Transactions of the Royal Society B: Biological Sciences* 371 (1705): 20150348. <https://doi.org/10.1098/rstb.2015.0348>.
- Harrison, Ben J., Carles Soriano-Mas, Jesus Pujol, Hector Ortiz, Marina López-Solà, Rosa Hernández-Ribas, Joan Deus, et al. 2009. "Altered Corticostriatal Functional Connectivity in Obsessive-Compulsive Disorder." *Archives of General Psychiatry* 66 (11): 1189–1200. <https://doi.org/10.1001/archgenpsychiatry.2009.152>.
- Hearne, Luke J., Michael Breakspear, Ben J. Harrison, Caitlin V. Hall, Hannah S. Savage, Conor Robinson, Saurabh Sonkusare, et al. 2023. "Revisiting Deficits in Threat and Safety Appraisal in Obsessive-compulsive Disorder." *Human Brain Mapping* 44 (18): 6418–28. <https://doi.org/10.1002/hbm.26518>.
- Heuvel, Odile A. van den, Guido van Wingen, Carles Soriano-Mas, Pino Alonso, Samuel R. Chamberlain, Takashi Nakamae, Damiaan Denys, Anna E. Goudriaan, and Dick J. Veltman. 2016. "Brain Circuitry of Compulsivity." *European Neuropsychopharmacology* 26 (5): 810–27.
- Hollunder, Barbara, Jill L. Ostrem, Ilkem Aysu Sahin, Nanditha Rajamani, Simón Oxenford, Konstantin Butenko, Clemens Neudorfer, et al. 2024. "Mapping Dysfunctional Circuits in the Frontal Cortex Using Deep Brain Stimulation." *Nature Neuroscience*, February, 1–14. <https://doi.org/10.1038/s41593-024-01570-1>.
- Howarth, Clare, Anusha Mishra, and Catherine N. Hall. 2021. "More than Just Summed Neuronal Activity: How Multiple Cell Types Shape the BOLD Response." *Philosophical Transactions of the Royal Society B: Biological Sciences* 376 (1815): 20190630. <https://doi.org/10.1098/rstb.2019.0630>.
- Humphries, Mark D., and K. N. Gurney. 2002. "The Role of Intra-Thalamic and Thalamocortical Circuits in Action Selection." *Network: Computation in Neural Systems* 13 (1): 131–56.
- Jirsa, Viktor K., and Anthony R. McIntosh. 2007. *Handbook of Brain Connectivity*. Vol. 1. Springer.
- Joel, Daphna, Yael Niv, and Eytan Ruppin. 2002. "Actor–Critic Models of the Basal Ganglia: New Anatomical and Computational Perspectives." *Neural Networks* 15 (4–6): 535–47.
- Kenwood, Margaux M., Ned H. Kalin, and Helen Barbas. 2022. "The Prefrontal Cortex, Pathological Anxiety, and Anxiety Disorders." *Neuropsychopharmacology* 47 (1): 260–75. <https://doi.org/10.1038/s41386-021-01109-z>.
- Klaus, Andreas, Gabriela J. Martins, Vitor B. Paixao, Pengcheng Zhou, Liam Paninski, and Rui M. Costa. 2017. "The Spatiotemporal Organization of the Striatum Encodes Action Space." *Neuron* 95 (5): 1171–1180.e7. <https://doi.org/10.1016/j.neuron.2017.08.015>.
- Klinger, Emmanuel, and Jan Hasenauer. 2017. "A Scheme for Adaptive Selection of Population Sizes in Approximate Bayesian Computation - Sequential Monte Carlo." In *Computational Methods in Systems Biology*, edited by Jérôme Feret and Heinz Koepl, 128–44. Lecture Notes in Computer Science. Cham: Springer International Publishing. https://doi.org/10.1007/978-3-319-67471-1_8.
- Koós, Tibor, and James M. Tepper. 1999. "Inhibitory Control of Neostriatal Projection Neurons by GABAergic Interneurons." *Nature Neuroscience* 2 (5): 467–72. <https://doi.org/10.1038/8138>.
- Kozloski, James. 2016. "Closed-Loop Brain Model of Neocortical Information-Based Exchange." *Frontiers in Neuroanatomy* 10.

- Li, Ping, Xiangyun Yang, Andrew J. Greenshaw, Sufang Li, Jia Luo, Haiying Han, Jing Liu, et al. 2018. "The Effects of Cognitive Behavioral Therapy on Resting-state Functional Brain Network in Drug-naive Patients with Obsessive–Compulsive Disorder." *Brain and Behavior* 8 (5): e00963. <https://doi.org/10.1002/brb3.963>.
- Liu, Changliang, Pragma Goel, and Pascal S. Kaeser. 2021. "Spatial and Temporal Scales of Dopamine Transmission." *Nature Reviews. Neuroscience* 22 (6): 345–58. <https://doi.org/10.1038/s41583-021-00455-7>.
- Mason, S. J., and N. E. Graham. 2002. "Areas beneath the Relative Operating Characteristics (ROC) and Relative Operating Levels (ROL) Curves: Statistical Significance and Interpretation." *Quarterly Journal of the Royal Meteorological Society* 128 (584): 2145–66. <https://doi.org/10.1256/003590002320603584>.
- Mathias, Elshin J., Allanah Kenny, Michael J. Plank, and Tim David. 2018. "Integrated Models of Neurovascular Coupling and BOLD Signals: Responses for Varying Neural Activations." *NeuroImage* 174 (July):69–86. <https://doi.org/10.1016/j.neuroimage.2018.03.010>.
- Mattheisen, M., J. F. Samuels, Y. Wang, B. D. Greenberg, A. J. Fyer, J. T. McCracken, D. A. Geller, et al. 2015. "Genome-Wide Association Study in Obsessive–Compulsive Disorder: Results from the OCGAS." *Molecular Psychiatry* 20 (3): 337–44. <https://doi.org/10.1038/mp.2014.43>.
- Mishra, Anusha, Catherine N. Hall, Clare Howarth, and Ralph D. Freeman. 2021. "Key Relationships between Non-Invasive Functional Neuroimaging and the Underlying Neuronal Activity." *Philosophical Transactions of the Royal Society B: Biological Sciences* 376 (1815): 20190622. <https://doi.org/10.1098/rstb.2019.0622>.
- Mitchell, Anna S., and Subhojit Chakraborty. 2013. "What Does the Mediodorsal Thalamus Do?" *Frontiers in Systems Neuroscience* 7 (August). <https://doi.org/10.3389/fnsys.2013.00037>.
- Mondragón-González, Sirenia Lizbeth, Christiane Schreiweis, and Eric Burguière. 2024. "Closed-Loop Recruitment of Striatal Interneurons Prevents Compulsive-like Grooming Behaviors." *Nature Neuroscience* 27 (6): 1148–56. <https://doi.org/10.1038/s41593-024-01633-3>.
- Munn, Brandon R., Eli J. Müller, Itia Favre-Bulle, Ethan Scott, Joseph T. Lizier, Michael Breakspear, and James M. Shine. 2024. "Multiscale Organization of Neuronal Activity Unifies Scale-Dependent Theories of Brain Function." *Cell* 187 (25): 7303–7313.e15. <https://doi.org/10.1016/j.cell.2024.10.004>.
- Naze, Sebastien, Luke J Hearne, James A Roberts, Paula Sanz-Leon, Bjorn Burgher, Caitlin Hall, Saurabh Sonkusare, et al. 2023. "Mechanisms of Imbalanced Frontostriatal Functional Connectivity in Obsessive-Compulsive Disorder." *Brain* 146 (4): 1322–27. <https://doi.org/10.1093/brain/awac425>.
- Olatunji, Bunmi O., Michelle L. Davis, Mark B. Powers, and Jasper A. J. Smits. 2013. "Cognitive-Behavioral Therapy for Obsessive-Compulsive Disorder: A Meta-Analysis of Treatment Outcome and Moderators." *Journal of Psychiatric Research* 47 (1): 33–41. <https://doi.org/10.1016/j.jpsychires.2012.08.020>.
- Ownby, Raymond L. 1998. "Computational Model of Obsessive-Compulsive Disorder: Examination of Etiologic Hypothesis and Treatment Strategies." *Depression and Anxiety* 8 (3): 91–103. [https://doi.org/10.1002/\(SICI\)1520-6394\(1998\)8:3<91::AID-DA1>3.0.CO;2-Q](https://doi.org/10.1002/(SICI)1520-6394(1998)8:3<91::AID-DA1>3.0.CO;2-Q).
- Pang, James C, James K Rilling, James A Roberts, Martijn P van den Heuvel, and Luca Cocchi. 2022. "Evolutionary Shaping of Human Brain Dynamics." Edited by Claus Hilgetag, Christian Büchel, and Bratislav Misic. *eLife* 11 (October):e80627. <https://doi.org/10.7554/eLife.80627>.
- Paraskevopoulou, Foteini, Melissa A. Herman, and Christian Rosenmund. 2019. "Glutamatergic Innervation onto Striatal Neurons Potentiates GABAergic Synaptic Output." *The Journal of Neuroscience* 39 (23): 4448–60. <https://doi.org/10.1523/JNEUROSCI.2630-18.2019>.
- Pennartz, Cyriel M. A., Joshua D. Berke, Ann M. Graybiel, Rutsuko Ito, Carien S. Lansink, Matthijs van der Meer, A. David Redish, Kyle S. Smith, and Pieter Voorn. 2009. "Corticostriatal Interactions during Learning, Memory Processing, and Decision Making." *Journal of Neuroscience* 29 (41): 12831–38. <https://doi.org/10.1523/JNEUROSCI.3177-09.2009>.

- Penny, Will D., Klaas E. Stephan, Jean Daunizeau, Maria J. Rosa, Karl J. Friston, Thomas M. Schofield, and Alex P. Leff. 2010. "Comparing Families of Dynamic Causal Models." *PLOS Computational Biology* 6 (3): e1000709. <https://doi.org/10.1371/journal.pcbi.1000709>.
- Peters, Sarah K., Katharine Dunlop, and Jonathan Downar. 2016. "Cortico-Striatal-Thalamic Loop Circuits of the Salience Network: A Central Pathway in Psychiatric Disease and Treatment." *Frontiers in Systems Neuroscience*, 104. <https://doi.org/10.3389/fnsys.2016.00104>.
- Piantadosi, Sean C., Elizabeth E. Manning, Brittany L. Chamberlain, James Hyde, Zoe LaPalombara, Nicholas M. Bannon, Jamie L. Pierson, Vijay M. K Nambodiri, and Susanne E. Ahmari. 2024. "Hyperactivity of Indirect Pathway-Projecting Spiny Projection Neurons Promotes Compulsive Behavior." *Nature Communications* 15 (1): 4434. <https://doi.org/10.1038/s41467-024-48331-z>.
- Rabinovich, Mikhail I., Ramón Huerta, Pablo Varona, and Valentin S. Afraimovich. 2008. "Transient Cognitive Dynamics, Metastability, and Decision Making." *PLOS Computational Biology* 4 (5): e1000072. <https://doi.org/10.1371/journal.pcbi.1000072>.
- Reiner, Anton, Natalie M. Hart, Wanlong Lei, and Yunping Deng. 2010. "Corticostriatal Projection Neurons – Dichotomous Types and Dichotomous Functions." *Frontiers in Neuroanatomy* 4 (October). <https://doi.org/10.3389/fnana.2010.00142>.
- Robbins, Trevor W., Paula Banca, and David Belin. 2024. "From Compulsivity to Compulsion: The Neural Basis of Compulsive Disorders." *Nature Reviews Neuroscience* 25 (5): 313–33. <https://doi.org/10.1038/s41583-024-00807-z>.
- Robbins, Trevor W., Matilde M. Vaghi, and Paula Banca. 2019. "Obsessive-Compulsive Disorder: Puzzles and Prospects." *Neuron* 102 (1): 27–47. <https://doi.org/10.1016/j.neuron.2019.01.046>.
- Roberts, James A., Karl J. Friston, and Michael Breakspear. 2017. "Clinical Applications of Stochastic Dynamic Models of the Brain, Part II: A Review." *Biological Psychiatry: Cognitive Neuroscience and Neuroimaging* 2 (3): 225–34.
- Rumelhart, David E., James L. McClelland, and CORPORATE PDP Research Group. 1986. *Parallel Distributed Processing: Explorations in the Microstructure of Cognition, Vol. 1: Foundations*. MIT press.
- Sato, Reo, Kanji Shimomura, and Kenji Morita. 2023. "Opponent Learning with Different Representations in the Cortico-Basal Ganglia Pathways Can Develop Obsession-Compulsion Cycle." *PLOS Computational Biology* 19 (6): e1011206. <https://doi.org/10.1371/journal.pcbi.1011206>.
- Saxena, Sanjaya, Arthur L Brody, Karron M Maidment, Jennifer J Dunkin, Mark Colgan, Shervin Alborzian, Michael E Phelps, and Lewis R Baxter. 1999. "Localized Orbitofrontal and Subcortical Metabolic Changes and Predictors of Response to Paroxetine Treatment in Obsessive-Compulsive Disorder." *Neuropsychopharmacology* 21 (6): 683–93. [https://doi.org/10.1016/S0893-133X\(99\)00082-2](https://doi.org/10.1016/S0893-133X(99)00082-2).
- Segal, Ashlea, Linden Parkes, Kevin Aquino, Seyed Mostafa Kia, Thomas Wolfers, Barbara Franke, Martine Hoogman, Christian F. Beckmann, Lars T. Westlye, and Ole A. Andreassen. 2023. "Regional, Circuit and Network Heterogeneity of Brain Abnormalities in Psychiatric Disorders." *Nature Neuroscience*, 1–17.
- Shephard, Elizabeth, Emily R. Stern, Odile A. van den Heuvel, Daniel L. C. Costa, Marcelo C. Batistuzzo, Priscilla B. G. Godoy, Antonio C. Lopes, et al. 2021. "Toward a Neurocircuit-Based Taxonomy to Guide Treatment of Obsessive-Compulsive Disorder." *Molecular Psychiatry* 26 (9): 4583–4604. <https://doi.org/10.1038/s41380-020-01007-8>.
- Shepherd, Gordon MG. 2013. "Corticostriatal Connectivity and Its Role in Disease." *Nature Reviews Neuroscience* 14 (4): 278.
- Shine, James M. 2021. "The Thalamus Integrates the Macrosystems of the Brain to Facilitate Complex, Adaptive Brain Network Dynamics." *Progress in Neurobiology* 199:101951.
- Sidibe, Mamadou, Mark D. Bevan, J. Paul Bolam, and Yoland Smith. 1997. "Efferent Connections of the Internal Globus Pallidus in the Squirrel Monkey: I. Topography and Synaptic Organization of the Pallidothalamic Projection." *Journal of Comparative Neurology* 382 (3): 323–47.

- Sip, Viktor, Meysam Hashemi, Timo Dickscheid, Katrin Amunts, Spase Petkoski, and Viktor Jirsa. 2023. "Characterization of Regional Differences in Resting-State FMRI with a Data-Driven Network Model of Brain Dynamics." *Science Advances* 9 (11): eabq7547. <https://doi.org/10.1126/sciadv.abq7547>.
- Smaers, Jeroen B., Aida Gómez-Robles, Ashley N. Parks, and Chet C. Sherwood. 2017. "Exceptional Evolutionary Expansion of Prefrontal Cortex in Great Apes and Humans." *Current Biology* 27 (5): 714–20. <https://doi.org/10.1016/j.cub.2017.01.020>.
- Spreizer, Sebastian, Martin Angelhuber, Jyotika Bahuguna, Ad Aertsen, and Arvind Kumar. 2017. "Activity Dynamics and Signal Representation in a Striatal Network Model with Distance-Dependent Connectivity." *ENeuro* 4 (4): 0348–16. <https://doi.org/10.1523/ENEURO.0348-16.2017>.
- Stefanie Geisler, and Roy A. Wise. 2008. "Functional Implications of Glutamatergic Projections to the Ventral Tegmental Area." *Reviews in the Neurosciences* 19 (4–5): 227–44. <https://doi.org/10.1515/REVNEURO.2008.19.4-5.227>.
- Stein, Dan J., Daniel L. C. Costa, Christine Lochner, Euripedes C. Miguel, Y. C. Janardhan Reddy, Roseli G. Shavitt, Odile A. van den Heuvel, and H. Blair Simpson. 2019. "Obsessive–Compulsive Disorder." *Nature Reviews Disease Primers* 5 (1): 1–21. <https://doi.org/10.1038/s41572-019-0102-3>.
- Stein, Dan J., and Eric Hollander. 1994. "A Neural Network Approach to Obsessive-Compulsive Disorder." *The Journal of Mind and Behavior* 15 (3): 223–37.
- Steiner, Heinz, and Kuei Y. Tseng. 2016. *Handbook of Basal Ganglia Structure and Function*. Academic Press.
- Swanson, Larry W., Olaf Sporns, and Joel D. Hahn. 2019. "The Network Organization of Rat Intrathalamic Macroconnections and a Comparison with Other Forebrain Divisions." *Proceedings of the National Academy of Sciences* 116 (27): 13661–69. <https://doi.org/10.1073/pnas.1905961116>.
- Tomkins, Adam, Eleni Vasilaki, Christian Beste, Kevin Gurney, and Mark D. Humphries. 2014. "Transient and Steady-State Selection in the Striatal Microcircuit." *Frontiers in Computational Neuroscience* 7 (January). <https://doi.org/10.3389/fncom.2013.00192>.
- Toni, Tina, David Welch, Natalja Strelkowa, Andreas Ipsen, and Michael P.H. Stumpf. 2008. "Approximate Bayesian Computation Scheme for Parameter Inference and Model Selection in Dynamical Systems." *Journal of The Royal Society Interface* 6 (31): 187–202. <https://doi.org/10.1098/rsif.2008.0172>.
- Van Essen, David C., Stephen M. Smith, Deanna M. Barch, Timothy EJ Behrens, Essa Yacoub, Kamil Ugurbil, and Wu-Minn HCP Consortium. 2013. "The WU-Minn Human Connectome Project: An Overview." *Neuroimage* 80:62–79.
- Vollweiler, Dennis, Jasmeet Kaur Shergill, Alexandra Hilse, Gaga Kochlamazashvili, Stefan Paul Koch, Susanne Mueller, Philipp Boehm-Sturm, Volker Haucke, and Tanja Maritzen. 2023. "Intersectin Deficiency Impairs Cortico-Striatal Neurotransmission and Causes Obsessive–Compulsive Behaviors in Mice." *Proceedings of the National Academy of Sciences* 120 (35): e2304323120. <https://doi.org/10.1073/pnas.2304323120>.
- Voorn, Pieter, Louk J. M. J. Vanderschuren, Henk J. Groenewegen, Trevor W. Robbins, and Cyriel M. A. Pennartz. 2004. "Putting a Spin on the Dorsal–Ventral Divide of the Striatum." *Trends in Neurosciences* 27 (8): 468–74. <https://doi.org/10.1016/j.tins.2004.06.006>.
- Wang, Huifang E., Marmaduke Woodman, Paul Triebkorn, Jean-Didier Lemarechal, Jayant Jha, Borana Dollomaja, Anirudh Nihalani Vattikonda, et al. 2023. "Delineating Epileptogenic Networks Using Brain Imaging Data and Personalized Modeling in Drug-Resistant Epilepsy." *Science Translational Medicine* 15 (680): eabp8982. <https://doi.org/10.1126/scitranslmed.abp8982>.
- Wang, Xiao-Jing. 2008. "Decision Making in Recurrent Neuronal Circuits." *Neuron* 60 (2): 215–34. <https://doi.org/10.1016/j.neuron.2008.09.034>.
- Welch, Jeffrey M., Jing Lu, Ramona M. Rodriguiz, Nicholas C. Trotta, Joao Peca, Jin-Dong Ding, Catia Feliciano, et al. 2007. "Cortico-Striatal Synaptic Defects and OCD-like Behaviours in Sapap3-Mutant Mice." *Nature* 448 (7156): 894–900. <https://doi.org/10.1038/nature06104>.
- Wong, Kong-Fatt, and Xiao-Jing Wang. 2006. "A Recurrent Network Mechanism of Time Integration in Perceptual Decisions." *Journal of Neuroscience* 26 (4): 1314–28. <https://doi.org/10.1523/JNEUROSCI.3733-05.2006>.

- Yin, Lining, Fang Han, Ying Yu, and Qingyun Wang. 2023. "A Computational Network Dynamical Modeling for Abnormal Oscillation and Deep Brain Stimulation Control of Obsessive–Compulsive Disorder." *Cognitive Neurodynamics* 17 (5): 1167–84. <https://doi.org/10.1007/s11571-022-09858-3>.
- Yin, Lining, Ying Yu, Fang Han, and Qingyun Wang. 2024. "Unveiling Serotonergic Dysfunction of Obsessive-Compulsive Disorder on Prefrontal Network Dynamics: A Computational Perspective." *Cerebral Cortex (New York, N.Y.: 1991)* 34 (6): bhae258. <https://doi.org/10.1093/cercor/bhae258>.

REVIEWER COMMENTS

We are grateful to the reviewers for their overall appreciation of our work. We address the remaining minor comments from Reviewer #3 below. Our response is in bold characters with indentation, and changes made in the manuscript are in blue.

Reviewer #3 (Remarks to the authors):

The authors have made a herculean effort to address this reviewers concerns. They have succeeded in articulating limitations of the model and have added numerous additional analyses that speak to the issues related to the empirical data on which the model is based. They have tempered their conclusions on how the model could inform new circuit based treatments. Overall a most impressive revision that in my opinion justifies publication. A couple of additional questions/concerns that if addressed would further strengthen the manuscript in my opinion.

We thank the reviewer for the positive appraisal of our previous revision.

1) Additional data using a less conservative statistical threshold has addressed inconsistencies between the Harrison et al and Naze et al data sets in terms of the hyperconnectivity of the anterior OFC and ACC. However it remains unclear why the main finding of the Naze et al study in the R frontal pole was not also found in the Harrison study given the less conservative analysis in the Harrison study.

R1: We believe that this inconsistency between the two studies is likely due to the complexity of imaging the frontal pole (e.g., distortions, signal dropout). Harrison et al. (2009) used a 1.5T scanner with a single-band sequence. Our study used a modern 3T scanner with an optimised multiband sequence to maximise the signal-to-noise ratio in the frontal pole. As demonstrated by our additional analyses (Supplementary Figures 1 and 2), these discrepancies are reconciled when considering a system-based assessment of the group differences.

2) While there is substantial evidence supporting RSFC differences in the ventromedial circuit, the data supporting hypoconnectivity in the resting state in the caudal VLPFC/insula found in the Naze study is less clear. A review of the Robbins Shepard and Von Heuvel papers seemed to substantiate a larger swath of mostly DLPFC caudate hypoconnectivity. The caudal VLPFC/opercular.anterior insula area is difficult in terms of low SNR and is an area that appears to be at the intersection of the salience language and premotor areas. Clarification of which areas have been proven to be hypofunctional in other studies in the lateral prefrontal cortex and what role this area might play in the genesis of OCD symptoms would be useful additions.

R2: We agree with the Reviewer that more studies to date have shown OCD frontostriatal *hyperconnectivity* in the ventromedial circuit compared to *hypoconnectivity* in the dorsolateral circuit. However, there has been growing evidence supporting the latter deregulation. This aligns with the view that there is a broader (and likely subject-specific) frontostriatal system dysregulation. Results summarised in a recent review (Robbins et al. Nature Reviews Neuroscience, 2024) support the hypothesis of *hypoconnectivity* between the caudal striatum/putamen and the dorsolateral prefrontal cortex. Nonetheless, beyond a system-level view of the current results, it remains hard to reliably isolate defined sub-regions within the lateral prefrontal cortex linked to OCD and its genesis.

To emphasise this point, we modified the Discussion as follows (line 469-470):

“We here note that the reported peak location of the ventromedial hyperconnectivity and dorsolateral hypoconnectivity differ across studies (see Supplementary Figures 1 and 2) and most likely subjects. However, these functional clusters belong to the same canonical networks thought to support different symptomatic dimensions of OCD^{2,3,13,14}. **Future studies are required to reliably define prefrontal regions underlying OCD and its symptom severity.**”